# Neuronal wiring diagram of an adult brain

Sven Dorkenwald[1,2], Arie Matsliah[1], Amy R. Sterling[1,3], Philipp Schlegel[4,5], Szi-chieh Yu[1], Claire E. McKellar[1], Albert Lin[1,6], Marta Costa[5], Katharina Eichler[5], Yijie Yin[5], Will Silversmith[1], Casey Schneider-Mizell[7], Chris S. Jordan[1], Derrick Brittain[7], Akhilesh Halageri[1], Kai Kuehner[1], Oluwaseun Ogedengbe[1], Ryan Morey[1], Jay Gager[1], Krzysztof Kruk[3], Eric Perlman[8], Runzhe Yang[1,2], David Deutsch[1,9], Doug Bland[1], Marissa Sorek[1,3], Ran Lu[1], Thomas Macrina[1,2], Kisuk Lee[1,10], J. Alexander Bae[1,11], Shang Mu[1], Barak Nehoran[1,2], Eric Mitchell[1], Sergiy Popovych[1,2], Jingpeng Wu[1], Zhen Jia[1], Manuel A. Castro[1], Nico Kemnitz[1], Dodam Ih[1], Alexander Shakeel Bates[4,12,13], Nils Eckstein[14], Jan Funke[14], Forrest Collman[7], Davi D. Bock[15], Gregory S. X. E. Jefferis[4,5], H. Sebastian Seung[1,2✉], Mala Murthy[1✉] & The FlyWire Consortium*

Connections between neurons can be mapped by acquiring and analysing electron microscopic brain images. In recent years, this approach has been applied to chunks of brains to reconstruct local connectivity maps that are highly informative[1–6], but nevertheless inadequate for understanding brain function more globally. Here we present a neuronal wiring diagram of a whole brain containing $5 \times 10^7$ chemical synapses[7] between 139,255 neurons reconstructed from an adult female *Drosophila melanogaster*[8,9]. The resource also incorporates annotations of cell classes and types, nerves, hemilineages and predictions of neurotransmitter identities[10–12]. Data products are available for download, programmatic access and interactive browsing and have been made interoperable with other fly data resources. We derive a projectome—a map of projections between regions—from the connectome and report on tracing of synaptic pathways and the analysis of information flow from inputs (sensory and ascending neurons) to outputs (motor, endocrine and descending neurons) across both hemispheres and between the central brain and the optic lobes. Tracing from a subset of photoreceptors to descending motor pathways illustrates how structure can uncover putative circuit mechanisms underlying sensorimotor behaviours. The technologies and open ecosystem reported here set the stage for future large-scale connectome projects in other species.

Although rudimentary nervous systems existed in more ancient animals, brains evolved around half a billion years ago[13], and are essential for the generation of sophisticated behaviours. It is widely accepted that dividing a brain into regions is helpful for understanding brain function, but questions remain on the utility of finer-grain information about connectivity. In fact, efforts to construct wiring diagrams at the level of neurons and synapses have been controversial[14,15]. Scepticism has flourished largely owing to a lack of technologies that could reconstruct such wiring diagrams[16,17], so obtaining such diagrams has remained hypothetical. The situation began to change in the 2000s owing to the efforts of a small community of researchers. Here we present a neuronal wiring diagram of a whole adult brain and, here and in the accompanying studies, we analyse its connectivity to highlight the utility of this endeavour.

Although small, the brain of *D. melanogaster* contains $10^5$ neurons and $10^8$ synapses that enable a fly to see, smell, hear, walk and fly. Flies engage in dynamic social interactions[18], navigate over distances[19] and form long-term memories[20]. Portions of fly brains have been reconstructed from electron microscopy images, which have sufficient resolution to reveal the fine branches of neurons and the synapses that connect them. The resulting wiring diagrams of neural circuits have provided crucial insights into how the brain generates social[21,22], memory-related[23] or navigation[24] behaviours. Wiring diagrams of other fly brain regions have been mapped and related to visual[2], auditory[25] and olfactory[23,26] functions. The circuit organization revealed by these wiring diagrams show similarities to mammalian brains[27,28].

These wiring diagrams and many others from mammals[4–6] have been derived from pieces of brain. However, recordings of *Drosophila* neural activity have revealed nearly brain-wide encoding of sensory[29] and motor[30] variables. These studies and others in vertebrates highlight that understanding how the brain processes sensory information or drives behaviour will require understanding global information flow at the scale of the entire brain.

[1]Princeton Neuroscience Institute, Princeton University, Princeton, NJ, USA. [2]Computer Science Department, Princeton University, Princeton, NJ, USA. [3]Eyewire, Boston, MA, USA. [4]Neurobiology Division, MRC Laboratory of Molecular Biology, Cambridge, UK. [5]Drosophila Connectomics Group, Department of Zoology, University of Cambridge, Cambridge, UK. [6]Center for the Physics of Biological Function, Princeton University, Princeton, NJ, USA. [7]Allen Institute for Brain Science, Seattle, WA, USA. [8]Yikes LLC, Baltimore, MD, USA. [9]Department of Neurobiology, University of Haifa, Haifa, Israel. [10]Brain and Cognitive Sciences Department, Massachusetts Institute of Technology, Cambridge, MA, USA. [11]Electrical and Computer Engineering Department, Princeton University, Princeton, NJ, USA. [12]Harvard Medical School, Boston, MA, USA. [13]Centre for Neural Circuits and Behaviour, The University of Oxford, Oxford, UK. [14]Janelia Research Campus, Howard Hughes Medical Institute, Ashburn, VA, USA. [15]Department of Neurological Sciences, Larner College of Medicine, University of Vermont, Burlington, VT, USA. *A full list of members and their affiliations appears in the Supplementary Information. A list of authors and their affiliations appears at the end of the paper. ✉e-mail: sseung@princeton.edu; mmurthy@princeton.edu

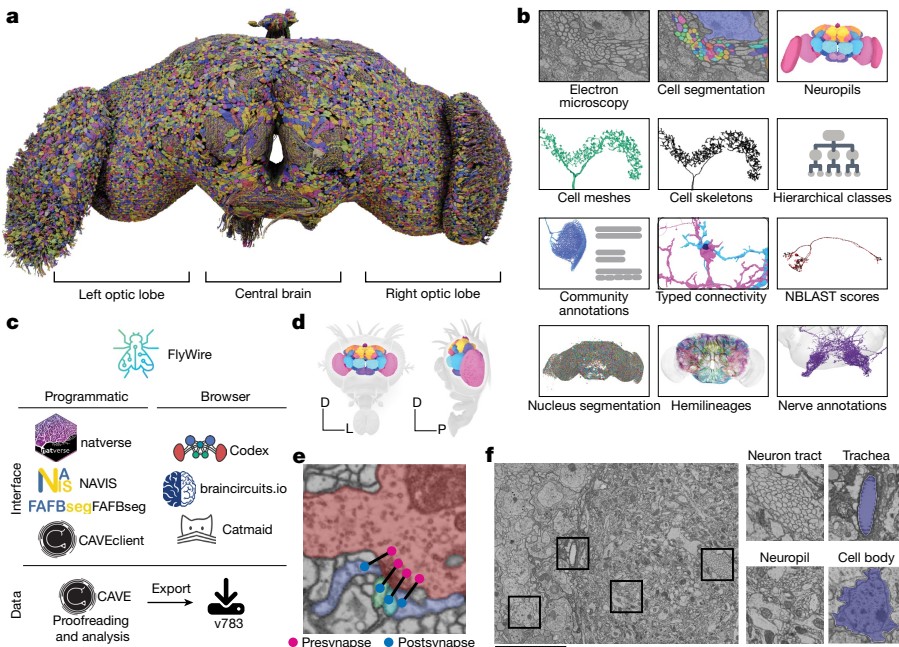

**Fig. 1 | A connectomic reconstruction of a whole fly brain. a**, All neuron morphologies reconstructed with FlyWire. All neurons in the central brain and both optic lobes were segmented and proofread. Note that image and dataset are mirror inverted relative to the native fly brain. **b**, An overview of many of the FlyWire resources that are being made available. FlyWire leverages existing resources for electron microscopy imagery by Zheng et al.[9], synapse predictions by Buhmann et al.[7] and Heinrich et al.[118], and neurotransmitter predictions by Eckstein et al.[10]. Annotations of the FlyWire brain dataset such as hemilineages, nerves and hierarchical classes are established in the accompanying paper[12]. **c**, FlyWire uses CAVE[50] for proofreading, data management and analysis back end. The data can be accessed programmatically through CAVEclient, navis, fafbseg and natverse[119], and through the browser in Codex, Catmaid Spaces and braincircuits.io. Static exports of the data are also available. **d**, The *Drosophila* brain can be divided into spatially defined regions based on neuropils[80] (Extended Data Fig. 1). Neuropils for the lamina are not shown. D, dorsal; L, lateral; P, posterior. **e**, Synaptic boutons in the fly brain are often polyadic such that there are multiple postsynaptic partners per presynaptic bouton. Each link between a pre- and a postsynaptic location is a synapse. **f**, Neuron tracts, trachea, neuropil and cell bodies can be readily identified from the electron microscopy data acquired by Zheng et al.[9]. Scale bar, 10 μm.

Until now, the closest antecedent to a wiring diagram of the whole brain has been the reconstruction of a fly 'hemibrain'[1], a pioneering resource that has already become indispensable to *Drosophila* researchers. It is estimated to contain around 20,000 neurons that are 'uncropped'—that is, minimally truncated by the borders of the imaged volume, and 14 million synapses between them. Our reconstruction of an entire adult brain contains 139,255 neurons (Fig. 1a and Supplementary Video 1) and 54.5 million synapses between these neurons. To aid exploration and analysis, this connectome has been densely annotated by the FlyWire Consortium. In our companion paper, Schlegel et al.[12] provide a curated brain-wide hierarchy of annotations including more than 8,400 distinct cell types, completing the description of this resource (and should therefore preferably be cited alongside this paper; https://codex.flywire.ai/about_flywire). These and many other data products (Fig. 1b and Supplementary Fig. 1) are available for download, programmatic access and interactive browsing and have been made interoperable with other fly data resources through a growing ecosystem of software tools (Fig. 1c). The primary portal to the data is the FlyWire Connectome Data Explorer (Codex; https://codex.flywire.ai/), which makes the information visualizable and queryable.

The wiring diagram from our whole-brain reconstruction is sufficiently complete to be designated a 'connectome' (defined in Discussion). It represents substantial progress over neuronal reconstructions of *Caenorhabditis elegans*[31,32] (300 neurons, 10⁴ synapses) and the 1st instar larva of *Drosophila*[33] (3,000 neurons, 5 × 10⁵ synapses). Our connectome advances beyond the hemibrain in several ways. For example, it includes the suboesophageal zone (SEZ) of the central brain, which is important for diverse functions such as gustation and mechanosensation[34,35], and contains many of the processes of neurons that descend from the brain to the ventral nerve cord to drive motor

behaviours. Additionally, it includes annotations for descending and ascending neurons[36] for many sexually dimorphic neurons (analysed by Deutsch et al. (manuscript in preparation); available at https://codex.flywire.ai) and an entire optic lobe[11]. Our reconstruction of both optic lobes goes far beyond existing maps of columnar visual circuitry. Connections between the optic lobes and central brain are included, as explored by refs. 37,38. Also included are neurons that extend into the brain through the nerves and neck connective, which are essential for tracing sensorimotor pathways, as illustrated here and in the accompanying studies[11,12,34,36–45].

Our reconstruction utilized image acquisition and analysis techniques that are distinct from those used for the hemibrain (Methods and Discussion). However, we have built directly on the hemibrain in an important way. Schlegel et al.[12] used the cell types proposed for the hemibrain as a starting point for cell typing neurons in the central brain in FlyWire. This approach was enabled by a growing ecosystem of software tools serving interoperability between different fly data sources (Fig. 1c). Additional annotations in the SEZ and optic lobes, which are largely absent from the hemibrain, were contributed by *Drosophila* research groups in the FlyWire Consortium as well as citizen scientists, and are described in more detail here and in the accompanying papers. Synapse predictions[7] and estimates of neurotransmitter identities[10] were also contributed by the community.

After matching, Schlegel et al.[12] also compared our wiring diagram with the hemibrain where they overlap and showed that cell-type counts and strong connections were largely in agreement. This means that the combined effects of natural variability across individuals and 'noise' due to imperfect reconstruction tend to be modest, so our wiring diagram of a single brain should be useful for studying any wild-type *Drosophila melanogaster* individual. However, there are known differences

between the brains of male and female flies[46]. In addition, principal neurons of the mushroom body, a brain structure required for olfactory learning and memory, show high variability[12]. Some mushroom body connectivity patterns have even been found to be near random[47], although deviations from randomness have since been identified[48]. In short, *Drosophila* wiring diagrams are useful because of their stereotypy, yet also open the door to studies of connectome variation.

In addition to describing the FlyWire brain resource, this Article also presents analyses that illustrate how the data products can be used. Additional whole-brain network analyses are provided by Lin et al.[49] and Pospisil et al.[39]. From the connectome, we derive a projectome, a reduced map of projections between 78 fly brain regions known as neuropils (Fig. 1d, Extended Data Fig. 1 and Supplementary Video 2). We trace synaptic pathways and analyse information flow from the inputs to the outputs of the brain, across both hemispheres, and between the central brain and the optic lobes. In particular, the organization of excitation and inhibition in pathways from photoreceptors in the ocelli to descending motor neurons immediately suggests hypotheses about circuit mechanisms of behaviour.

## Reconstruction of a whole fly brain

Images of an entire adult female fly brain (Fig. 1e,f) were previously acquired by serial section transmission electron microscopy and released into the public domain by Zheng et al.[9]. We previously realigned the electron microscopy images, automatically segmented all neurons in the images, created a computational system that allows interactive proofreading of the segmentation[50], and assembled an online community[8] (FlyWire). During the initial phase, much of the proofreading was done by a distributed community of *Drosophila* research groups in the FlyWire Consortium, and focused on neurons of interest to these groups. During the later phase, the remaining neurons were mainly proofread by centralized teams at Princeton and Cambridge, with contributions from citizen scientists worldwide. The recruitment and training of proofreaders and their workflows are described in the Methods.

Chemical synapses were automatically detected in the images as pairs of presynapse–postsynapse locations[7]. The whole brain contains 0.0175 mm³ of neuropil volume and around 130 million synapses. This equates to 7.4 synapses per μm³, a much higher density than that of mammalian cortex[51,52] (less than 1 synapse per μm³). The central brain and left and right optic lobes (including the lamina) contain 0.0103, 0.0036 and 0.0036 mm³ of neuropil volume, respectively, with synapse counts in approximately the same proportion. Synapses were combined with proofread neurons using the Connectome Annotation Versioning Engine[50] (CAVE) to yield the connectome.

We next assessed completeness and accuracy of proofreading. We had already shown that FlyWire proofreading can yield accurate results[8] through comparison with light microscopic reconstructions of neurons that are known to be highly stereotyped across individual flies. A second method is to subject reconstructed neurons to an additional round of proofreading, which was previously shown to yield few changes[8]. Because proofreading workflows and personnel have changed over time, and accuracy can vary across brain regions, we repeated this evaluation by subjecting 826 neurons from the central brain to a further round of proofreading. Relative to this additional round, our proofread dataset achieved an average $F_1$ score of 99.2% by volume (Extended Data Fig. 2a,b).

By quantifying how many of the automatically detected synapses are attached to proofread segments, as opposed to being isolated in tiny 'orphan' segments, we can estimate completeness of the proofreading. We found high attachment rates of presynapses (approximately 122 million presynapses (93.7%) attached), whereas attachment rates of postsynapses were lower (approximately 58.1 million postsynapses (44.7%) attached) owing to less proofreading and reattachment of twigs, which

contain most of the postsynapses[8] (Extended Data Fig. 2c,d). Attachment rates were generally in agreement between the two hemispheres of FlyWire and with the hemibrain (Extended Data Fig. 2e–g) and varied by neuropil (Supplementary Fig. 2). As with the hemibrain[1], false negative synapses are the dominant type of error but false positives also exist. For this reason, analyses using the connectome should consider thresholding to remove spurious connections. Thresholds should be adjusted to the individual analyses. For the analyses presented below (and connections indicated at https://codex.flywire.ai), we use a threshold of five synapses to determine a connection between two neurons. The accompanying paper by Matsliah et al.[11] found a threshold of two synapses appropriate for analysing connections in the optic lobes. Assuming that such errors are statistically independent, accuracy is expected to be high for detection of connections involving multiple synapses[1,12,53].

We estimate that FlyWire's brain reconstruction took around 33 person-years of manual proofreading. The reconstruction remains open for proofreading and annotations, and new versions of the resource will be released in future (the analysis presented here is from version 783). This enables correction of remaining errors as they are discovered and further rounds of validation to be performed.

## Intrinsic neurons of the brain

A brain is defined as a structure of the nervous system that is co-localized with the sense organs in the head of an animal. Often left implicit in the definition is the idea of centralization—that most central nervous system (CNS) neurons are located in the brain. The idea involves a subtlety arising from the fact that neurons are spatially extended objects. If all of the synapses of a neuron are wholly contained in the brain, we say that the neuron is intrinsic to the brain. This contrasts with a neuron that straddles the brain and other CNS regions. The fraction of intrinsic neurons can be interpreted as the degree to which the CNS is centralized in the brain.

Of the 139,255 proofread neurons in FlyWire (Supplementary Video 1), 118,501 are intrinsic to the brain (Fig. 2a–c), which is defined as the central brain and optic lobes (Fig. 1a). Intrinsic neurons of the brain make up three-quarters of the adult fly nervous system[54–56] and amount to 85% of brain neurons. Their predominance means that the brain communicates primarily with itself, and only secondarily with the outside world (Fig. 2b).

For comparison, intrinsic neurons of the larval fly brain make up one-quarter to one-third of its nervous system[33]. Intrinsic neurons of the *C. elegans* brain make up 8–15% of its nervous system (Methods).

## Afferent and efferent neurons

Brain neurons that are not intrinsic can be divided into two categories, depending on the locations of their cell bodies. For afferent (sensory and ascending) neurons, the cell body is outside the brain, whereas for efferent (descending, motor and endocrine) neurons, the cell body is contained in the brain. It is generally accurate to think of an afferent neuron as a brain input, and an efferent neuron as a brain output. The relation to information flow is actually more subtle, however, as most fly neurites carry a mixture of presynapses and postsynapses on both dendrites and axons[10,33,53].

Our companion paper[12] exhaustively identifies all afferent and efferent neurons contained in cross sections of nerves and the neck connective running between the brain and ventral nerve cord (VNC) (Fig. 2d). Almost 95% of these neurons were in the neck connective, antennal nerve and maxillary–labial nerve. Although afferents are truncated in our reconstruction, Schlegel et al.[12] and other community members[35,57] were able to determine the sensory organs corresponding to the 5,375 non-visual sensory neurons (Fig. 2e,f) on the basis of morphology and nerve assignments. Non-visual sensory neurons enter

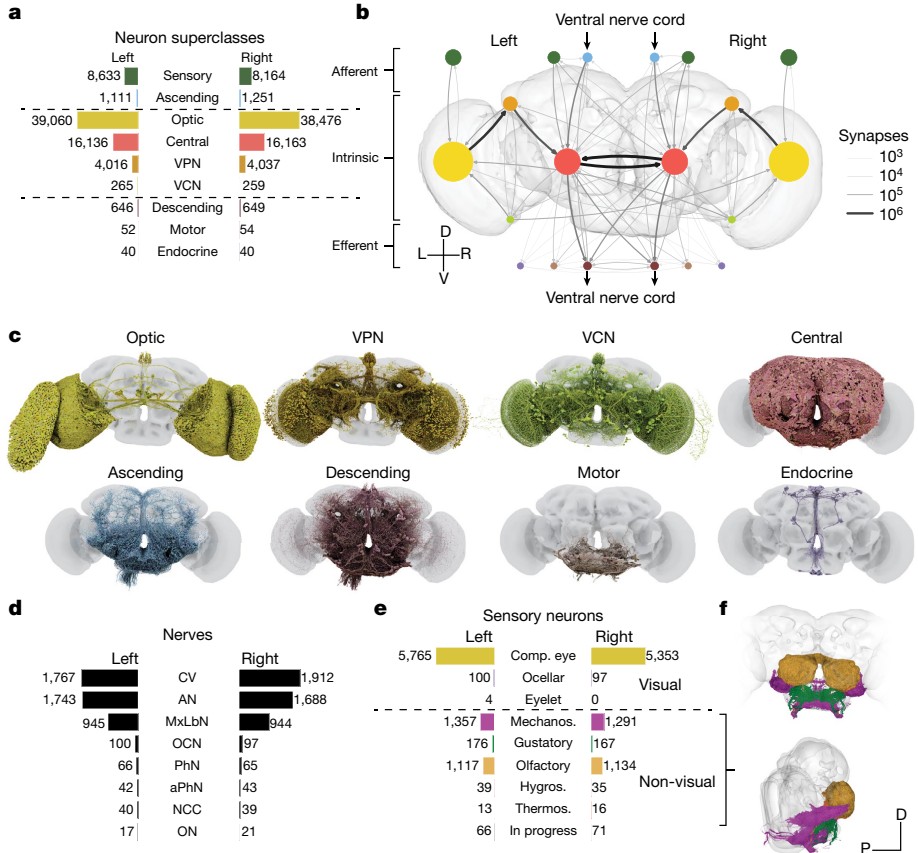

**Fig. 2 | Neuron categories. a**, We grouped neurons in the fly brain by 'flow': intrinsic, afferent or efferent. Each flow class is further divided into 'superclasses' on the basis of location and function. Neuron annotations are described in more detail in our companion paper[12]. 201 neurons were not assigned to a hemisphere and are thus omitted from this panel. **b**, Using these neuron annotations, we created an aggregated synapse graph between the superclasses in the fly brain. D, dorsal; L, left; R, right; V, ventral. **c**, Renderings of all neurons in each superclass. **d**, There are eight nerves into each hemisphere in addition to the ocellar nerve and the cervical connective nerve. All neurons traversing the nerves have been reconstructed and accounted for. AN, antennal nerve; aPhN, accessory pharyngeal nerve; CV, cervical (neck) connective nerve; MxLbN, maxillary–labial nerve; NCC, nervii corpora cardiaca; OCN, ocellar nerve; ON, occipital nerve; PhN, pharyngeal nerve. **e**, Sensory neurons can be subdivided by the sensory modality that they respond to. Almost all sensory neurons have been typed by modality. The counts for the medial ocelli were omitted and are shown in Fig. 7b. Comp. eye, compound eye; hygros., hygrosensory; mechanos., mechanosensory; thermos., thermosensory. **f**, Renderings of all non-visual sensory neurons. Scale bar, 100 μm.

the brain through nerves (Fig. 2d) that mostly terminate in the antennal lobe or the SEZ (we define the SEZ as containing the following neuropils: saddle (SAD), gnathal ganglia (GNG), antennal mechanosensory and motor centre (AMMC) and prow (PRW) (neutropils definitions are provided in Extended Data Fig. 1))[58]. The antennal lobe is the first relay centre for processing of olfactory information, and many of the olfactory receptor neuron (ORN) inputs to the antennal lobe were also reconstructed in the hemibrain. The SEZ receives more diverse inputs, including the projections of both mechanoreceptor and gustatory receptor neurons—these projections were not contained in the hemibrain. The nerves contained few efferent neurons, among which were head motor neurons ($n = 106$) or endocrine neurons ($n = 80$) (Fig. 2a–c). Many efferent neurons have branches in the SEZ, including most of the 106 motor neurons.

Visual afferents are by far the most numerous type of sensory input, and enter the brain directly rather than through nerves. There are photoreceptor axons projecting from the compound eyes ($n = 11,118$), ocelli ($n = 273$) and eyelets ($n = 8$, of which 4 have been identified).

The neurons traversing the neck connective were grouped into 1,303 efferent (descending) and 2,362 afferent (ascending) neurons (Fig. 2a–c). Cell-type annotations for many of these neurons are available[36], facilitating a matching of reconstructions from two separate electron microscopy datasets of a VNC[54–56,59] and enabling circuits spanning the whole CNS (brain and VNC) to be at least schematically mapped.

## Optic lobes and central brain

Of the 118,501 intrinsic neurons, 32,388 are fully contained in the central brain and 77,536 are fully contained in the optic lobes and ocellar ganglia (this number excludes the photoreceptors, which are sensory afferent neurons). The domination of the count by visual areas reflects the nature of *Drosophila* as a highly visual animal.

The optic lobes and ocellar ganglia also contain 8,053 neurons—the visual projection neurons[12] (VPNs)—that project into the central brain. We provide a more detailed analysis of connections in the ocellar ganglion in Fig. 7. Many VPNs are columnar types that tile the visual field. VPNs target specific neuropils (for example, anterior optic tubercle (AOTU), posterior lateral protocerebrum (PLP) and posterior ventrolateral protocerebrum (PVLP)) or optic glomeruli[60,61] in the central brain. The influence of VPNs can be very strong; 892 central neurons receive more than half their synapses from VPNs.

The hemibrain already characterized several VPN types along with their outputs in the central brain[1]. Our whole-brain reconstruction reveals many other aspects of VPN connectivity, such as their inputs in the medulla, lobula and lobula plate[62]. In addition to feedforward

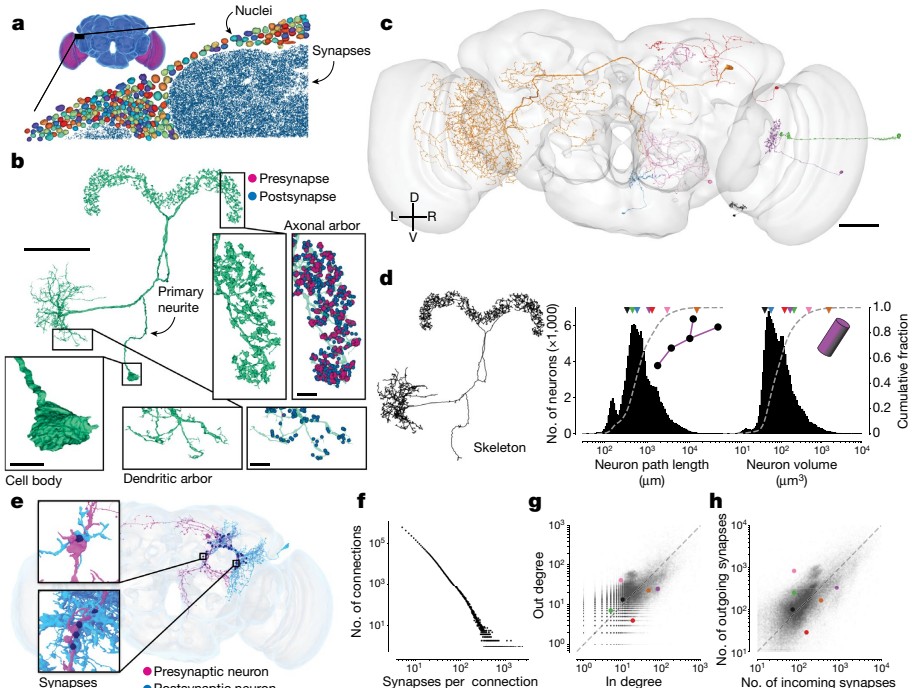

**Fig. 3 | Neuron and connection sizes. a**, The synapse-rich (synapses shown in blue) neuropil is surrounded by a layer of nuclei (random colours) located at the outside of the brain as well as between the optic lobes (purple) and the central brain (blue). **b**, An LPsP (lateral accessory lobe–posterior slope–protocerebral bridge)[1] neuron can be divided into morphologically distinct regions. Synapses (purple and blue) are found on the neuronal twigs and only rarely on the backbone. **c**, We selected seven diverse neurons as a reference for **d**–**h**. **d**, The morphology of a neuron can be reduced to a skeleton (left) from which the path length can be measured. The histograms show the distribution of path length (middle) and volume (right; the sum of all internal voxels) for all neurons. The triangles on top of the distributions indicate the measurements of the neurons in **c**.

**e**, Connections in the fly brain are usually multisynaptic, as in this example of neurons connecting with 71 synapses. **f**, The number of connections with a given number of synapses. **g**, In degree and out degree of intrinsic neurons in the fly brain are linearly correlated ($R = 0.76$). The dashed line is the unity line. Coloured dots indicate measurements of the neurons in **c**. **h**, The number of synapses per neuron varies between neurons by more than one order of magnitude and the number of incoming and outgoing synapses is linearly correlated ($R = 0.81$). Only intrinsic neurons were included in this plot. The dashed line is the unity line. Coloured dots indicate measurements of the neurons in **c**. Scale bars: 50 μm, **b** (main image) and **c**; 10 μm, **b** (expanded views).

targeting of central neurons, VPNs make 20% of their synapses onto other VPNs and 21% onto optic lobe neurons. Ganguly et al.[38] and Garner et al.[37] further investigated the visual projections to the central complex and the mushroom body.

There are 524 neurons that project from the central brain to the optic lobes. We call these visual centrifugal neurons[61] (VCNs). They are distinct from previously defined types of VCNs that are fully contained in the optic lobe and their functions are mostly unknown. VCNs are 15 times less numerous than VPNs. Nevertheless, half of all optic lobe neurons receive five or more synapses from VCNs, showing that much early visual processing incorporates feedback from the central brain. Centrifugal inputs to the retina are found in many vertebrate species, including humans[63].

Many VCNs arborize broadly in the optic lobe, appearing to cover the entire visual field. Some VCNs, however, cover only a subset of columns within a portion of the visual field. A few optic lobe neurons receive as many as 50% of their synapses from VCNs. These belong to the class of peptidergic neurons involved in circadian rhythms[40]. Tm5c is a columnar type (necessary for the preference of *Drosophila* for UV over visible light[64]), with more than 10% of its inputs coming from VCNs.

## Neuron superclasses

The neuron classes introduced above are organized into a hierarchy, as explained in our companion paper[12]. The three 'flow' classes (afferent, intrinsic and efferent) are divided into the nine superclasses (Fig. 2a). A simplified representation of the connectome as a graph in which nodes are superclasses is shown in Fig. 2b. Node sizes reflect neuron number

and link widths indicate connection number. This is the first of several simplified representations of the connectome that we introduce here.

## Neurons and glia

A basic property of the fly brain is that cell bodies are spatially segregated from neurites. Cell bodies reside near the surface ('rind') of the brain (Fig. 3a), surrounding a synapse-rich interior that comprises mainly of entangled neurons and glia, fibre bundles or tracts, and tubules of the tracheal system (Fig. 1f and Supplementary Fig. 3a).

A typical non-sensory *Drosophila* neuron is unipolar and consists of a primary neurite (also known as cell body fibre) that leaves the cell body (soma), enters the neuropil, and branches into secondary and higher-order neurites (Fig. 3b). Secondary neurites can sometimes be classified as axons if presynapses clearly dominate, or as dendrites if postsynapses clearly dominate[10,33,53]. Such an axon–dendrite distinction was made, for example, when defining VPNs and VCNs above.

However, in general, a mixture of presynapses and postsynapses is found on all non-primary neurites[10,33,53,65] (Fig. 3b). In addition, the soma of insect neurons is separated from the main processes (Fig. 3b). Given this structure, the concept that signals pass from dendrites to soma to axon, which is often a good approximation for mammalian neurons, does not apply for non-sensory neurons in the fly.

Neurons vary greatly in size and shape (Fig. 3c). We computed skeletons for all reconstructed neurons (Fig. 3d) to measure neuronal path lengths. The median path length of an intrinsic neuronal arbor was 685 μm (Fig. 3d). It has been argued that branched arbors are optimal for achieving a high degree of connectivity with other neurons[66].

Neurons with short path lengths are interesting exceptions, and can be found in both the optic lobes and central brain. Path length and volume of intrinsic neurons both varied over two orders of magnitude (Fig. 3d; path length percentiles: 0.1%, 0.138 mm; 99.9%, 19.15 mm; volume percentiles: 0.1%, 16 μm³; 99.9%, 3,001 μm³). The whole brain contains approximately 122 million attached presynapses with a total neuronal path length of around 149 m, an average of 0.82 presynapses per micrometre of path length.

Sizes vary significantly between different cell superclasses (Extended Data Figs. 3a–f and 4). Optic lobe neurons are on average much shorter than central brain neurons (0.69 mm versus 2.13 mm on average) and take up a smaller volume (0.0069 mm³ versus 0.0086 mm³ total neuronal volume), which is why the optic lobes dominate the brain by neuron number but not by volume or synapse count. Visual centrifugal neurons are among the largest in the brain, and are larger on average than VPNs (4.92 mm versus 1.56 mm path length on average). We measured much shorter path lengths and volumes for afferent neurons because only part of their axonal arbors is contained within the brain (Extended Data Fig. 3b,e), whereas arbors of efferent, motor and descending neurons which also have some of their arbor outside the brain, were among the largest we measured (Extended Data Fig. 3c,f).

A small fraction of brain volume is made up of glial cells, which are categorized into six types[67]. We estimated that 13% of the cell bodies in the electron microscopy dataset are non-neuronal or glial. Only a few astrocyte-like glia have been proofread (Supplementary Fig. 3b). Sheet-like fragments of ensheathing glia are readily found near fibre bundles in the automated reconstruction. Further proofreading of glia could be prioritized in the future if there is community demand.

## Synapses and connections

Our connectome includes only chemical synapses; the identification of electrical synapses awaits a future electron microscopy dataset with higher resolution (Discussion). Therefore, we use the term 'synapse' to mean chemical synapse. A *Drosophila* synapse is generally polyadic, meaning that a single presynapse communicates with multiple target postsynapses (Fig. 1e). In FlyWire, a polyadic synapse is represented as multiple synapses, each of which is a pair of presynaptic and postsynaptic locations[7]. Polyadic synapses are common in other invertebrate species, such as *C. elegans*, and exist in some mammalian brain structures (for example, retina).

We define a connection from neuron A to neuron B as the set of synapses from A to B. A connection typically contains multiple synapses, and the number of synapses can be large (Fig. 3e,f). Connections with fewer than 10 synapses are typical, but a single connection can comprise more than 100 synapses ($n = 15,837$) or even more than 1,000 synapses ($n = 27$). The strongest connection that we identified was from a VCN (LT39) onto a wide-field lobula neuron (mALC2), and contained more than 2,400 synapses.

Setting a threshold of at least five synapses for determining a strong connection is likely to be adequate for avoiding false positives in the dataset while not missing connections (Methods). We observed 2,700,513 such connections between 134,181 identified neurons. There are several reasons to focus on strong connections. First, a connection with many synapses is expected to be strong in a physiological sense, other things being equal[68]. Second, strong connections are more reproducible across individuals[12]. Third, higher accuracy (both precision and recall) of automatic detection is expected for strong connections, assuming that errors are statistically independent[1,53].

One of the most basic properties of a node in any network is its degree, the number of nodes to which it is linked to. To characterize the degree distribution in the *Drosophila* connectome, we focused on intrinsic neurons because unlike afferent and efferent neurons, they do not suffer from undercounting of connections owing to truncation.

For any neuron, in degree is defined as its number of presynaptic partners (input neurons), and out degree is defined as its number of postsynaptic partners (output neurons). The median in degree and out degree of intrinsic neurons are 11 and 13 (Fig. 3g), respectively, with the restriction mentioned above to connections involving five or more synapses. These median values do not appear to be substantially different from the median in degree and out degree of 10 and 19, respectively, for neurons in the *C. elegans* hermaphrodite, considering that it contains several hundred times fewer neurons than *Drosophila*.

The neuron in the *Drosophila* brain with maximum degree is a visual GABAergic (γ-aminobutyric acid-producing) interneuron (CT1), with 6,399 postsynaptic partners and 5,080 presynaptic partners (CT1 in the left hemisphere). Most neuropils of the *Drosophila* brain contain one or a few large GABAergic neurons private to that neuropil, with high in degree and out degree (see Lin et al.[49] for further analysis on connectivity motifs); these neurons are considered to be important for local feedback gain control[69]. The *Drosophila* brain contains neurons with much higher degree than—for example—the *C. elegans* hermaphrodite[32] for which the neuron with maximum degree is a command interneuron (AVAL) with 110 postsynaptic partners and 64 presynaptic partners.

The number of synapses established by a neuron is correlated with its total neurite path length ($R = 0.80$ (presynapse), $R = 0.89$ (postsynapse); Extended Data Fig. 3g). Presynapse and postsynapse counts are similarly correlated per neuron ($R = 0.81$; Fig. 3h). We tested whether large neurons tend to use their many synapses to create stronger connections with individual neurons versus more connections with many different neurons. The total number of synapses established by a neuron was much better correlated with its in and out degrees ($R = 0.93$ and $R = 0.94$, respectively) than its average connection strength ($R = 0.25$ and $R = 0.3$, respectively; Extended Data Fig. 3h,i). This indicates that on average, neurons scale their number of target neurons much more than the strength of an individual connection. It remains to be tested whether the additional target neurons are from the same type or from different cell types.

Connections and neurons are not necessarily the functional units of neural computation. For certain large fly neurons, the arbors are composed of multiple compartments that function somewhat independently[70]. These subcellular compartments, rather than whole cells, should perhaps be regarded as nodes of the connectome. In this case, CT1 would be replaced by many nodes with lower degrees, and the connection from LT39 to mALC2 would be replaced by many connections with fewer synapses between compartments of these neurons. A connectome of neuronal compartments can in principle be studied using our resource, which includes the location of every synapse.

## Neurotransmitter identity

A statistical prediction of the small molecule neurotransmitter (GABA (γ-aminobutyric acid), glutamate, acetylcholine, serotonin, dopamine and octopamine) secreted by each neuron is available. A number of validations suggest that the predictions are highly accurate in aggregate[10], but for any given synapse the prediction could be wrong. We assume that every neuron secretes a single small molecule neurotransmitter and combine the predictions for all outgoing synapses to an estimate that we assign to all outgoing synapses of a neuron—that is, we provisionally assume that neurons obey Dale's law, although it is known that co-transmission does occur in the fly brain[71].

GABAergic neurons had higher degrees on average than glutamatergic and cholinergic neurons (median in- and out degrees of intrinsic neurons: GABA, 14 incoming and 16 outgoing partners; glutamate, 11 incoming and 13 outgoing partners; acetylcholine, 10 incoming and 13 outgoing partners; Extended Data Fig. 3j). Across all neuron categories, we found that GABAergic neurons were on average longer than glutamatergic and cholinergic neurons (median length of intrinsic

neurons: GABA, 0.88 mm; glutamate, 0.85 mm; acetylcholine, 0.63 mm; Extended Data Fig. 3k).

As a rule, we assume that cholinergic neurons are excitatory and GABAergic and glutamatergic neurons are inhibitory[72,73]. Lin et al.[49] identified all GABAergic and glutamatergic neurons that are bidirectionally coupled with large numbers of cholinergic neurons. This reciprocal inhibitory–excitatory motif is widespread throughout the fly brain.

## From connectome to projectome

For mammals, tracer injection studies have mapped the axonal projections between brain regions of mouse[74,75] and macaque[76]. In the fly, large numbers of light microscopy reconstructions of single neurons have been aggregated to map projections between brain regions[77,78]. Such maps have been called projectomes[79] or mesoscale connectomes[14]. In such techniques, the sampling of axons is difficult to control, and therefore accurate quantification of projection strength is challenging.

Here we computed a projectome from a synapse-level connectome (Fig. 4a and Extended Data Fig. 5). The interior of the fly brain has been subdivided into hierarchical neuropil regions[80] (Fig. 1 and Extended Data Fig. 1). Our fly projectome is defined as a map of projections between these neuropil regions. Because cell bodies are spatially separated from neuropils, a fly neuron cannot typically be assigned to a single brain region. This is unlike the situation for a mammalian neuron, which is conventionally assigned to the region containing its cell body. A typical fly neuron belongs to multiple neuropils.

The projectome is a neuropil–neuropil matrix, and is computed as follows. Each intrinsic neuron contributes to the projections between neuropils where it has presynaptic and postsynaptic sites. We weighted neuron projections by the product of the respective number of synapses and normalized the result for every neuron such that the neuropil–neuropil matrix sums to the total number of intrinsic neurons. Each column corresponds to all the neurons projecting to a neuropil and each row corresponds to to all neurons projecting out of it (Fig. 4b). Each square then represents the summed fractional weight of all neurons projecting between two neuropils (Fig. 4c,d). We added afferent and efferent neurons to the matrix by calculating the sum of the weighted neuron projections per superclass to and from all neuropils, respectively.

Whereas each neuropil is connected to many others, most neurons have synaptic sites in only a few neuropils (Fig. 4e). The largest weights in the projectome tend to be internal to individual neuropils, such as within the medulla or within the fan-shaped body[49]. The largest inter-neuropil projections overall are lobula to medulla, whereas within the central brain the largest inter-neuropil projections are mushroom body, medial lobe to mushroom body, calyx.

We repeated this process to construct projectomes for each fast neurotransmitter type (Extended Data Fig. 5). Some neuropil–neuropil connections exist strongly for one neurotransmitter but not others. For example, the neuropils making up the central complex (fan-shaped body, ellipsoid body, protocerebral bridge and noduli) and the mushroom body (calyx, pendunculus, vertical lobe and medial lobe) are largely linked by excitatory connections.

We observed a strong symmetry between projections in the left and right hemisphere as well as with the central neuropils located on the midline (Extended Data Fig. 6a,b); this highlights the strong similarity between the two sides of the brain. We observed that contralateral projections (projections from one side of the brain to the other) were generally weaker than projections to the same or ipsilateral neuropil (Extended Data Fig. 6c). The strongest contralateral projections are between left and right superior protocerebrum, followed by left and right anterior ventrolateral protocerebrum. Of note, projection weights were not strongly correlated to inter-neuropil distance. Although the strongest projections are often between nearby neuropils, there are also many nearby neuropils that do not share strong connections (Extended Data Fig. 6d).

The SEZ (Fig. 4f) is the ventral portion of the central brain, and has been shown to contribute to a variety of behaviours[58]. It is almost entirely unrepresented in the hemibrain reconstruction[1], and is only partially reconstructed in the larval brain[33]. The five neuropils in the SEZ (left and right AMMC, GNG, SAD and PRW; Fig. 4f; breakdown by neuropil in Supplementary Figs. 4 and 5) amount to 17.8% of central brain neuropil volume (0.0018 mm³ out of 0.0103 mm³); they contain afferents mostly from non-visual sensory neurons (mechanosensory and taste) and ascending neurons, as well as a large number of efferents (motor, endocrine and descending neurons; descending neurons receive on average 52% of their inputs in one of the five SEZ neuropils). The SEZ is thus important for information flow to and from the brain. Judging from the projectome (Fig. 4a), the SEZ neuropils interact with almost all parts of the brain. Notable exceptions are the central complex (ellipsoid body, fan-shaped body, protocerebral bridge and noduli) and the mushroom body, suggesting less crosstalk between those circuits and neurons in the SEZ (explored in more detail in Fig. 6; see also Pospisil et al.[39]).

## Hemispheric organization

Our reconstruction includes both left and right brain hemispheres. This is important for tracing sensorimotor pathways that cross from one side to the other, and more generally for understanding interactions between the two hemispheres. The projectome (Fig. 4a) reveals that most projections are ipsilateral or between neuropils on the same side of the brain.

The low fraction of non-ipsilateral neurons is primarily due to their scarceness in the optic lobes. Only 139 neurons (0.2%) in the optic lobes cross hemispheres and cross the central brain without making synapses there (Supplementary Fig. 6)—these neurons are considered to be 'fully contained' in the optic lobes because our definition depends only on synapse locations. These neurons mediate direct interactions between the two optic lobes, and their rarity suggests that these interactions represent a smaller fraction of the computations that occur within the optic lobes. Integration of information from both eyes may rely more on the abundant crossing connections between the central brain targets (AOTU, PLP and PVLP) of VPNs.

A higher proportion (40%) of central brain neurons are non-ipsilateral, largely owing to central neuropils, similar to those of the central complex and SEZ. To classify non-ipsilateral neurons, we began by examining the spatial distributions of their postsynapses (inputs). We divided the neuropils into three categories. Left and right categories included the neuropils that come in mirror-symmetric pairs. Centre included the seven remaining neuropils that are located on the midline. For each neuron, we computed the proportions of its postsynapses in left, right and centre neuropils (Extended Data Fig. 7). Each neuron was assigned to the dominant category, and near-ties were rare. The exceptions are symmetric neurons with cell bodies at the midline of the brain (Supplementary Fig. 7, $n = 89$).

Next, we explored how many neurons of left and right categories have presynapses (outputs) in the other hemisphere. Similar to the analysis of the 1st instar larval connectome[33], we found that neurons projecting to the other hemisphere can be grouped into bilateral neurons, those with outputs in both hemispheres, and contralateral neurons, which almost exclusively had presynapses in the other hemisphere (Fig. 4g–i). Notably, a much larger fraction of VCNs projected to the contralateral hemisphere than VPNs, and both VCNs and neurons of the central brain contain a large fraction of bilateral neurons (Fig. 4h). As stated above, this analysis again revealed the dominance of ipsilateral connections in the brain. Whereas mixing between the hemispheres is more rare, mixing between sensory modalities within a hemisphere is common (Fig. 6).

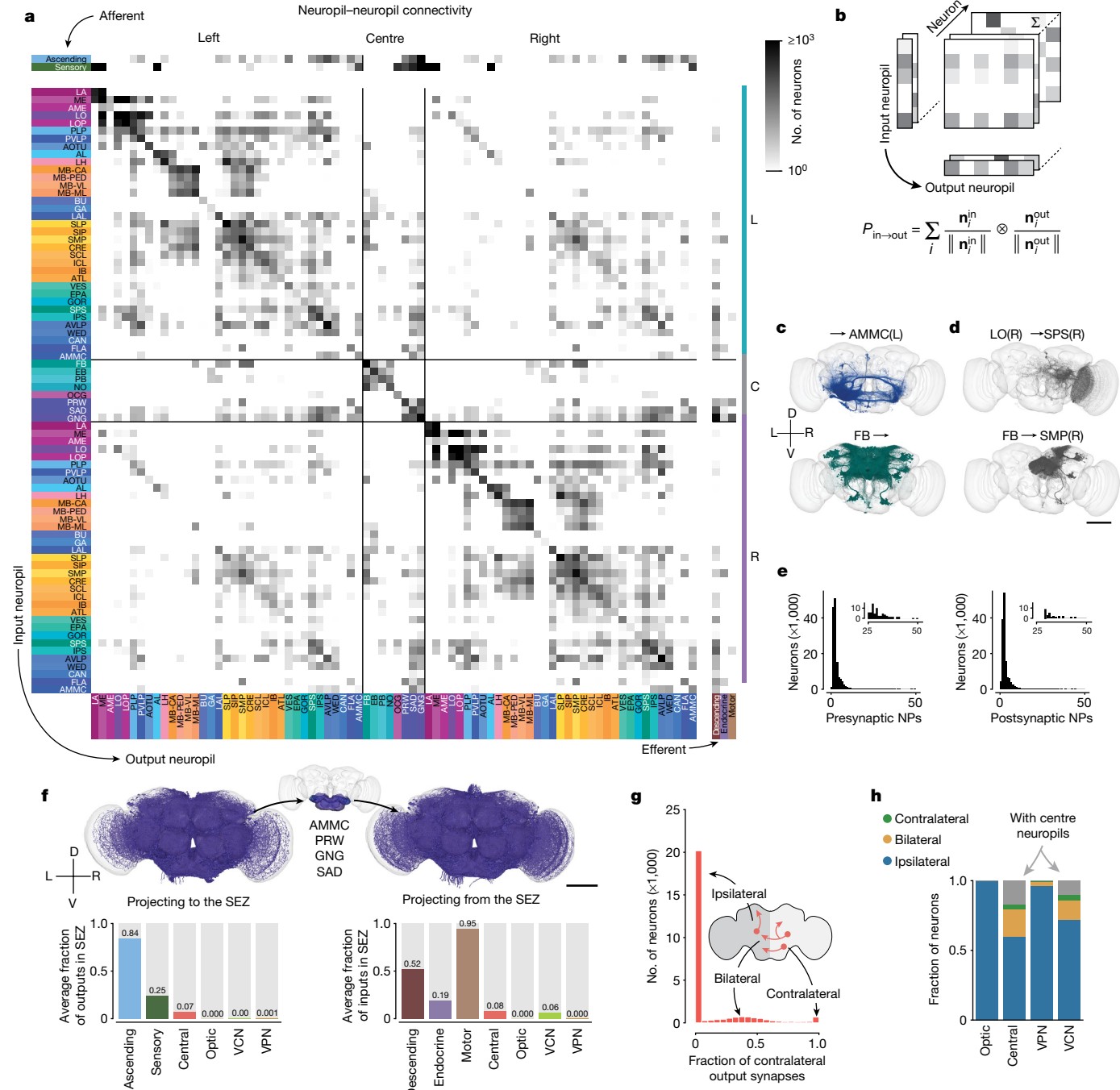

$$P_{\text{in}\rightarrow\text{out}} = \sum_{i} \frac{\mathbf{n}_i^{\text{in}}}{\|\mathbf{n}_i^{\text{in}}\|} \otimes \frac{\mathbf{n}_i^{\text{out}}}{\|\mathbf{n}_i^{\text{out}}\|}$$

**Fig. 4 | Neuropil projections and analysis of crossing neurons. a**, Whole-brain neuropil–neuropil connectivity matrix. The main matrix was generated from intrinsic neurons, and afferent and efferent neuron classes are shown on the side. Incoming synapses onto afferent neurons and outgoing synapses from efferent neurons were not considered for this matrix. See Extended Data Fig. 5 for neurotransmitter-specific matrices. Neuropils are defined in Extended Data Fig. 1. C, centre neuropils; L, left neuropils; R, right neuropils. **b**, Cartoon describing the generation of the matrix in **a**. The connectivity of each neuron is mapped onto synaptic projections between different neuropils. $\mathbf{n}_i^{\text{out}}$ and $\mathbf{n}_i^{\text{in}}$ are vectors of numbers of synapses for each neuropil and neuron. **c**,**d**, Examples from the matrix in **a** with each render corresponding to one row or column in

the matrix (**c**) and examples from the matrix with each render corresponding to one square in the matrix (**d**). **e**, Most neurons have pre- and postsynaptic locations in fewer than four neuropils. Insets show a closer view of the long tail of the distribution. NPs, neuropils. **f**, Renderings (subset of 3,000 each) and input and output fractions of neurons projecting to and from the SEZ. The SEZ is composed roughly of five neuropils (the AMMC has left and right homologues). Average input and output fractions were computed by summing the row and column values of the SEZ neuropils in the superclass-specific projection matrices. **g**, Fraction of contralateral synapses for each central brain neuron. **h**, Fraction of ipsilateral, bilateral and contralateral neurons projecting to and from the centre neuropils per superclass. Scale bars, 100 µm.

Many types of fly neurons are known to exhibit striking stereotypy across individuals, as well as across both hemispheres of the same individual. Schlegel et al.[12] show quantitatively using FlyWire brain and hemibrain data that these two types of stereotypy are similar in degree.

## Optic lobes, columns and beyond

So far we have mentioned neurons that connect the optic lobes with each other, or with the central brain. The intricate circuitry within each

optic lobe is also included in the FlyWire brain connectome. Matsliah et al.[11] analysed and typed all neurons intrinsic to the right optic lobe. Photoreceptor axons terminate in the lamina and medulla, neuropils of the optic lobes (Fig. 5a,b). Each eye contains approximately 800 ommatidia that map to columns in the lamina that are arranged in a hexagonal lattice (Fig. 5c). This structure repeats in subsequent neuropils from lamina to medulla to lobula to lobula plate. The neuropils have been finely subdivided into layers that are perpendicular to the columns[81]. The 2D visual field is mapped onto each layer and any given cell type tends to synapse in some subset of the layers. Cell types vary greatly in size, with uni-columnar cell types being the smallest; (for example, Mi4); at the other extreme are large cells that span almost all columns (for example, Dm17); in between the extremes are many multi-columnar cell types (for example, Dm12) (Fig. 5c).

Mi4 is a true 'tiling' type—that is, its arbors cover the visual field with little or no overlap, and have similar size and shape (Fig. 5c). Dm12 arbors overlap with each other, but the spatial arrangement is still regular. These and other distal medullary cell types were previously characterized by multicolour light microscopy[82]. Our electron microscopy reconstructions reveal even more detailed information about the spatial patterning of these types (for example, co-fasciculation of neurites of neighbouring Dm12 cells; Fig. 5c). More importantly, the FlyWire reconstruction encompasses all multi-columnar cell types, including those outside the medulla. Judging from the many examples we have studied throughout the optic lobe, it seems that regular coverage of the visual field without gaps is a defining criterion for most cell types, similar to mammalian retina[83]. There are, however, exceptional cell types that cover the visual field in an irregular manner. For example, there are exactly two LPi14 cells per optic lobe[84]. The shapes of each pair are complementary, as if they were created by cutting the visual field into two pieces with a jigsaw (Fig. 5d); this tiling was not evident when reconstructing only a portion of an optic lobe[84].

Much of the existing research on wide-field visual motion processing has relied on the simplifying idea that the computations are mostly in columnar circuits, and the columnar outputs are finally integrated by large tangential cells in the lobula plate. This research has been aided by wiring diagrams containing connections between cells in the same column or neighbouring columns[2]. In previous studies, an absence of information across columns has necessitated treating each column as identical in simulations of the optic lobe[85]. The FlyWire brain connectome contains not only the columnar neurons, but also all neurons that extend across columns (Fig. 5c). These neurons are both excitatory and inhibitory, and can support interactions between even distant columns. This opens the possibility of a much richer understanding of optic lobe computations and is further explored by Christenson et al.[41] in investigating hue selectivity.

## Analysis of information flow

Although afferent and efferent neurons make up a numerically small proportion of the brain (estimated 13.9% and 1.1%, respectively), they are important because they connect the brain to the outside world. Examining connections of these neurons is useful when attempting to predict the functions of intrinsic neurons from the connectome. For example, one might try to identify the shortest path in the connectome from an afferent (input) neuron that leads to a given intrinsic neuron. The sensory modality of the afferent neuron could provide a clue regarding the function of the intrinsic neuron. This approach, although intuitive, ignores connection strengths and multiplicities of parallel pathways. We therefore use a probabilistic model to estimate information flow in the connectome[26], starting from a set of seed neurons (Fig. 6a and Methods).

The likelihood of a neuron being traversed increases with the fraction of inputs from already traversed neurons up to an input fraction of 30%, after which traversal is guaranteed (Fig. 6a). We ran the traversal

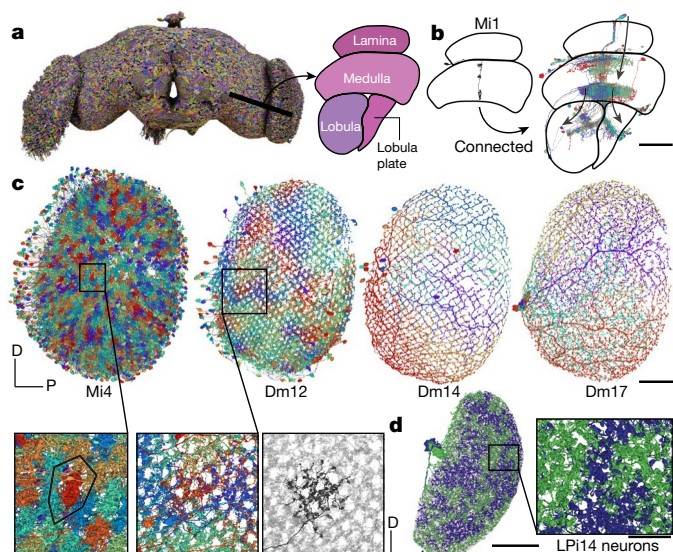

**Fig. 5 | Optic lobes. a**, Rendering of a subset of the neurons in the fly brain. A cut through the optic lobe is highlighted and neuropils are annotated. **b**, A single Mi1 neuron (left) and all neurons that share a connection with the single Mi1 neuron (at least five synapses) (right). Three large neurons (CT1, OA-AL2b2 and Dm17) were excluded for the visualization. **c**, Top, Mi4, Dm12, Dm14 and Dm17 neurons in the right optic lobe, as annotated by Matsliah et al.[11]. Bottom, expanded views of the outlined regions in Mi4 and Dm12 show the local structure. For Dm12, the right image shows a single neuron in black and all other Dm12 neurons are in background. **d**, The two LPi14 neurons in the right lobula plate (neuropil shown in background). Scale bars: 50 μm, **b** and **c,d** (main image); 10 μm, **c,d** (expanded views).

model for every subset of afferent neurons as seeds ($n = 12$ input modalities to the central brain (Fig. 2e and Supplementary Fig. 8; full list in Methods)). We then measured the flow distance from these starting neurons to all intrinsic and efferent neurons of the central brain. For instance, the neurons reached early from gustatory neurons (Fig. 6b) match second-order projection neurons identified by Snell et al.[86] using *trans*-Tango.

To visualize information flow for neurons with inputs in the central brain in a common space, we treated the traversal distances starting from each seed population as a neuron embedding and built a uniform manifold approximation and projection (UMAP) from all of these embeddings (Fig. 6c). Within the map, we found that neurons of the same cell class (for example, two groups of Kenyon cells, all mushroom body output neurons, all antennal lobe local neurons and all central complex neurons) cluster, indicating that cell types can in part be defined by their proximity to different input neurons. Next, we displayed traversal order on top of the UMAP plot to compare traversal orders starting from different modalities (Fig. 6c,d). We find that almost every neuron in the central brain can be reached by starting from any modality—this 'small world' property of the network is covered in more detail by Lin et al.[49] Comparing orders revealed that almost all neurons in the central brain are reached early starting from some modality, with the exception of neurons in the central complex (Fig. 6c,d and Extended Data Fig. 9), highlighting that the central complex is dominated by internal computations[24]. Kenyon cells were contained in two clusters—one of which is targeted very early from olfactory receptor neurons and the other is targeted early by VPNs[87].

We then ranked all neurons by their traversal distance from each set of starting neurons and normalized the order to percentiles. For instance, a neuron at the 20th percentile had a lower rank than 80% of neurons. This enabled us to determine how early information from each afferent modality reached various targets, including the descending neurons,

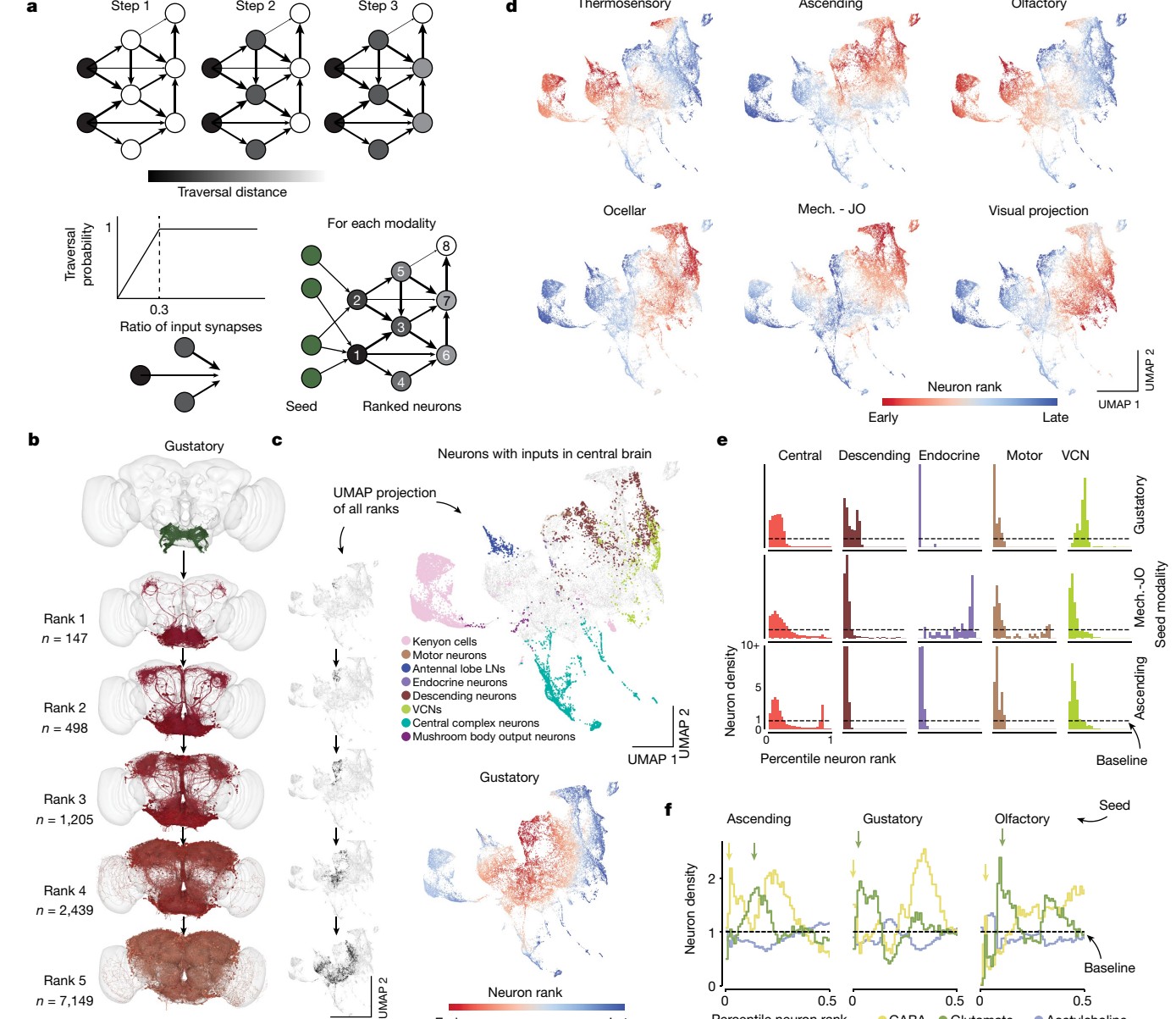

**Fig. 6 | Information flow through the *Drosophila* central brain. a**, We applied an information flow model for connectomes[26] to the connectome of the central brain neurons. Neurons are traversed probabilistically according to the ratio of incoming synapses from neurons that are in the traversed set. The information flow calculations were seeded with the afferent classes of neurons (including the sensory categories). **b**, We rounded the traversal distances to assign neurons to layers. For gustatory neurons, we show a subset of the neurons (up to 1,000) that are reached in each layer. Neurons are coloured according to the traversal distance in **c**. **c**, UMAP analysis of the matrix of traversal distances, resulting in a 2D representation of each neuron in the central brain. Neurons from the same class co-locate (see also Extended Data Fig. 9). The small UMAP plots aligned with layers in **b** show where the neurons for each rank from the gustatory

neurons fall within the distribution (black dots). Bottom, we coloured neurons in the UMAP plot by the rank order in which they are reached from gustatory seed neurons. Red neurons are reached earlier than blue neurons. LN, local neuron. **d**, As in **c**, bottom, for multiple seed neuron sets (see Extended Data Fig. 8c for the complete set). Mech. - JO, mechanosensory−Johnston's organ. **e**, For each sensory modality, we used the traversal distances to establish a neuron ranking. Graphs show the distributions of neurons of each superclass within the specific rankings for each sensory modality (see Extended Data Fig. 8a for the complete set). **f**, Neurons were assigned to neurotransmitter types. Graphs show their distribution within the traversal rankings similar to **d**. Arrows highlight the sequence of GABA−glutamate peaks found for almost all sensory modalities (see Extended Data Fig. 8b for the complete set).

endocrine neurons, motor neurons and VCNs (Fig. 6e and Extended Data Fig. 8a). Endocrine neurons are closest to the gustatory sensory neurons, whereas motor and descending neurons were reached early for mechanosensory and visual afferents (Extended Data Fig. 8a).

We next tested whether the afferent cell classes target inhibitory neurons early or late. We found that putative inhibitory neurons (neurons predicted to express GABA and glutamate) were overrepresented in the set of early neurons (Fig. 6f). Surprisingly, we identified a sequence of

GABAergic and glutamatergic peaks in the sequence of targeted neurons that was replicated for almost all afferent modalities (Extended Data Fig. 8b).

Our information flow analysis provides a compressed representation of the connectome, but currently ignores signs of connections (neurotransmitter identity) and the biophysics of neurons and synapses, and therefore terms such as 'early' and 'late' should not be interpreted as true latencies to sensory stimulation. Shiu et al.[34] and Pospisil et al.[39]

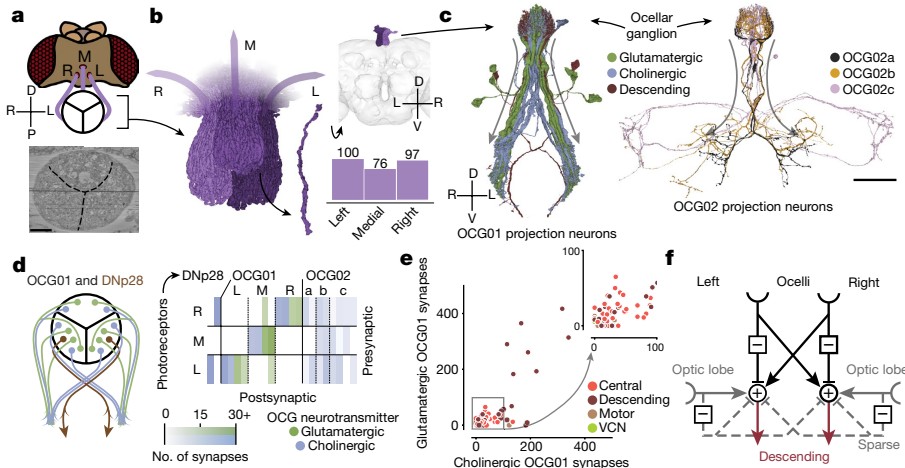

**Fig. 7 | Ocellar circuits and their integration with VPNs. a**, Overview of the three ocelli (left (L), medial (M) and right (R)) positioned on the top of the head. Photoreceptors from each ocellus project to a specific subregion of the ocellar ganglion which are separated by glia (marked with black lines on the electron micrograph (bottom)). Left and right are flipped in accordance with the orientation of the dataset (Methods). **b**, Renderings of the axons of the photoreceptors (left) and their counts (bottom right). Top right, location of the ocellar ganglion relative to the brain. **c**, Renderings of OCG01, OCG02 and DNp28 neurons with arbors. 'Information flow' from presynapses and postsynapses is indicated by arrows along the arbors. **d**, Connectivity matrix of connections between photoreceptors and ocellar projection neurons, including two descending neurons (DNp28). **e**, Comparison of number of glutamatergic and cholinergic synapses from ocellar projection neurons from the lateral eyes onto downstream neurons coloured by superclass ($R = 0.78$, $P < 10^{-26}$). **f**, Summary of the observed connectivity between ocellar projection neurons, VPNs and descending neurons. Scale bars, 100 μm.

use the connectome to model *Drosophila* brain dynamics and include connection weights (number of synapses) and putative connection signs (excitatory or inhibitory).

## Cell types and other annotations

Neurons in *Drosophila* are considered to be identifiable across hemispheres and individuals, enabling cell-type classification of all neurons in FlyWire's brain dataset. Such classification is useful for generating testable hypotheses about circuit function from the connectome. FlyWire community members, many of whom are experts in diverse regions of the fly brain, have shared 133,700 annotations of 114,209 neurons (Supplementary Fig. 9), including comprehensive cell typing in the optic lobe[11], the majority of sexually dimorphic neurons and sensory neurons[35], as well as a diversity of cell types throughout the brain, including the SEZ (Fig. 2f). Each neuron in FlyWire is also given a unique identifier on the basis of the neuropil through which it receives and sends most of its information. Curation of these annotations continues, and we invite further community efforts to identify cell types, which can be contributed through Codex (https://codex.flywire.ai).

In addition, matching between cell types identified in the hemibrain[1] and both hemispheres of FlyWire's brain dataset provides additional annotations for neurons contained in both datasets. Our companion paper[12] provides annotations for more than 8,400 unique cell types via such matching. All cell annotations can be queried in Codex. Some of these have already been mentioned, such as the 'flow' annotations of intrinsic versus afferent versus efferent, superclass annotations (Fig. 2), connectivity tags (such as rich club, broadcaster or highly reciprocal)[42], neurotransmitter predictions[10] and left–right annotations for cell body location[88], in addition to lineages or groups of neurons derived from a single neuroblast.

## Ocellar circuit, from inputs to outputs

The completeness of the FlyWire brain connectome enables tracing complete pathways from sensory inputs to motor outputs. We demonstrate this capability by examining circuits that emanate from the ocellar ganglion and leveraging cell-type information. In addition to the large compound eyes, flying insects have smaller visual sensory organs[89], including the three ocelli on the dorsal surface of the head cuticle (Fig. 7a). The ocelli project a blurry image of light-level changes in the UV and blue region of the spectrum[90,91] and are thought to be useful for flight control and orientation relative to the horizon[92]. Notably, although the role of the ocelli has been hypothesized (for example, light-level differences between the eyes when the fly is shifted off axis should quickly drive righting motions of the head, wings and body to stabilize gaze and re-orient the body), little is known about the circuitry downstream of this sensory organ that would mediate this function.

We find that photoreceptor axons ($n = 273$) from the three ocelli innervate three distinct regions of the ocellar ganglion separated by glial sheets (Fig. 7a,b). The ocellar ganglion additionally contains 63 neurons that we categorized into four broad groups (Fig. 7c and Extended Data Fig. 10a): local neurons ($n = 16$), 2 types of interneurons, divided on the basis of their arborizations and caliber (OCG01 ($n = 12$), OCG02 ($n = 8$)), descending neurons (DNp28, $n = 2$), and centrifugal or feedback neurons ($n = 25$). Ocellar local neurons are small and connect sparsely with photoreceptors from all ocelli.

Twelve OCG01 interneurons and two descending neurons (DNp28, one per lateral ocellus) represent the main pathway from the ocellar ganglion to the central brain. DNp28 projects to the intermediate, haltere, wing and neck tectula of the ventral nerve cord[55,93]. In each ocellus, half of the OCG01s were inferred to express glutamate (likely inhibitory), and the other half were inferred to express acetylcholine (likely excitatory). There are four OCG01s per ocellus (Fig. 7d). OCG01s tile the ocellar ganglion, indicating that their receptive fields tile the visual fields of the ocelli (Extended Data Fig. 10b,c). OCG02 axons are much thinner than OCG01 axons, and likely transmit signals more slowly. Two OCG02 subgroups (a and b) innervate similar neuropils to the OCG01s (inferior posterior slope (IPS) and superior posterior slope (SPS)), and OCG02c neurons target the PLP, a brain region that also receives input from VPNs from the compound eyes[60].

Neurons downstream from OCG01s in the IPS, SPS and GNG receive inhibitory input from the ipsilateral ocellus and excitatory input from the contralateral ocellus (Fig. 7d, left), and the amount of synaptic input from each ocellus is tightly correlated (Fig. 7e, $R = 0.78$,

$P < 10^{-26}$)—this balance is likely to be a key ingredient in how signals are integrated (the descending circuits are activated by a signal difference between the eyes). We found that 15 different descending neurons each receive more than 200 synapses from the OCG01 neurons. For example, 2 descending neurons in each hemisphere received more than 30% of their synaptic inputs in the brain from ocellar projection neurons: DNp20/DNOVS1 (left: 57%, right: 44%) and DNp22/DNOVS2 (left: 36%, right: 33%). DNOVS1 and other descending neurons with strong input from OCG01s generally also receive strong input from ipsilateral VPNs (neurons that connect the optic lobe to the central brain) (Extended Data Fig. 10d). For example, DNOVS1 is known to be activated by rotational optic flow fields across the compound eye, and projects to the neck motor system[94,95]. A handful of glutamatergic (putative inhibitory) VPNs also sparsely innervate descending neurons in both hemispheres. As the ocelli transmit mainly information about light levels, the dense integration with motion direction signals from the compound eyes was not previously appreciated, but should aid in precision adjustments of head and body movements for gaze stabilization and flight control[96].

There is also extensive feedback from the brain directly to the ocellar ganglion via 25 ocellar centrifugal neurons (OCC). We found striking targeting specificity of two OCC subgroups (OCC01a and OCC01b, predicted to be cholinergic) which synapse onto all OCG01 and DNp28 neurons with strong connections compared with their overall synaptic budget (Extended Data Fig. 10e). The OCC01s receive input in a wide range of neuropils, notably the SEZ, as well as IPS and SPS, the same neuropils that receive inputs from the OCG projection neurons (Extended Data Fig. 10f). The role of the OCCs in gating visual information and potentially driving the OCGs in the absence of photoreceptor activity remains to be determined.

On the basis of our analysis of connectivity, we hypothesize how the pathways from the ocelli to descending neurons function (Fig. 7f). As in a Braitenberg vehicle for phototaxis[97], excitation and inhibition are organized so that the head and body of the fly should roll around the anteroposterior axis to orient the ocelli towards light. In this compact example, the whole-brain connectome, which extends from brain inputs to outputs, uncovers new pathways and facilitates the generation of testable hypotheses for circuit mechanisms of sensorimotor behaviour.

## Discussion

### Connectome analysis

We use the term 'connectome' to mean a neuronal wiring diagram of an entire nervous system, or at least an entire brain[98]. This is in keeping with the intent of the original definition[14], which emphasized comprehensiveness. Similarly, the term 'genome' refers to the entire DNA sequence of an organism, or at least the entirety of genes. Our neuronal wiring diagram of a whole fly brain arguably crosses the threshold for being called a connectome, although it would be reasonable to insist that a connectome should include the ventral nerve cord as well as the brain. Either way, the comprehensiveness of our wiring diagram has significant benefits for brain research and enables many kinds of studies that were not previously possible using wiring diagrams of portions of the fly brain. The optic lobes and the SEZ are two prominent regions that are mostly absent from the hemibrain. Both sides of the brain are now included, which enables the tracing of pathways that cross the midline. Owing to the presence of afferent and efferent neurons, pathways can be traced from sensory inputs to intrinsic neurons and brain outputs (motor, endocrine and descending neurons). This was done in a global manner to analyse the neuropil projectome, by using the information flow model, and more specifically to uncover the structure and hypothesize a circuit mechanism for behaviours supported by the ocelli. A set of companion studies provides additional global analyses of the connectome and studies of specific families of pathways[11,12,34,36–45].

For the first time, one can now compare entire connectomes of different species, starting with *D. melanogaster* and *C. elegans*, as touched on here and explored in more depth by Lin et al.[49]. It also enables comparison of connectomes of the same species at different developmental stages[33]. Although FlyWire is currently the only adult fly connectome, it can be compared with the hemibrain reconstruction in regions where they overlap to detect wiring differences between adults of the same species and to validate and extend cell-type definitions[12].

Finally, the connectome now enables brain simulations—partial connectomes of the early visual system of the fly[3] had already inspired simulations of visual processing[85]. This effort has now been extended to leverage the full connectome[34,39] and to—for example—predict taste responses of neurons[34]. These simulations assume that that physiological connection strength is proportional to anatomical synapse count, either globally[34], or for synapses sharing the same presynaptic and postsynaptic cell types[85], and have inferred connection signs (excitatory versus inhibitory) from neurotransmitter identity as predicted from electron microscopy images[10] or from transcriptomics[99]. Ongoing discoveries regarding the biophysics of fly neurons will guide efforts to make simulations more realistic. For example, inhibition can be shunting rather than subtractive in some fly neurons[100], and the conductance of an inhibitory synapse can be ten times higher than that of an excitatory synapse[101]. Whereas the simulations mentioned above were based on point neuron models, future simulations could utilize multicompartmental neuron models constructed using the synapse locations and reconstructed neuronal morphologies provided by FlyWire, as well as emerging data about ion channels and receptors from transcriptomics and proteomics.

### Electron microscopy data acquisition and reconstruction

The hemibrain[1] was reconstructed from $8 \times 8 \times 8$ nm$^3$ images acquired by focused ion beam scanning electron microscopy[102–104] (FIB-SEM), a form of block face electron microscopy[105,106]. By contrast, FlyWire's reconstruction is based on a full adult fly brain (FAFB) dataset[9] of $4 \times 4 \times 40$ nm$^3$ images acquired by serial section transmission electron microscopy (ssTEM). Initially, the lower $z$ resolution and higher prevalence of artefacts made alignment and reconstruction of ssTEM datasets challenging. These were cited by the hemibrain effort to justify the use of FIB-SEM despite its higher cost, slower speed and complex operation requiring many 20-μm slabs to be imaged individually and then stitched together[1]. Computational advances have now closed this gap[107] and FAFB images were accurately aligned with a new approach that leverages convolutional nets[108]. The hemibrain images were automatically segmented using flood-filling convolutional nets[109], whereas FlyWire used the older, less computationally expensive approach of boundary-detecting convolutional nets[110,111]. Overall, from acquisition to reconstruction to analysis to dissemination, the technology stack used by FlyWire is distinct from that used for the hemibrain. A notable overlap is the use of neuroglancer[112] for browser-based 3D visualization.

FlyWire's whole-brain automated segmentation was proofread with an estimated 33 person-years of effort (Methods), whereas hemibrain proofreading required 50 person-years for a part of the brain[1]. Notably, the accuracy of our proofread wiring diagram is similar to that of the hemibrain (Extended Data Fig. 2 and Supplementary Fig. 3). For both FIB-SEM and ssTEM, incomplete attachment of twigs to backbones is currently the main factor that limits the accuracy of reconstructing synaptic connectivity, and in both cases synaptic connectivity is limited to chemical synapses. Higher resolution might enable the reconstruction of electrical synapses, which are included in the *C. elegans* connectome[31,32].

### Limitations of our reconstruction

We showed that the attachment rates of twigs is sufficient to facilitate detection of nearly all large connections[8] (those with more than nine synapses). Nonetheless, the observed synapse counts underrepresent

the actual number of synapses and some connections with few synapses remain undetected. Substantial improvements in twig attachment are unobtainable with further proofreading, as increasing the postsynaptic attachment rate from 44.7% to 50% would require further proofreading of more than 700,000 fragments. Therefore, increases in twig attachments will rely on improvements in image acquisition, image alignment and automated reconstruction. Although proofreading was largely carried out in a neuropil-agnostic manner, attachment rates differ between neuropils (Supplementary Fig. 2) owing to differences in the number of synapses on twigs and backbones and how challenging a neuropil was to reconstruct. Although these effects are largely symmetric, the optic lobe was affected by a one-sided artefact. The left lamina was partially severed from the medulla in the left hemisphere (Fig. 1a), reducing the reconstruction accuracy for some of the lamina neurons on one side (Supplementary Fig. 2).

The automated synapse detection currently used by FlyWire was performed by Buhmann et al.[7] in an independent effort. By combining the FlyWire brain reconstruction with these synapses, the resulting connectome inherits the limitations from both. In the case of the synapse detection, users should be aware that the ground truth was limited to a few neuropils. As a result, synapse detection performance is lower for some cell types, and we are aware that sensory neurons are particularly affected. FlyWire's reconstruction is compatible with any synapse prediction method and improved synapse prediction will be made available in the future.

## Imaging larger

Imaging a larger volume would open up other interesting opportunities. Reconstructing and proofreading a full CNS would enable the mapping of all pathways linking the brain and VNC. In the meantime, it is already possible to establish correspondences between FlyWire and FANC[36], a reconstruction of a VNC from another female fly[54,59]. The first *C. elegans* connectome was obtained similarly as a mosaic drawn from multiple worms[31]. Imaging an entire fly, both CNS and body, would enable the addition of sensory organs and muscles to the reconstruction. This also has precedent in the *C. elegans* connectome[32], which includes neuromuscular junctions, the *Platynereis dumerilii* larva[113], and the 1st instar *Drosophila* larva for which a whole-animal electron microscopy dataset was recently published[114].

Technologies developed for FlyWire have already been applied to millimetre-scale chunks of mammalian brain[4,5], which are more than 50 times larger in volume than a fly brain. The US National Institutes of Health has begun a transformative project to reconstruct a whole mouse brain from an exabyte of electron microscopy images[115] and a report from the Wellcome trust recently examined the road to a whole mouse brain connectome[116].

## Openness

The 1996 Bermuda Principles mandated daily release of Human Genome Project sequences into the public domain[117]. We believe that openness is also important for large-scale connectomics projects, particularly because these projects are expensive, require coordinated effort and take several years to complete—sharing connectomes only after proofreading and annotation are completed prevents scientific discovery that can occur while the connectome is being completed. Shortly after its inception in 2019, FlyWire has been open to any *Drosophila* researcher, and set forth clear principles for coordination of scientific effort that prioritized attribution through keeping track of edits to and annotations of the dataset. Hundreds of scientists and proofreaders from more than 50 laboratories joined FlyWire with more than 200 of them contributing more than 100 edits (Supplementary Table 1) and 86 contributing ten or more annotations (Supplementary Table 2). As a result of openness, there are multiple studies that used completed portions of FlyWire's brain connectome as proofreading proceeded (Supplementary Table 3). Openness has also enabled FlyWire to move faster by incorporating data sources from the community. The electron microscopy data on which FlyWire's brain connectome is built was shared in 2018 by Bock and colleagues[9]. The synapse data was published by Buhmann et al.[7], neurotransmitter labels were published by Eckstein et al.[10], numerous annotations were contributed by Schlegel et al.[12], neck connective neuron annotations were contributed by Stürner et al.[36], optic lobe annotations were contributed by Matsliah et al.[11] and so far, more than 90,000 cell annotations have been shared by the community. Many cells have received multiple annotations from these sources, and discrepancies will continue to be adjudicated by the community, a process that has improved accuracy in cell-type classification. Overall, we anticipate that similar approaches based on an open ecosystem will enable connectomics to scale more efficiently, economically and equitably.

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

## The FlyWire Consortium

Krzysztof Kruk[3], Doug Bland[1], Zairene Lenizo[16], Austin T. Burke[1], Kyle Patrick Willie[1], Alexander Shakeel Bates[4,5,12,13], Nikitas Serafetinidis[3], Nashra Hadjerol[16], Ryan Willie[1], Katharina Eichler[5], Ben Silverman[1], John Anthony Ocho[16], Joshua Bañez[16], Yijie Yin[5], Rey Adrian Candilada[16], Sven Dorkenwald[1,2], Jay Gager[1], Anne Kristiansen[3], Nelsie Panes[16], Arti Yadav[17], Remer Tancontian[16], Shirleyjoy Serona[16], Jet Ivan Dolorosa[16], Kendrick Joules Vinson[16], Dustin Garner[18], Regine Salem[16], Ariel Dagohoy[16], Philipp Schlegel[4,5], Jaime Skelton[3], Mendell Lopez[16], Laia Serratosa Capdevila[5], Griffin Badalamente[5], Thomas Stocks[3], Anjali Pandey[17], Darrel Jay Akiatan[16], James Hebditch[1], Celia David[1], Dharini Sapkal[17], Shaina Mae Monungolh[16], Varun Sane[5], Mark Lloyd Pielago[16], Miguel Albero[16], Jacquilyn Laude[16], Márcia dos Santos[5], David Deutsch[1,9], Zeba Vohra[17], Kaiyu Wang[14], Allien Mae Gogo[16], Emil Kind[19], Alvin Josh Mandahay[16], Chereb Martinez[16], John David Asis[16], Chitra Nair[17], Dhwani Patel[17], Marchan Manaytay[16], Imaan F. M. Tamimi[5], Clyde Angelo Lim[16], Philip Lenard Ampo[16], Michelle Darapan Pantujan[16], Alexandre Javier[16], Daril Bautista[16], Rashmita Rana[17], Jansen Seguido[16], Bhargavi Parmar[17], John Clyde Saguimpa[16], Merlin Moore[1], Markus William Pleijzier[4], Mark Larson[20], Joseph Hsu[5], Itisha Joshi[17], Dhara Kakadiya[17], Amalia Braun[21], Cathy Pilapil[16], Marina Gkantia[5], Kaushik Parmar[17], Quinn Vanderbeck[12], Claire E. McKellar[1], Irene Salgarella[5], Christopher Dunne[5], Eva Munnelly[5], Chan Hyuk Kang[22], Lena Lörsch[23], Jinmook Lee[22], Lucia Kmecova[24], Gizem Sancer[25], Christa Baker[1], Szi-chieh Yu[1], Jenna Joroff[12], Steven Calle[24], Yashvi Patel[17], Olivia Sato[20], Siqi Fang[5], Janice Salocot[16], Farzaan Salman[26], Sebastian Molina-Obando[23], Paul Brooks[5], Mai Bui[27], Matthew Lichtenberger[3], Edward Tamboboy[16], Katie Molloy[20],

Alexis E. Santana-Cruz[24], Anthony Hernandez[3], Seongbong Yu[22], Marissa Sorek[1,3], Arzoo Diwan[17], Monika Patel[17], Travis R. Aiken[3], Sarah Morejohn[1], Sanna Koskela[14], Tansy Yang[14], Daniel Lehmann[3], Jonas Chojetzki[23], Sangeeta Sisodiya[17], Selden Koolman[1], Philip K. Shiu[28], Sky Cho[27], Annika Bast[23], Brian Reicher[20], Marlon Blanquart[5], Lucy Houghton[18], Hyungjun Choi[22], Maria Ioannidou[23], Matt Collie[20], Joanna Eckhardt[1], Benjamin Gorko[18], Li Guo[18], Zhihao Zheng[1], Alisa Poh[29], Marina Lin[27], István Taisz[4], Wes Murfin[30], Álvaro Sanz Díez[31], Nils Reinhard[32], Peter Gibb[12], Nidhi Patel[17], Sandeep Kumar[1], Minsik Yun[33], Megan Wang[1], Devon Jones[1], Lucas Encarnacion-Rivera[34], Annalena Oswald[23], Akanksha Jadia[17], Mert Erginkaya[35], Nik Drummond[5], Leonie Walter[19], Ibrahim Tastekin[35], Xin Zhong[19], Yuta Mabuchi[36], Fernando J. Figueroa Santiago[24], Urja Verma[17], Nick Byrne[20], Edda Kunze[19], Thomas Crahan[18], Ryan Margossian[3], Haein Kim[36], Iliyan Georgiev[3], Fabianna Szorenyi[24], Atsuko Adachi[31], Benjamin Bargeron[37], Tomke Stürner[4,5], Damian Demarest[38], Burak Gür[23], Andrea N. Becker[3], Robert Turnbull[5], Ashley Morren[3], Andrea Sandoval[28], Anthony Moreno-Sanchez[39], Diego A. Pacheco[12], Eleni Samara[21], Haley Croke[39], Alexander Thomson[14], Connor Laughland[14], Suchetana B. Dutta[19], Paula Guiomar Alarcón de Antón[19], Binglin Huang[18], Patricia Pujols[24], Isabel Haber[20], Amanda González-Segarra[28], Albert Lin[1,6], Daniel T. Choe[40], Veronika Lukyanova[41], Marta Costa[5], Nino Mancini[37], Zequan Liu[42], Tatsuo Okubo[12], Miriam A. Flynn[14], Gianna Vitelli[37], Meghan Laturney[28], Feng Li[14], Shuo Cao[43], Carolina Manyari-Diaz[37], Hyunsoo Yim[40], Anh Duc Le[39], Kate Maier[37], Seungyun Yu[22], Yeonju Nam[22], Daniel Bąba[3], Amanda Abusaif[28], Audrey Francis[44], Jesse Gayk[17], Sommer S. Huntress[45], Raquel Barajas[35], Mindy Kim[20], Xinyue Cui[36], Amy R. Sterling[1,3], Gabriella R. Sterne[28], Anna Li[12], Keehyun Park[22], Georgia Dempsey[5], Alan Mathew[5], Jinseong Kim[22], Taewan Kim[22], Guan-ting Wu[46], Serene Dhawan[47], Margarida Brotas[35], Cheng-hao Zhang[46], Shanice Bailey[5], Alexander Del Toro[28], Arie Matsliah[1], Kisuk Lee[1,10], Thomas Macrina[1,2], Casey Schneider-Mizell[7], Sergiy Popovych[1,2], Oluwaseun Ogedengbe[1], Runzhe Yang[1], Akhilesh Halageri[1], Will Silversmith[1], Stephan Gerhard[48], Andrew Champion[4,5], Nils Eckstein[14], Dodam Ih[1], Nico Kemnitz[1], Manuel A. Castro[1], Zhen Jia[1], Jingpeng Wu[1], Eric Mitchell[1], Barak Nehoran[1,2], Shang Mu[1], J. Alexander Bae[1,11], Ran Lu[1], Eric Perlman[8], Ryan Morey[1], Kai Kuehner[1], Derrick Brittain[7], Chris S. Jordan[1], David J. Anderson[43], Rudy Behnia[31], Salil S. Bidaye[37], Davi D. Bock[15], Alexander Borst[21], Eugenia Chiappe[35], Forrest Collman[7], Kenneth J. Colodner[45], Andrew Dacks[26], Barry Dickson[14], Jan Funke[14], Denise Garcia[39], Stefanie Hampel[24], Volker Hartenstein[49], Bassem Hassan[19], Charlotte Helfrich-Forster[32], Wolf Huetteroth[50], Gregory S. X. E. Jefferis[4,5], Jinseop Kim[22], Sung Soo Kim[18], Young-Joon Kim[33], Jae Young Kwon[22], Wei-Chung Lee[12], Gerit A. Linneweber[19], Gaby Maimon[44], Richard Mann[31], Mala Murthy[1], Stéphane Noselli[51], Michael Pankratz[38], Lucia Prieto-Godino[47], Jenny Read[41], Michael Reiser[14], Katie von Reyn[39], Carlos Ribeiro[35], Kristin Scott[28], Andrew M. Seeds[24], Mareike Selcho[50], H. Sebastian Seung[1,2], Marion Silies[23], Julie Simpson[18], Scott Waddell[52], Mathias F. Wernet[19], Rachel I. Wilson[12], Fred W. Wolf[53], Zepeng Yao[54], Nilay Yapici[36] & Meet Zandawala[32]

[16]SixEleven, Davao City, Philippines. [17]ariadne.ai, Buchrain, Switzerland. [18]University of California, Santa Barbara, CA, USA. [19]Freie Universität Berlin, Berlin, Germany. [20]Harvard, Boston, MA, USA. [21]Department Circuits–Computation–Models, Max Planck Institute for Biological Intelligence, Planegg, Germany. [22]Sungkyunkwan University, Seoul, South Korea. [23]Johannes-Gutenberg University Mainz, Mainz, Germany. [24]Institute of Neurobiology, University of Puerto Rico Medical Sciences Campus, San Juan, Puerto Rico. [25]Department of Neuroscience, Yale University, New Haven, CT, USA. [26]Department of Biology, West Virginia University, Morgantown, WV, USA. [27]Program in Neuroscience and Behavior, Mount Holyoke College, South Hadley, MA, USA. [28]University of California, Berkeley, CA, USA. [29]University of Queensland, Brisbane, Queensland, Australia. [30]Independent researcher, Fort Collins, CO, USA. [31]Zuckerman Institute, Columbia University, New York, NY, USA. [32]Julius-Maximilians-Universität Würzburg, Würzburg, Germany. [33]Gwangju Institute of Science and Technology, Gwangju, South Korea. [34]Stanford University School of Medicine, Stanford, CA, USA. [35]Champalimaud Foundation, Lisbon, Portugal. [36]Cornell University, Ithaca, NY, USA. [37]Max Planck Florida Institute for Neuroscience, Jupiter, FL, USA. [38]University of Bonn, Bonn, Germany. [39]Drexel, Philadelphia, PA, USA. [40]Seoul National University, Seoul, South Korea. [41]Newcastle University, Newcastle, UK. [42]RWTH Aachen University, Aachen, Germany. [43]Caltech, Pasadena, CA, USA. [44]Rockefeller University, New York, NY, USA. [45]Mount Holyoke College, South Hadley, MA, USA. [46]National Hualien Senior High School, Hualien, Taiwan. [47]The Francis Crick Institute, London, UK. [48]Aware, Zurich, Switzerland. [49]University of California, Los Angeles, CA, USA. [50]Institute of Biology, Leipzig University, Leipzig, Germany. [51]Université Côte d'Azur, CNRS, Inserm, iBV, Nice, France. [52]University of Oxford, Oxford, UK. [53]University of California, Merced, CA, USA. [54]University of Florida, Gainesville, FL, USA.

# Methods

## Specimen, alignment and segmentation

Multiple brains of 7-day-old (iso) w1118 × (iso) Canton-S G1 adult female flies were screened by Zheng et al.[9] and one was picked for electron microscopy imaging. Zheng et al.[9] published the original electron microscopy stack (FAFB14) which we previously realigned[8] (FAFB14.1) using a neural network trained to predict pairwise displacement fields[108,120]. We produced transformations between the FAFB14 and FAFB14.1 which are accessible via natverse, navis and flybrains. We automatically segmented all cells in the dataset[8] using a neuronal boundary-detecting neural network[121] and mean affinity agglomeration[107,122].

## Neuropils

Meshes for individual neuropils were based on work by Ito et al.[80]. More specifically, we took meshes previously generated from a full brain segmentation of the JFRC2 template brain which are also used by the Virtual Fly Brain project (see also https://natverse.org/nat.flybrains/reference/JFRC2NP.surf.html). These meshes were moved from JFRC2 into FlyWire (FAFB14.1) space through a series of non-rigid transforms. In addition, we also generated two neuropil meshes for the laminae and for the ocellar ganglion. For these, the FlyWire synapse cloud was voxelized with 2 μm isotropic resolution, meshed using the marching cube algorithm using Python and manually post-processed in Blender 3d.

We calculated a volume for each neuropil using its mesh. In the aggregated volumes presented in the paper we assigned the lamina, medulla, accessory medulla, lobula and lobula plate to the optic lobe. The remaining neuropils but the ocellar ganglion were assigned to the central brain.

## Neuropil synapse assignments

We assigned synapses to neuropils based on their presynaptic location. We used ncollpyde (https://pypi.org/project/ncollpyde/) to calculate whether the location was within a neuropil mesh and assigned the synapse accordingly. Some synapses remained unassigned after this step because the neuropils only resemble rough outlines of the underlying data. We then assigned all remaining synapses to the closest neuropil if the synapse was within 10 μm from it. The remaining synapses were left unassigned.

## Correction of left–right inversion

Our reconstruction used the FAFB electron microscopy dataset[9]. A number of consortium members (A. Bates, P. Kandimalla and S. Noselli) alerted us that the FAFB imagery seemed to be left–right inverted based on the cell types innervating the asymmetric body[123]. Eventually a left–right inversion during FAFB imaging was confirmed. All side annotations in figures, in Codex and elsewhere are based on the true biological side. For technical reasons, we were unable to invert the underlying FAFB image data and therefore continue to show images and reconstructions in the same orientation as in Zheng et al.[9], although we now know that in such frontal views the fly's left is on the viewer's left. For full details of this issue including approaches to display FAFB and other brain data with the correct chirality, please see the companion paper[12].

## Proofreading system

FlyWire uses CAVE[50] for hosting the proofreadable segmentation and all of its annotations. CAVE's proofreading system is the PyChunkedGraph which has been described in detail elsewhere[8,124].

## Proofreading annotations

Any user in FlyWire was able to mark a cell as complete, indicating that a cell was good for analysis. However, such annotations did not prevent future proofreading of a cell as commonly smaller branches were added later on. We created an annotation table for these completion markings. Each completion marking was defined by a point in space and the cell segment that overlapped with this point at any given time during proofreading was associated with the annotation. We created a webservice allowing users to submit completion markings for any cell. For convenience, we added an interface to this surface directly into Neuroglancer such that users can submit completion information for cells right after proofreading (Supplementary Fig. 10). When users submitted completion annotations we also recorded the current state of the cell. We encouraged users to submit new completion markings for a cell that they edited to indicate that edits were intentional. Recording the status of a cell at submission enabled us to calculate volumetric changes to a cell through further proofreading and flag cells for review if they received substantial changes without new completion markings.

## Onboarding proofreaders

Proofreaders came from several distinct labour pools: community members, citizen scientists from Eyewire (Flyers), and professional proofreading teams at Princeton and Cambridge. Proofreaders at Princeton consisted of staff at Princeton University and at SixEleven. Similarly, proofreading at Cambridge was performed by staff at Cambridge University and Ariadne. All proofreaders completed the built-in interactive tutorial and were directed to self-guided proofreading training. For practice and learning purposes, the Sandbox, a complete replica of the FlyWire data, allowed new users to freely make edits and explore without affecting the actual 'Production' dataset. When ready, an onboarding coordinator tested the new proofreader before giving access to the Production dataset[8]. Later onboarding called for users to send demonstration Sandbox edits that were reviewed by the onboarding coordinator. A new class of view-only users was introduced in early 2023, allowing researchers early data access for analysis purposes. All early access users attended a live onboarding session in Zoom prior to being granted edit or view access.

## Training the professional proofreading team

The professional proofreading team received additional proofreading training. Correct proofreading relies on a diverse array of 2D and 3D visual cues. Proofreaders learned about 3D morphology, resulting from false merger or false split without knowing what types of cells they are. Proofreaders studied various types of ultrastructures as the ultrastructures provide valuable 2D cues and serve as reliable guides for accurate tracing. Before professional proofreaders were admitted into Production, each of them practiced on average >200 cells in a testing dataset where additional feedback was given. In this dataset, we determined the accuracy of test cells by comparing them to ground-truth reconstructions. To improve proofreading quality, peer learning was highly encouraged.

## Recruitment of citizen scientists

The top 100 players from Eyewire, a gamified electron microscopy reconstruction platform that crowdsources reconstructions in mouse retina and zebrafish hindbrain[125], received an invitation to beta test proofreading in FlyWire. A new set of user onboarding and training materials were created for citizen scientists, including: a blog, forum and public Google docs. We created bite-sized introduction videos, a comprehensive FlyWire 101 resource, as well as an Optic Lobe Cell Guide to aid users in understanding the unique morphology of flies. A virtual Citizen Science Symposium introduced players to the project, after which the self-dubbed 'Flyers' began creating their own resources, such as a new comprehensive visual guide to cell types, conducting literature reviews, and even developing helpful FlyWire plugins. As of publication, FlyWire has 12 add-on apps ranging from a batch processor to cell naming helper (https://blog.flywire.ai/2022/08/11/flywire-addons/).

## Proofreading strategy to complete the connectome

As previously described[8], proofreading of the connectome was focused on the microtubule-rich 'backbones' of neurons. Microtubule-free

'twigs' were only added if discovered incidentally or sought out specifically by members of the community. After proofreading, users marked neuronal segments as 'complete' indicating that neurons were ready for analysis but further changes remained possible. While *Drosophila* neuroscientist members of the FlyWire community generally contributed proofreading for their neurons of interest, the bulk of the segments was proofread by professional proofreaders in the following way. First, we proofread all segments with an automatically detected nucleus in the central brain[88] by extending it as much as possible and removing all false mergers (pieces of other neurons or glia attached). Second, we proofread the remaining segments in descending order of their synapse count (pre+post) up to a predefined size threshold of 100 synapses. Third, we proofread remaining segments if they had at least one connection containing at least 15 synapses.

## Quality assurance

To assess quality, a group of expert centralized proofreaders conducted a review of 3,106 segments in the central brain. These specific neurons were chosen based on certain criteria such as significant change since being marked complete and small overall volume. An additional 826 random neurons were included in the review pool as well. Proofreaders were unaware which neurons were added for quality measurement and which ones because they were flagged by a metric. We compared the 826 neurons before and after the review and found that the initial reconstruction scored an average F1-Score of 99.2% by volume (Extended Data Fig. 2a,b). F1-Score is defined as the harmonic mean of recall (*R*) and precision (*P*) with precision defined as the ratio of true positives (TP) among positively classified elements (TP + FP (false positives)) and recall defined as the ratio of TPs among all true elements (TP + FN (false negatives)).

$$P = \frac{TP}{TP + FP}$$
$$R = \frac{TP}{TP + FN}$$
$$F1 = \frac{2 \times P \times R}{P + R}$$

## Quantification of proofreading effort

Any quantification of the total proofreading time that was required to create the FlyWire resource is a rough estimate because of the distributed nature of the community, the interlacing of analysis and proofreading and the variability in how proofreading was performed. The second public release, version 783, required 3,013,513 edits. We measured proofreading times during early proofreading rounds that included proofreading of whole cells in the central brain. We collected timings and number of edits for 29,135 independent proofreading tasks after removing outliers with more than 500 edits. From these data, we were able to calculate an average time per edit. However, we observed that proofreading times per edit were much higher for proofreading tasks that required few edits (<5). That meant that our measurements were not representative for the second round of proofreading which went over all segments with >100 synapses. These usually required 1–5 edits. We adjusted for that by computing estimates for proofreading speeds of both rounds by limiting the calculations to a subset of the timed tasks: (round 1) The average time per edit in our proofreading time dataset, (round 2) the average time of tasks with 1–5 edits. We averaged these times for an overall proofreading time because the number of tasks in each category were similar. The result was an average time of 79 s per edit which adds up to an estimate of 33.1 person-years assuming a 2,000 h work year.

## Attachment rates

We adopted the attachment rate (also referred to as 'completion rate') calculations from the hemibrain[1]. Every presynaptic and postsynaptic location was assigned to a segment. Using the neuropil assignments, we then calculated the fraction of presynapses that were assigned to segments marked as proofread for each neuropil and analogous for postsynaptic locations.

## Comparison with the hemibrain

We retrieved the latest completion rates and synapse numbers for the hemibrain from neuprint (v1.2.1). In some cases, neuropil comparisons were not directly possible because of redefined regions in the hemibrain dataset. We excluded these regions from the comparison.

## Crowdsourced annotation

The large FlyWire community and diversity of expertise enabled us to crowdsource the identification of neurons. There is no limit to the number of annotations a neuron can receive. A standardized format is encouraged but not required. One user might first report that a neuron is a descending interneuron, whereas another might add that it is the giant fibre descending neuron, and yet another might add all its synonyms and citations from the literature. Contributors' names are visible so they can be consulted if there is disagreement. The disadvantage to this approach is that there is no single precise name for every neuron, but the advantage is a richness of information and dialogue. The annotations are not meant to be a finished, static list, but rather a continually growing, living data source. These annotations were solicited from the FlyWire community through town halls, email announcements, interest groups in the FlyWire forum, online instructions, and by personal contact from the community manager. Citizen scientists also contributed annotations, after receiving training on particular cell types by experts.

## Neuron categorizations

Neuron categorization, sensory modality annotations and nerve assignments are described in detail in our companion paper[12]. In brief, neurons were assigned to one of three 'flow' classes: afferent (to the brain), intrinsic (within the brain) and efferent (out of the brain). Intrinsic neurons had their entire arbor within the FlyWire brain dataset. This included cells that projected to and from the SEZ. Next, each flow class was divided into superclasses in the following way. afferent: sensory, ascending. intrinsic: central, optic, visual projection (from the optic lobes to the central brain), visual centrifugal (from the central brain to the optic lobes). efferent: endocrine, descending, motor.

## Quantification of intrinsic neurons

We define whether a neuron is 'intrinsic' to a region on the basis of its synapse locations, rather than its arbor. In other words, the neurites of an intrinsic neuron are allowed to exit the region, provided that they do not make synapses after leaving. Information about *C. elegans* synapse locations was obtained from the diagrams in White et al.[31].

The 'brain' of *C. elegans* can be defined as the neuropil extending from the ring-shaped structure around the pharynx to the excretory pore. (We follow the authors who call this region the nerve ring plus the anterior portion of the ventral nerve cord, though some authors refer to the combined structure as the nerve ring.) Nine neurons (RIR, RIV, RMDD, RMD and RMDV) are intrinsic to the nerve ring itself. An additional 26 neurons (AIA, AIB, AIM, AIN, AIY, AIZ, RIA, RIB, RIC, RIH, RIM, RIS, RMF and RMH) are intrinsic to the combined structure, for a total of 35 intrinsic neurons in the brain.

It should be understood that this estimate has 'error bars' because of definitional ambiguities. Ten motor neurons (RMH, RMF and RMD) could arguably be removed from the list, as it is unclear whether motor neurons qualify as intrinsic neurons. Or the brain could be enlarged by moving the posterior border further behind the excretory pore, which would add 10 neurons (RIF, RIG, RMG, ADE and ADA). To make these ambiguities explicit, we estimate 35 ± 10 intrinsic neurons. Of the 302 CNS neurons, 180 make synapses in the brain[126]. Therefore, neurons intrinsic to the brain make up about 15 to 25% of brain neurons, and 8 to 15% of CNS neurons.

## Skeletonization and path length calculation

We generated skeletons for all neurons marked as proofread using skeletor (https://github.com/navis-org/skeletor), which implements multiple skeletonization algorithms such as TEASAR[127]. In brief, neuron meshes from the exported segmentation (LOD 1) were downloaded and skeletonized using the wavefront method in skeletor. These raw skeletons were then further processed (for example, to remove false twigs and heal breaks) and produce downsampled versions using navis[128] (https://github.com/navis-org/navis). A modified version of this skeletonization pipeline is implemented in fafbseg (https://github.com/navis-org/fafbseg-py).

## Quantifying cell volume and surface area

We calculated cell volumes and surface areas using CAVE's L2Cache[50]. Volumes were computed by counting all voxels within a cell segment and multiplying the count by the voxel resolution. Area calculations were more complicated and were performed by overlap through shifts in voxel space. We shifted the binarized segment in each dimension individually and extracted the overlap of false and true voxels. For each dimension, we counted the extracted voxels and multiplied the count by the voxel resolution of the given dimensions. Finally, we added up per dimension area estimates. This measurement will overestimate area slightly but smoothed measurements are ill-defined and were too compute intensive.

## Synaptic connections

We imported the automatically predicted synapses from Buhmann et al.[7], which we combined with the synapse segmentations by Heinrich et al.[118] to assign scores to all synapses to improve precision. Buhmann et al. introduced a machine learning model to predict for each voxel whether it is part of a postsynaptic site. For voxels classified as postsynaptic a vector to the presynaptic site is predicted which is then used to created synaptic connections. Hence, synaptic partners predicted by Buhmann et al. are represented by a connector between a postsynaptic and a presynaptic location without further annotation about the size of the synapse. Heinrich et al. on the other hand segmented the synaptic clefts. Buhmann et al. suggested using the probability maps from Heinrich et al. to improve performance by locating the highest probability score along their predicted connectors (called score in the next paragraph).

The synapse classifier by Buhmann et al. was trained on ground truth from the CREMI challenge (https://cremi.org). The three CREMI datasets contain three $5 \times 5 \times 5$ μm cubes from the calyx in FAFB14 with 1,965 synapses. While the classifier from Buhmann et al. was trained and validated on only this dataset, they evaluated its performance on multiple regions (calyx, lateral horn, ellipsoid body and protocerebral bridge). It should be noted that performance varies by region.

The dataset published by Buhmann et al. contained ~244 million synapses. We removed synapses from the imported list if they fulfilled any of the following criteria: (1) either the pre- or postsynaptic location remained unassigned to a segment (proofread or unproofread); (2) It had a score ≤50. Additionally, we removed duplicate synapse annotations between the same pre- and postsynaptic partners, defined as those within a distance of 100 nm from another synapse annotation according to their presynaptic coordinate. After filtering, we were left with ~130 million synapses.

Eckstein et al.[10] created a machine learning model to predict neurotransmitter identities for all synapses from Buhmann et al. based on the electron microscopy imagery alone. Each synapse was assigned a probability for one of six neurotransmitters: acetylcholine, glutamate, GABA, serotonin, dopamine and octopamine. They used neurotransmitter identities published for individual neuronal cell types and built a dataset with 3,025 neurons with known transmitter type assuming Dale's law applies. Eckstein et al. reported a per-synapse accuracy of 87% and a per neuron (majority vote) accuracy of 94%.

The methods described in this section used the FAFB14 version of the electron microscopy stack. We applied a transformation to all synapses to map them into the FlyWire FAFB14.1 space. The vector field for the transformation had a resolution of $64 \times 64 \times 40$ nm.

## Connection threshold

For all the analyses presented in this paper, save for synapse distributions, we employed a consistent threshold of >4. Our decision to use a synapse threshold on connections was due partly to the fact that synapses in the FlyWire's brain dataset were not manually proofread. For these analyses, many of which demonstrate the high interconnectivity of the fly brain, we chose a conservative threshold to ensure that considered connections are real. Use of a threshold is also in keeping with previous work analysing wiring diagrams in *Drosophila*[1]. Thus, we are probably undercounting the number of true connections. The distribution of synapse counts (Fig. 3f) does not display any bimodality that could be used to set the threshold. Therefore, the choice of 5 synapses per connection is a reasonable but arbitrary one. By analysing the network properties of the FlyWire brain connectome, Lin et al. found that statistical properties of the whole-brain network, such as reciprocity and clustering coefficient, are robust to our choice of threshold[49]. The FlyWire data are available without an imposed threshold, so users can choose their own appropriate threshold for their specific use case.

## Neuropil projectome construction

Under the simplifying assumptions that information flow through the neuron can be approximated by the fraction of synapses in a given region, and that inputs and outputs can be treated independently, we can construct a matrix representing the projections of a single neuron between neuropils. The fractional inputs of a given neuron are a $1 \times N$ vector containing the fraction of incoming synapses the neuron has in each of the $N$ neuropils, and the fractional outputs are a similar vector containing the fraction of outgoing synapses in each of the $N$ neuropils. We multiply these vectors against each other to generate the $N \times N$ matrix of the neuron's fractional weights. Summing these matrices across all intrinsic neurons produces a matrix of neuropil-to-neuropil connectivity (Fig. 4a). In this projectome, all neurons contribute an equal total weight of one.

## Dominant input side

We assigned neuropils to the left and right hemispheres or the centre if the neuropil has no homologue. We then counted how many postsynapses each neuron had in each of these three regions and assigned it to the one with the largest count.

## Contralateral and bilateral neuron analysis

For each neuron, we calculated the fraction of presynapses in the left and right hemisphere. The hemisphere opposite its dominant input side was named the contralateral hemisphere. We excluded neurons that had either most of their presynapses or most of their postsynapses in the centre region.

## Rank analysis and information flow

We used the information flow algorithm implemented by Schlegel et al.[26,128] (https://github.com/navis-org/navis) to calculate a rank for each neuron starting with a set of seed neurons. The algorithm traverses the synapse graph of neurons probabilistically. The likelihood of a neuron being added to the traversed set increased linearly with the fraction of synapses it receives from already traversed neurons up to 30% and was guaranteed above this threshold. We repeated the rank calculation for all sets of afferent neurons as seed as well as the whole set of sensory neurons. The groups we used are: olfactory receptor neurons, gustatory receptor neurons, mechanosensory Johnston's organ neurons, head and neck bristle mechanosensory neurons, mechanosensory taste

peg neurons, thermosensory neurons, hygrosensory neurons, VPNs, visual photoreceptors, ocellar photoreceptors and ascending neurons.

Additionally, we created input seeds by combining all listed modalities, all sensory modalities, and all listed modalities with visual sensory groups excluded.

For each modality we performed 10,000 runs, which were averaged. We then ordered the neurons according to their rank and assigned them a percentile based on their location in the order. To compute a reduced dimensionality, we treated the vector of all ranks (one for each modality) as neuron embedding and calculated two dimensional embeddings using UMAP[129] with the following parameters: n_components=2, min_dist=0.35, metric = "cosine", n_neighbors=50, learning_rate = .1, n_epochs=1000.

## Reporting summary

Further information on research design is available in the Nature Portfolio Reporting Summary linked to this article.

## Data availability

All data have been made publicly available. Codex (https://codex.flywire.ai/), braincircuits.io and Catmaid spaces (https://fafb-flywire.catmaid.org/) facilitate non-programmatic access. Most of the data can be directly download from codex (https://codex.flywire.ai/api/download). All data, including the volumetric data and meshes, can be programmatically accessed through CAVE and cloudvolume. We provide tutorials for programmatic access at https://github.com/seung-lab/FlyConnectome. Data dumps of the connectivity data (https://doi.org/10.5281/zenodo.10676866) and flow calculations (https://doi.org/10.5281/zenodo.12588557) are made available on zenodo for download.

## Code availability

FlyWire uses CAVE for hosting of its proofreading and analysis platform for which all code is publicly available at https://github.com/CAVEconnectome. The code for Codex is available at https://github.com/murthylab/codex.

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

**Acknowledgements** The authors thank J. Wiggins, G. McGrath and D. Barlieb for computer system administration; M. Husseini for project administration; J. Maitin-Shepard for Neuroglancer; P. N. Gomez for help with GPU-cluster deployment; the management at SixEleven and Ariadne for coordination and proofreader management; and T. Sloan for creating the whole-brain renderings and videos. We thank O. Hobert, S. Cook and S. Emmons for contributing their expertise on *C. elegans*. M.M. and H.S.S. acknowledge support from the National Institutes of Health (NIH) BRAIN Initiative RF1 MH117815, RF1 MH129268 and U24 NS126935, from the Princeton Neuroscience Institute, as well as assistance from Google and Amazon. D.D.B. was supported by NIH NIMH BRAIN Initiative grant 1RF1MH120679-01 and a Neuronex2 award (NSF 2014862). G.S.X.E.J. and D.D.B. were supported by a Wellcome Trust Collaborative Award (203261/Z/16/Z). G.S.X.E.J. was supported by Wellcome Trust Collaborative Award 220343/Z/20/Z and a Neuronex2 award (MRC MC_EX_MR/T046279/1), and received core support from the MRC (MC-U105188491). A.L. was supported by the NSF through the Center for the Physics of Biological Function (PHY-1734030). I. Tastekin was supported with a Marie Skłodowska-Curie postdoctoral fellowship (H2020-WF-01-2018-867459 to Ibrahim Tastekin) and by the Portuguese Research Council (Grant PTDC/MED-NEU/4001/2021). A.S. and S.H. were supported by National Institute Of Neurological Disorders And Stroke of the National Institutes of Health under Award Number RF1NS121911. D. Brittain, C.S.-M. and F.C. thank the Allen Institute for Brain Science founder, P. G. Allen, for his vision, encouragement and support. This work was also supported by the Intelligence Advanced Research Projects Activity via Department of Interior/Interior Business Center contract no. D16PC0005 to H.S.S. The US Government is authorized to reproduce and distribute reprints for Governmental purposes notwithstanding any copyright annotation thereon. The views and conclusions contained herein are those of the authors and should not be interpreted as necessarily representing the official policies or endorsements, either expressed or implied, of Intelligence Advanced Research Projects Activity, Department of Interior/Interior Business Center or the US Government.

**Author contributions** Members of the FlyWire Consortium contributed proofreading and annotations (Supplementary Tables 1 and 2). S.G. provided braincircuits.io. T.M. and N.K. realigned the dataset with methods developed by E.M., B.N. and T.M. and infrastructure developed by S.P. and Z.J. J.A.B. and S.M. wrote code for masking defects and misalignments. K.L. trained the convolutional net for boundary detection, using ground-truth data realigned by D.I. J.W. used the convolutional net to generate an affinity map that was segmented by R.L. N.K., M.A.C., O.O., A.H., C.S.J., K. Kuehner and A.R.S. adapted and improved Neuroglancer for proofreading and annotations. J.G., K. Kruk, S.D. F.C. and C.S.-M. created interactive analysis and annotation tools for the community. A.M. created Codex with help from A.R.S., S.D., K. Kuehner and R.M. A.R.S. and A.M. created the website. A.R.S., C.E.M. and M.S. onboarded community members and tested new proofreaders. A.R.S., M.S., C.S.J. and C.E.M. designed tutorials. C.E.M., A.R.S. and M.S. provided community support. S.D., F.C., C.S.-M., C.S.J., A.H., D. Brittain and W.S. built and maintained CAVE for FlyWire and managed user access. S.D., P.S., A.M. and E.P. curated the data and made it available for download. E.P. and D.D.B. provided a coordinate mapping service. A.S.B., N.E., G.S.X.E.J. and J.F. provided neurotransmitter information. S.-c.Y., C.E.M., M.C., K.E., Y.Y. and P.S. trained and managed proofreaders. S.D., S.-c.Y., P.S. and G.S.X.E.J. led the targeted proofreading effort. S.D., P.S., A.M., A.C. and K. Kuehner maintained the proofreading management platforms. S.D. evaluated the proofreading accuracy. S.D., A.L., H.S.S., D.D. and R.Y. analysed the data. S.D., D. Bland and S.-c.Y. annotated and analysed the ocellar circuit. S.D., H.S.S., M.M., A.L., P.S. and A.R.S. wrote the manuscript with feedback from A.S.B., W.H., G.S.X.E.J. and contributions from all authors. H.S.S., M.M., G.S.X.E.J. and D.D.B. sponsored large-scale proofreading. G.S.X.E.J. and D.D.B. led the Cambridge effort. M.M. and H.S.S. led the overall effort.

**Competing interests** T.M., K.L., S.P., D.I., N.K. and H.S.S. declare financial interests in Zetta AI.

**Additional information**
**Correspondence and requests for materials** should be addressed to H. Sebastian Seung or Mala Murthy.

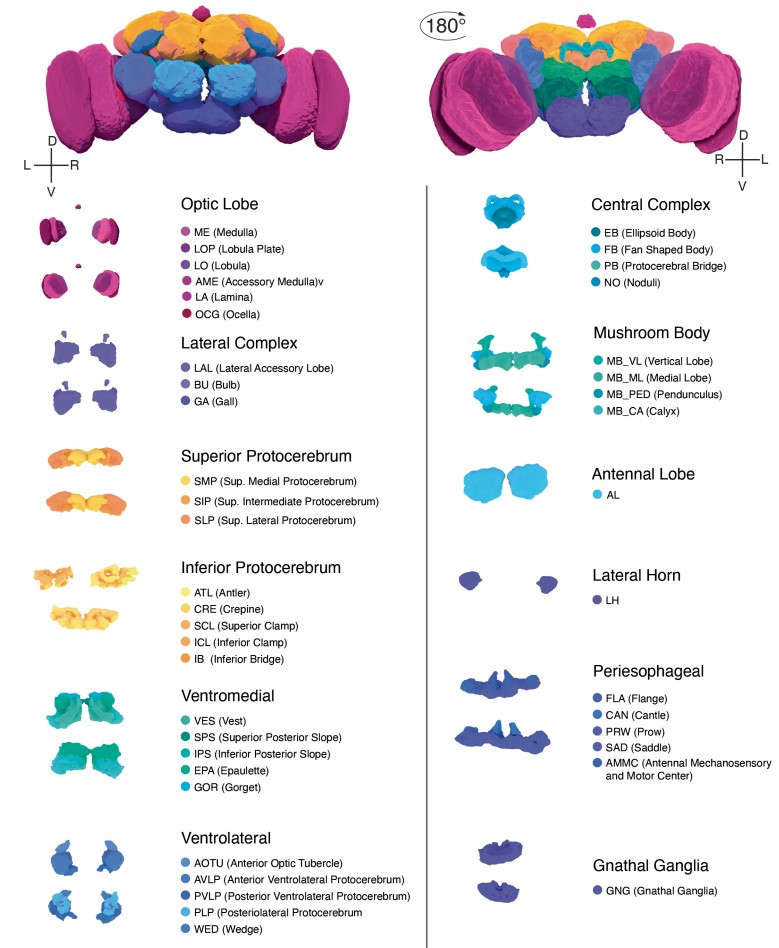

**180°**

**Optic Lobe**
- ME (Medulla)
- LOP (Lobula Plate)
- LO (Lobula)
- AME (Accessory Medulla)v
- LA (Lamina)
- OCG (Ocella)

**Lateral Complex**
- LAL (Lateral Accessory Lobe)
- BU (Bulb)
- GA (Gall)

**Superior Protocerebrum**
- SMP (Sup. Medial Protocerebrum)
- SIP (Sup. Intermediate Protocerebrum)
- SLP (Sup. Lateral Protocerebrum)

**Inferior Protocerebrum**
- ATL (Antler)
- CRE (Crepine)
- SCL (Superior Clamp)
- ICL (Inferior Clamp)
- IB (Inferior Bridge)

**Ventromedial**
- VES (Vest)
- SPS (Superior Posterior Slope)
- IPS (Inferior Posterior Slope)
- EPA (Epaulette)
- GOR (Gorget)

**Ventrolateral**
- AOTU (Anterior Optic Tubercle)
- AVLP (Anterior Ventrolateral Protocerebrum)
- PVLP (Posterior Ventrolateral Protocerebrum)
- PLP (Posteriolateral Protocerebrum)
- WED (Wedge)

**Central Complex**
- EB (Ellipsoid Body)
- FB (Fan Shaped Body)
- PB (Protocerebral Bridge)
- NO (Noduli)

**Mushroom Body**
- MB_VL (Vertical Lobe)
- MB_ML (Medial Lobe)
- MB_PED (Pendunculus)
- MB_CA (Calyx)

**Antennal Lobe**
- AL

**Lateral Horn**
- LH

**Periesophageal**
- FLA (Flange)
- CAN (Cantle)
- PRW (Prow)
- SAD (Saddle)
- AMMC (Antennal Mechanosensory and Motor Center)

**Gnathal Ganglia**
- GNG (Gnathal Ganglia)

**Extended Data Fig. 1 | Neuropils of the fly brain.**

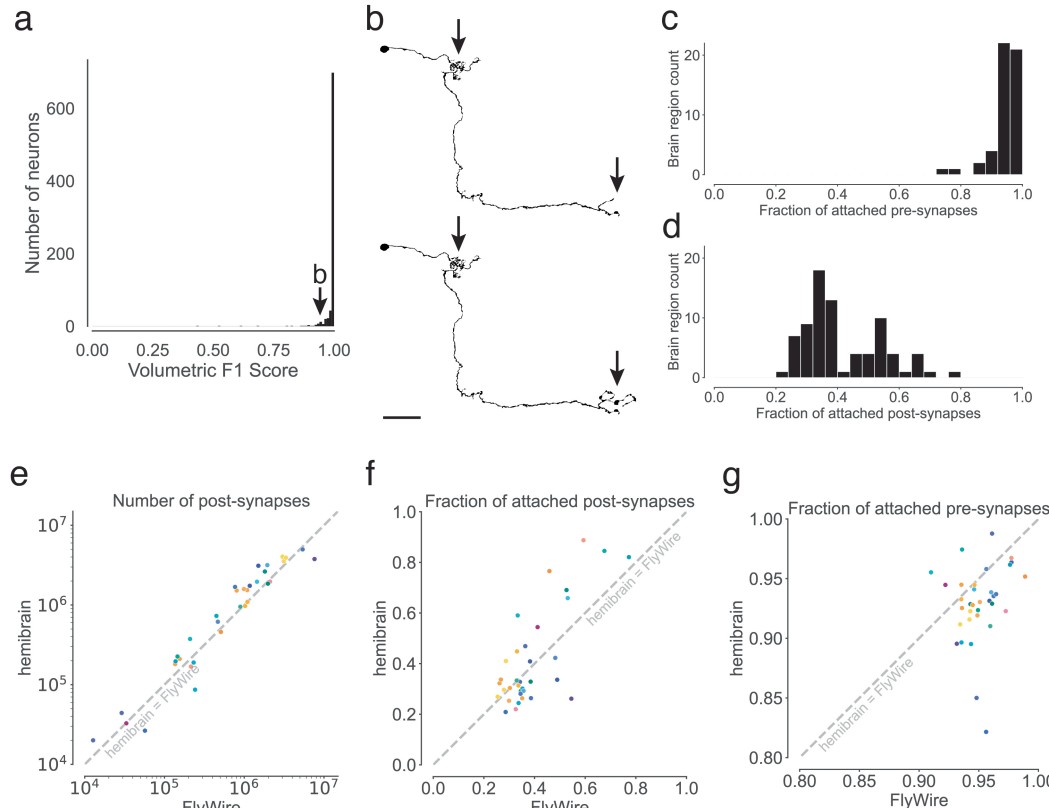

**Extended Data Fig. 2 | Completeness and accuracy of FlyWire's reconstruction.** (a) shows the result of our evaluation of proofread segments in the central brain. Experts attempted further proofreading of 826 neurons. We computed volumetric overlaps between the original and the final segment to calculate precision, recall, and F1 Scores. (b) Examples (top: before, bottom: after) of the changes made during further proofreading for a neuron scoring an F1-Score of 0.936. Arrows highlight locations that changed. (c,d) For each neuropil, we quantified what fraction of the synapses within it are pre- and postsynaptically attached to a proofread segment. (c) displays the distribution for presynaptic attachment and (d) the distribution for postsynaptic attachment. (e, f, g) Comparisons between FlyWire's reconstruction and the hemibrain were made for overlapping neuropils. Dots represent neuropils and are colored according to Extended Data Fig. 1. (e) Comparison of the number of automatically detected synapses. The axes are log-transformed. (f) Comparison of post-synaptic completion rates and (g) pre-synaptic completion rate. The axes are truncated.

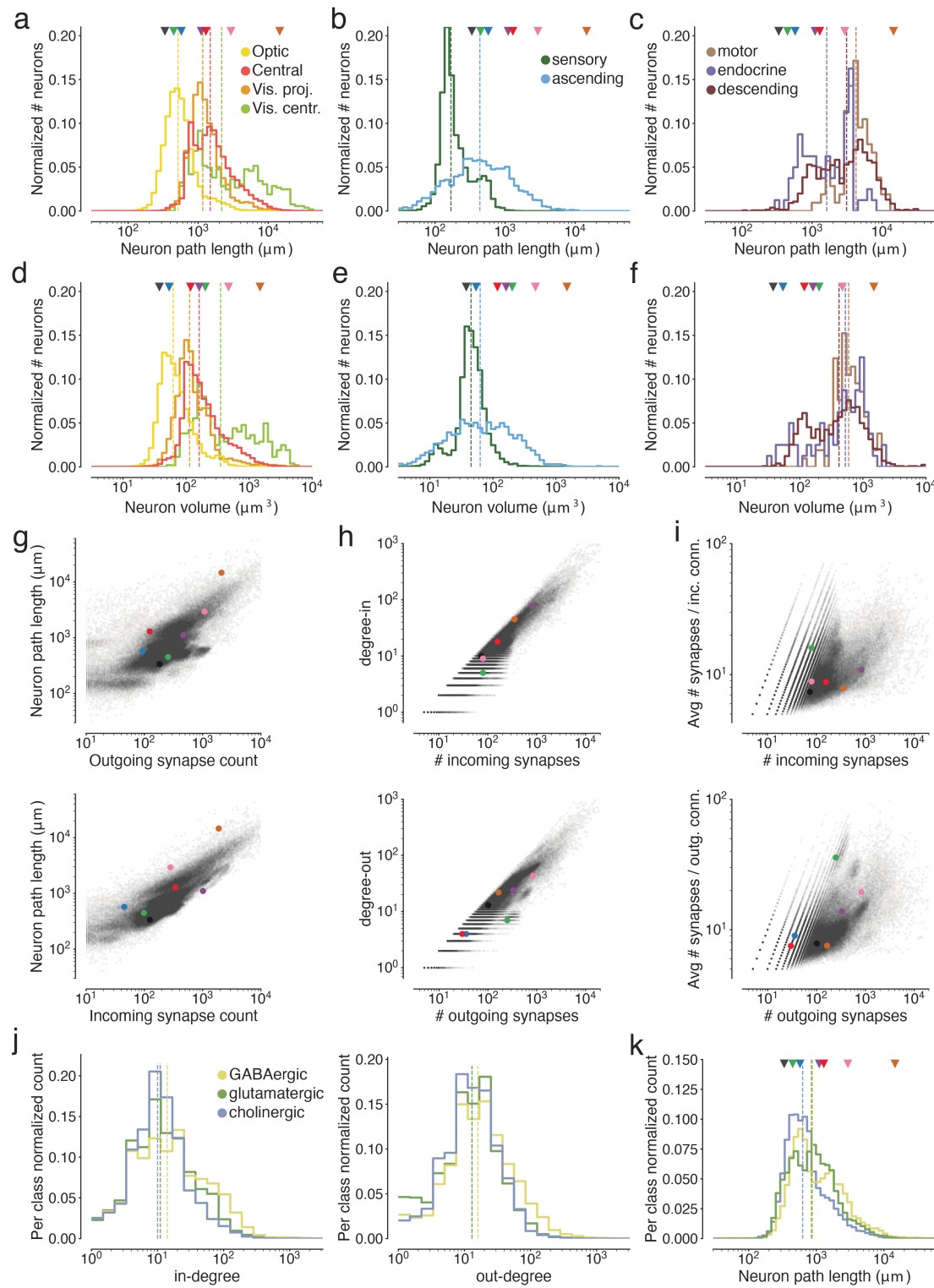

**Extended Data Fig. 3 | Measurements of neuron size.** Colored markers refer to neurons in Fig. 3b. Vertical dashed lines are medians. (a) Neuron path lengths of intrinsic neurons, (b) afferent neurons, and (c) efferent neurons by super-class. (d) Volumes of intrinsic neurons, (e) afferent neurons, and (f) efferent neurons by super-class. (g) Comparisons of path lengths and number of incoming and outgoing synapses. (h) For intrinsic neurons, comparisons of the in- and out-degrees with the number of incoming and outgoing synapses. Every dot is a neuron. (i) Comparison of average connection strengths (synapses per connection) with the number of synapses. Every dot is a neuron. (j) In- and out-degree distributions by neurotransmitter type. (k) Neuron path lengths by neurotransmitter type.

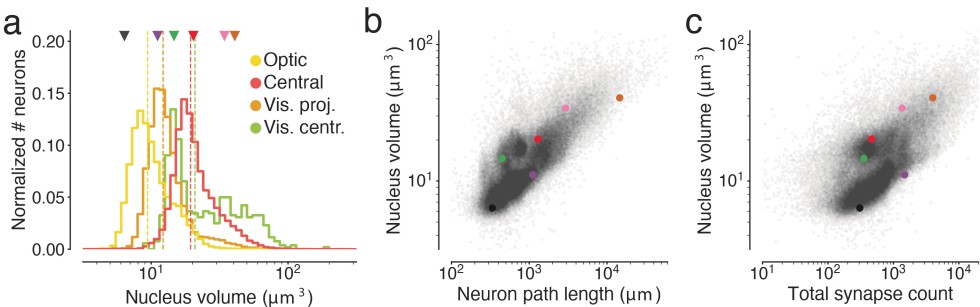

**Extended Data Fig. 4 | Measurements of neuron size.** Colored markers refer to neurons in Fig. 3b. Vertical dashed lines are medians. (a) Nucleus volume of intrinsic neurons, (b) Comparisons of nucleus volume and path length for intrinsic neurons and (c) nucleus volume and total synapse count.

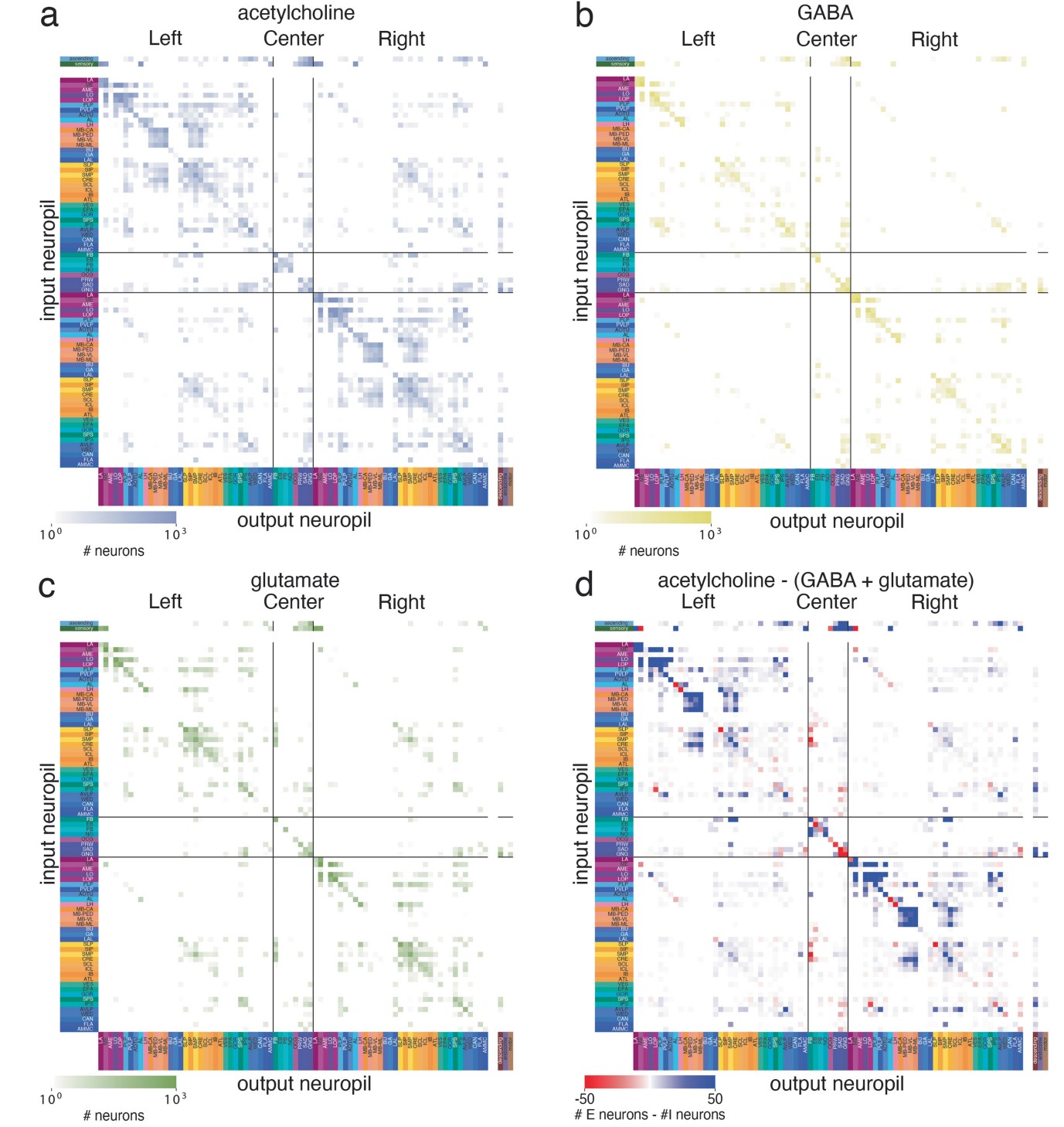

**Extended Data Fig. 5 | Neuropil-neuropil projection maps.** (a) Projection maps produced as in Fig. 4a limited to connections from cholinergic, (b) GABAergic, and (c) glutamatergic neurons. (d) The difference between the putative excitatory (acetylcholine) and the putative inhibitory (GABA, glutamate) projection maps.

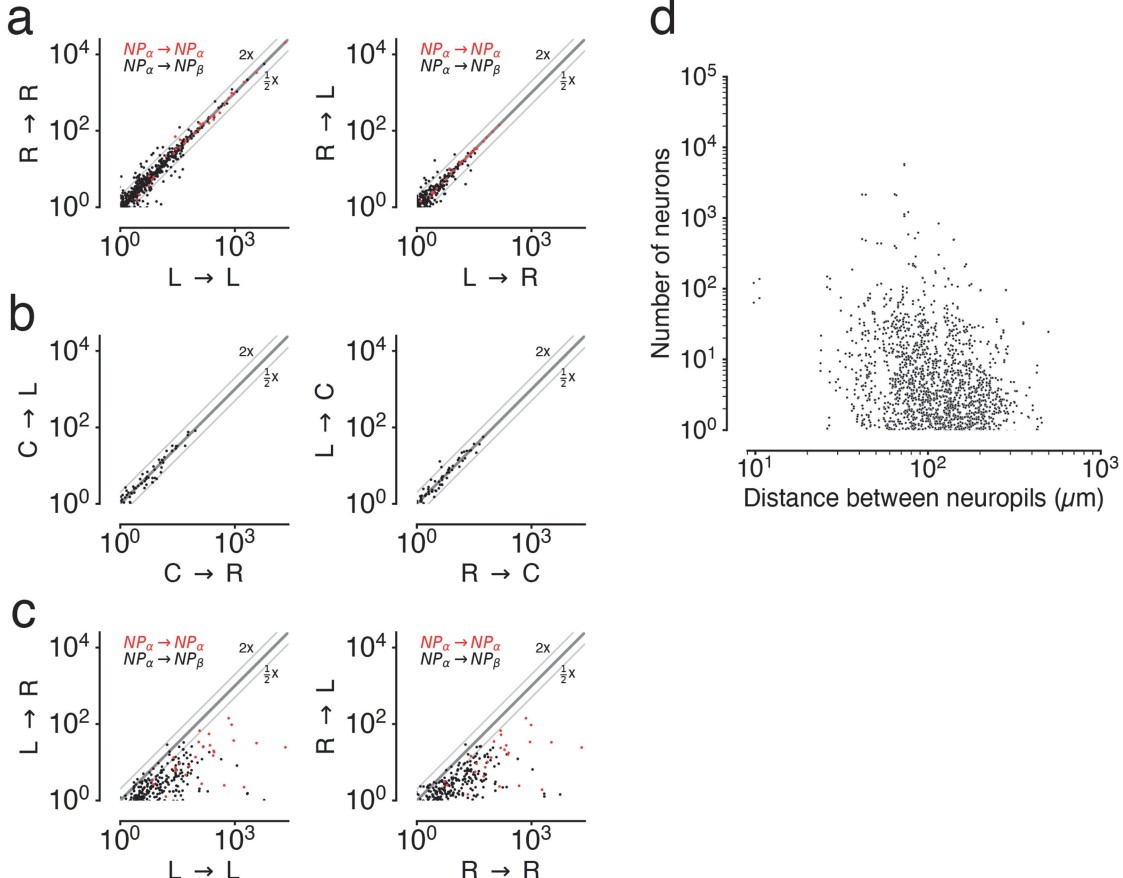

**Extended Data Fig. 6 | Neuropil-neuropil projections compared between hemispheres.** Each dot is a neuropil-neuropil projection in one hemisphere and the axes show the fractional weights as calculated in Fig. 4a,b. Red dots are comparisons between the same neuropils in different hemispheres (e.g. AMMC(L) -> VLP(L) vs AMMC(R) -> VLP(R). (a) Comparison of projections between neuropils in both hemispheres and between hemispheres.

(b) Comparisons of projections with the center neuropils. (c) Comparisons of projections between ipsilateral and contralateral neuropil projections. (d) Comparisons of the distances between neuropil centroids with the fractional neuron weights. Connections within neuropils were excluded and neuropil pairs connected with <1 fractional neuron weight are not shown.

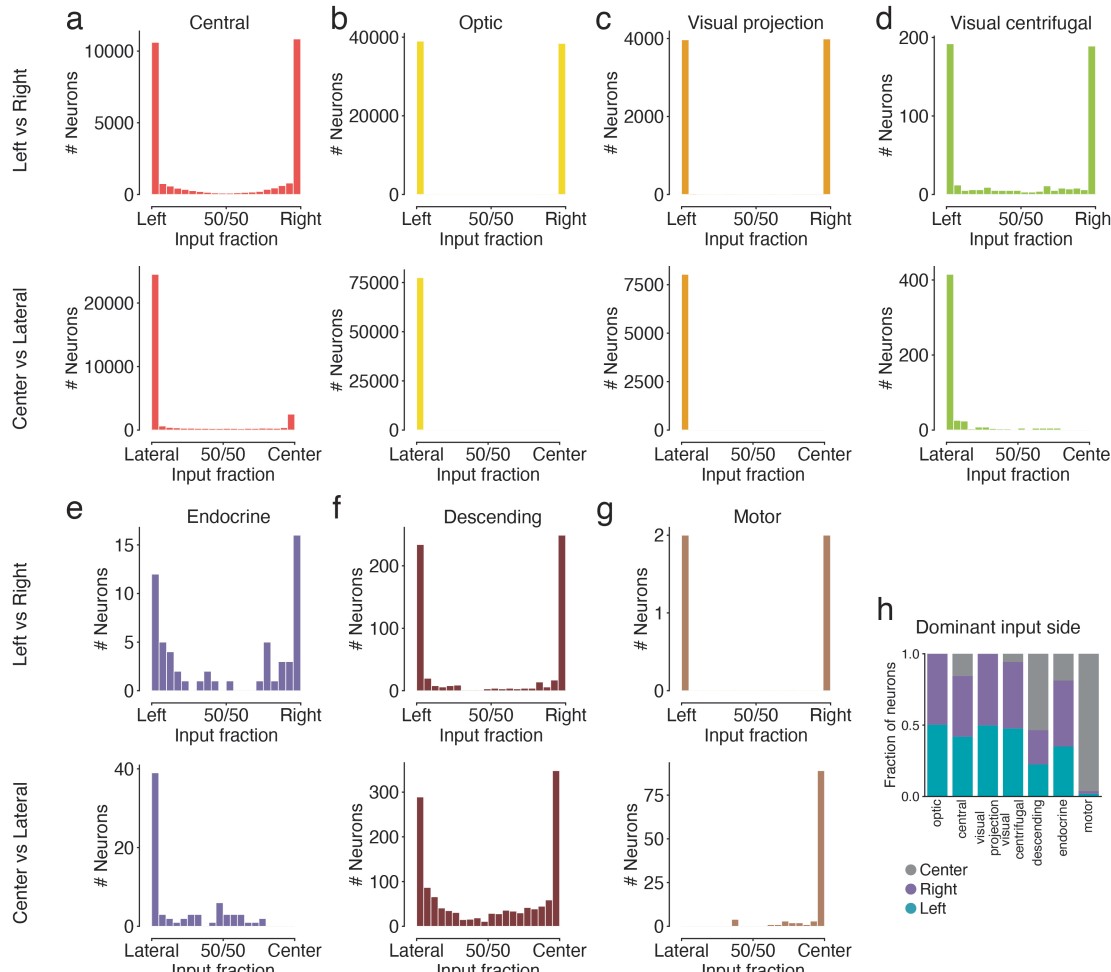

**Extended Data Fig. 7 | Input side analysis.** We assigned postsynaptic locations to either the center region or the left or right hemisphere. (a-g) For each superclass, (top plot). The lower plot shows the fraction of synapses in the center vs the lateral regions for all neurons. (h) Each neuron was assigned to the side where it received most of its inputs.

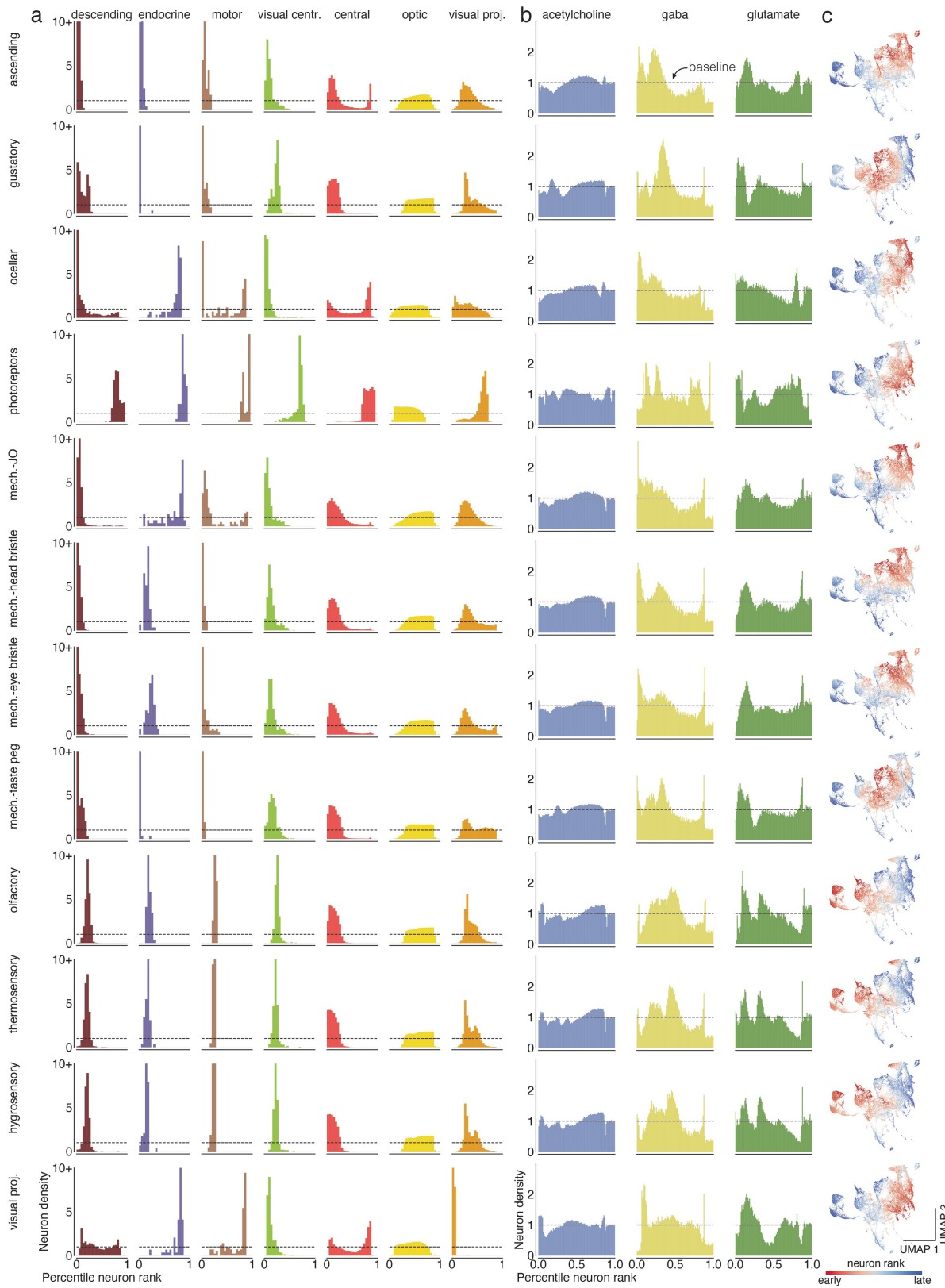

**Extended Data Fig. 8 | Percentile ranks for every modality.** (a) For each sensory modality (rows) we used the traversal distances to establish a neuron ranking. Each panel shows the distributions of neurons of each super-class within the sensory modality specific rankings. (b) Same as in (a) for the fast neurotransmitters. (c) Neurons in the central brain shown in the UMAP plot are colored by the rank order in which they are reached from a given seed neuron set. Red neurons are reached earlier than blue neurons.

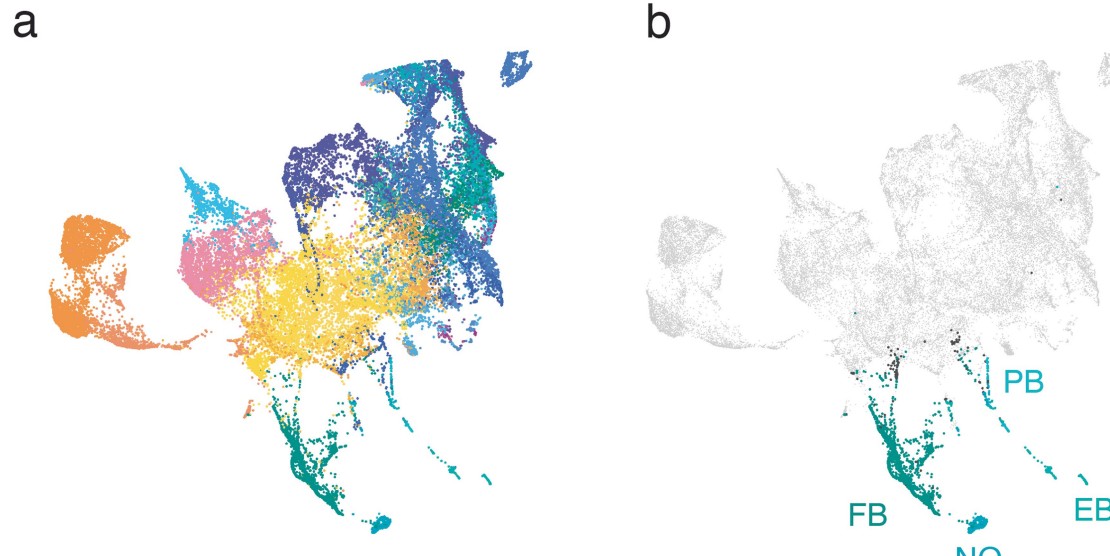

a

b

PB

FB

EB

NO

**Extended Data Fig. 9 | Rank-based UMAP projection and neuropils.** (a) Every neuron in the central brain was assigned to the neuropil where it received the most synapses. Every dot is then colored by the assigned neuropil (see Extended Data Fig. 1 for neuropil colormap). (b) Same as in a but limited to the central complex neurons. Neurons in the central complex with an assigned neuropil other than the ones shown are colored black.

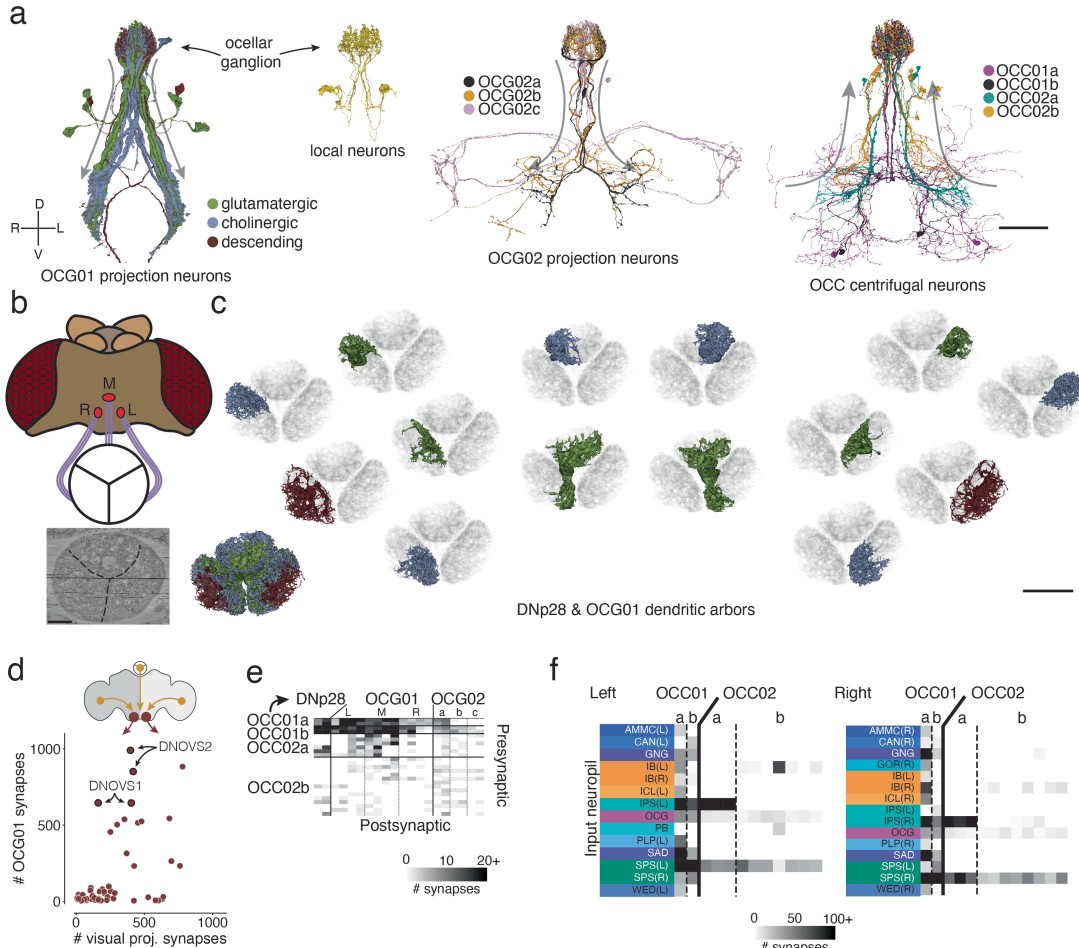

**Extended Data Fig. 10 | Ocellar circuit.** (a) Renderings of all neurons (excluding the photoreceptors) with arbors in the ocellar ganglion. "Information flow" from pre- and postsynapses is indicated by arrows along the arbors. (b) Overview of the three ocelli (left, medial, right) which are positioned on the top of the head. Photoreceptors from each ocellus project to a specific subregion of the ocellar ganglion which are separated by glia (marked with black lines on the EM). (c) Top view of the dendritic arbors within the ocellar ganglion of each DNp28 (brown) and OCG01 (blue: cholinergic, green: glutamatergic). The render on the lower shows all 12 OCG01s and 2 DNp28s. Each other render shows one neuron in color and all others in the background in gray for reference. (d) Comparison of number of synapses from OCG01 neurons and visual projection neurons onto descending neurons. (e) Connectivity matrix for connections between ocellar centrifugal neurons and ocellar projection neurons. (f) Inputs to ocellar centrifugal neurons by neuropil. Scale bars: 100 μm (a), 20 μm (c).

# Reporting Summary

## Statistics

For all statistical analyses, confirm that the following items are present in the figure legend, table legend, main text, or Methods section.

| n/a | Confirmed | |
|---|---|---|
| ☐ | ☒ | The exact sample size (*n*) for each experimental group/condition, given as a discrete number and unit of measurement |
| ☒ | ☐ | A statement on whether measurements were taken from distinct samples or whether the same sample was measured repeatedly |
| ☐ | ☐ | The statistical test(s) used AND whether they are one- or two-sided<br>*Only common tests should be described solely by name; describe more complex techniques in the Methods section.* |
| ☒ | ☐ | A description of all covariates tested |
| ☒ | ☐ | A description of any assumptions or corrections, such as tests of normality and adjustment for multiple comparisons |
| ☒ | ☐ | A full description of the statistical parameters including central tendency (e.g. means) or other basic estimates (e.g. regression coefficient) AND variation (e.g. standard deviation) or associated estimates of uncertainty (e.g. confidence intervals) |
| ☒ | ☐ | For null hypothesis testing, the test statistic (e.g. *F*, *t*, *r*) with confidence intervals, effect sizes, degrees of freedom and *P* value noted<br>*Give P values as exact values whenever suitable.* |
| ☒ | ☐ | For Bayesian analysis, information on the choice of priors and Markov chain Monte Carlo settings |
| ☒ | ☐ | For hierarchical and complex designs, identification of the appropriate level for tests and full reporting of outcomes |
| ☐ | ☒ | Estimates of effect sizes (e.g. Cohen's *d*, Pearson's *r*), indicating how they were calculated |

*Our web collection on statistics for biologists contains articles on many of the points above.*

## Software and code

Policy information about availability of computer code

| Data collection | https://github.com/CAVEconnectome |
|---|---|
| Data analysis | https://github.com/seung-lab/FlyConnectome |

For manuscripts utilizing custom algorithms or software that are central to the research but not yet described in published literature, software must be made available to editors and reviewers. We strongly encourage code deposition in a community repository (e.g. GitHub). See the Nature Portfolio guidelines for submitting code & software for further information.

## Data

Policy information about availability of data

All manuscripts must include a data availability statement. This statement should provide the following information, where applicable:
- Accession codes, unique identifiers, or web links for publicly available datasets
- A description of any restrictions on data availability
- For clinical datasets or third party data, please ensure that the statement adheres to our policy

All data is made publicly available. Codex (codex.flywire.ai), braincircuits.io, and Catmaid spaces (fafb-flywire.catmaid.org) facilitate non-programmatic access. Most of the data can be directly download from codex (codex.flywire.ai/api/download). All data, including the volumetric data and meshes, can be programmatically accessed through CAVE and cloudvolume. We provide tutorials for programmatic access here: https://github.com/seung-lab/FlyConnectome

# Research involving human participants, their data, or biological material

Policy information about studies with [human participants or human data](). See also policy information about [sex, gender (identity/presentation), and sexual orientation]() and [race, ethnicity and racism]().

| | |
|---|---|
| Reporting on sex and gender | *Use the terms sex (biological attribute) and gender (shaped by social and cultural circumstances) carefully in order to avoid confusing both terms. Indicate if findings apply to only one sex or gender; describe whether sex and gender were considered in study design; whether sex and/or gender was determined based on self-reporting or assigned and methods used.*<br>*Provide in the source data disaggregated sex and gender data, where this information has been collected, and if consent has been obtained for sharing of individual-level data; provide overall numbers in this Reporting Summary. Please state if this information has not been collected.*<br>*Report sex- and gender-based analyses where performed, justify reasons for lack of sex- and gender-based analysis.* |
| Reporting on race, ethnicity, or other socially relevant groupings | *Please specify the socially constructed or socially relevant categorization variable(s) used in your manuscript and explain why they were used. Please note that such variables should not be used as proxies for other socially constructed/relevant variables (for example, race or ethnicity should not be used as a proxy for socioeconomic status).*<br>*Provide clear definitions of the relevant terms used, how they were provided (by the participants/respondents, the researchers, or third parties), and the method(s) used to classify people into the different categories (e.g. self-report, census or administrative data, social media data, etc.)*<br>*Please provide details about how you controlled for confounding variables in your analyses.* |
| Population characteristics | *Describe the covariate-relevant population characteristics of the human research participants (e.g. age, genotypic information, past and current diagnosis and treatment categories). If you filled out the behavioural & social sciences study design questions and have nothing to add here, write "See above."* |
| Recruitment | *Describe how participants were recruited. Outline any potential self-selection bias or other biases that may be present and how these are likely to impact results.* |
| Ethics oversight | *Identify the organization(s) that approved the study protocol.* |

Note that full information on the approval of the study protocol must also be provided in the manuscript.

# Field-specific reporting

Please select the one below that is the best fit for your research. If you are not sure, read the appropriate sections before making your selection.

☒ Life sciences ☐ Behavioural & social sciences ☐ Ecological, evolutionary & environmental sciences

For a reference copy of the document with all sections, see [nature.com/documents/nr-reporting-summary-flat.pdf]()

# Life sciences study design

All studies must disclose on these points even when the disclosure is negative.

| | |
|---|---|
| Sample size | This manuscript is based on one image dataset from one fly brain |
| Data exclusions | No data was excluded |
| Replication | We did not replicate this study on other animals |
| Randomization | N/A |
| Blinding | N/A |

# Reporting for specific materials, systems and methods

We require information from authors about some types of materials, experimental systems and methods used in many studies. Here, indicate whether each material, system or method listed is relevant to your study. If you are not sure if a list item applies to your research, read the appropriate section before selecting a response.

## Materials & experimental systems

| n/a | Involved in the study |
|-----|----------------------|
| ☒ | ☐ Antibodies |
| ☒ | ☐ Eukaryotic cell lines |
| ☒ | ☐ Palaeontology and archaeology |
| ☐ | ☒ Animals and other organisms |
| ☒ | ☐ Clinical data |
| ☒ | ☐ Dual use research of concern |
| ☒ | ☐ Plants |

## Methods

| n/a | Involved in the study |
|-----|----------------------|
| ☒ | ☐ ChIP-seq |
| ☒ | ☐ Flow cytometry |
| ☒ | ☐ MRI-based neuroimaging |

## Animals and other research organisms

Policy information about studies involving animals; ARRIVE guidelines recommended for reporting animal research, and Sex and Gender in Research

| | |
|---|---|
| Laboratory animals | 7-day-old [iso] w1118 x [iso] Canton-S G1 adult female flies |
| Wild animals | N/A |
| Reporting on sex | Female |
| Field-collected samples | N/A |
| Ethics oversight | N/A. Sample was collected by prior study |

Note that full information on the approval of the study protocol must also be provided in the manuscript.

## Plants

| | |
|---|---|
| Seed stocks | *Report on the source of all seed stocks or other plant material used. If applicable, state the seed stock centre and catalogue number. If plant specimens were collected from the field, describe the collection location, date and sampling procedures.* |
| Novel plant genotypes | *Describe the methods by which all novel plant genotypes were produced. This includes those generated by transgenic approaches, gene editing, chemical/radiation-based mutagenesis and hybridization. For transgenic lines, describe the transformation method, the number of independent lines analyzed and the generation upon which experiments were performed. For gene-edited lines, describe the editor used, the endogenous sequence targeted for editing, the targeting guide RNA sequence (if applicable) and how the editor was applied.* |
| Authentication | *Describe any authentication procedures for each seed stock used or novel genotype generated. Describe any experiments used to assess the effect of a mutation and, where applicable, how potential secondary effects (e.g. second site T-DNA insertions, mosiacism, off-target gene editing) were examined.* |

