## [Peer Review File · Nature]

Manuscript Title: Neuronal wiring diagram of an adult brain

Reviewer Comments & Author Rebuttals

Reviewer Reports on the Initial Version:

Referees' comments:

Referee #1 (Remarks to the Author):

Dorkenwald et al report the complete connectome of the whole adult fruit fly brain. This is a landmark achievement and this is a landmark paper. It is of course challenging to summarize both all the methodology leading to this accomplishment and the many discoveries resulting from this connectome mapping study in the format of one (or two) Nature papers. Indeed, there are many (under preparation) companion papers cited in this paper. Therefore, the goal of this paper appears to be to provide a road map to the accomplishments and discoveries of this multi-disciplinary team of scientists. I will review this paper from that perspective — how well does this manuscript summarize the work and provide a useful introduction to both experts in fruit fly systems neuroscience and to the broader community of interested neuroscientists, biologists, computer scientists, and engineers.

In general, the paper is well written and covers a lot of ground in a clear and concise manner. I'm very enthusiastic about this paper. Here are my major comments:

Major comment 1

Flywire used existing FAFB volume and computational tools (both prev published by others and custom developed for flywire) to segment neuron, detect pre/post synapses, predict neurotransmitter type, attach synapse to neuron skeletons, and cell type assignment. However it is unclear to what extent the "connectome" construction is complete, and which aspects are automated due to flywire development and which aspects require more human intervention to achieve completion. It would be helpful to see a flow chart diagram to show what resources flywire developed/invested effort (in terms of person years as well as dollars) in to go from a sample brain to EM volume to a "connectome", along with amount of work left that could be automated vs require further human effort. For example, a majority of neuropils have <50% of post-synapses attached to a proofread segment (Ext Fig 1-3). while the lower completion rates is not unexpected, it would be useful to comment on how to deal with this in interpretation or how to treat synapse counts.

Major comment 2

This type of project requires concerted effort across many people and academic groups. While it is understandable to not rehash detailed analysis/findings by other manuscripts in prep, this

manuscript contains more than 10 "in prep" citations that truncated any meaningful/deeper discussion of the data/result presented in this manuscript.

For example:

CAVE, the proofreading tool to assign synapses to neuron segment, is an "in prep" citation. given one of the major contribution of the paper is to assign synapse to neurons for creating a wiring diagram, it would be helpful to at least summarize what technical advances were made that allowed the flywire community to generate a "complete" diagram (and distinct from prior art). Is this something that can translate to new EM volumes? to different organisms (e.g. mammalian brains).

There are numerous other instances where the authors simply refers to "in prep" and not discuss more:

gustation, mechanosensation analysis using flywire diagram Shiu et al. 40, Eichler et al. 41
sexually-dimorphic neurons, deutch et al in prep
optic lobe and central brain, kind, garner et al in prep
whole brain network analysis, Lin et al in prep
does not go into any info on optic lobe, but refers to Heckman and Clowney in prep
some VCN are peptidergic for circadian rhythm, Reinhard, Fukuda et al in prep
tracheal system analysis? Colodner et al in prep
hue selectivity, Christenson et al in prep

If the purpose is to highlight how much utility the flywire resource is able to provide, may be better to summarize these succinctly together. is it primarily the completeness of the diagram that enables new science? if so, highlighting these researches together in one section would make it more obvious that it is worth having a "complete" connectome.

Major comment 3

The network analysis provided here are "neuropil projections" and "flows" (fig 2b, 4, 6) and some statistics on the connectome graph (in vs out node degrees, etc). and a connectivity analysis is provided for ocellar circuit (fig 7). The authors generally analyzed organization based on hemisphere or neuropil differences, which do not require cell typing. The authors avoided doing central brain analysis since that's covered by the companion manuscript. But given optic lobe neurons account for 73655 out of 114423 intrinsic neurons, there is surprisingly no analysis provided for optic lobe neurons in fig 5. If this is due to another "in prep" paper, consider incorporating the optic lobe anatomy into an earlier figure if the purpose is to demonstrate completeness.

In general, fig 5 contains little in the way of concrete "results" and could be improved by adding analysis of the optic lobe. For instance, if the goal of fig 5b is to show the "true tiling", it would be nice to analyze the nature of the lattice or compute inter neuron distance/angle distributions, etc. It would be nice to add additional high level connectomic analyses of the optic lobe.

Disappointingly, very little new information is provided in the "Optic lobes: columns and beyond" section. Specific celltypes like Mi1, Dm12, LPi1-2 are singled out for comment here, but the

discussion do not necessarily reveal new information concerning these cell types.

Minor comments

1. The definition of segmentation in the nomenclature table at the start of the paper could be improved. "segmentation: product of automated pipeline", does not convey much useful information. Other definitions could also be written more clearly.
2. A synapse thresholds of 5 and 10 synapses are variously used in Dorkenwald et al and Schlegel et al. Would be nice to be consistent? You are likely establishing conventions for the field?
3. Can you comment on the neurotransmitter type predicted for the visual centrifugal neurons and whether they play a neuromodulatory role? An example is given of OA-AL2b2 but it would be nice to hear if the other VCNs are also octopaminergic,
4. Would be nice to have a table of neuropil and neuron/cell type acronyms to full names along with the hierarchy. Did I miss it?
5. I must confess that I don't find the information flow section and fig 6 to be super enlightening. Fig 6b is nice to see as an analysis of how gustatory information potentially propagates through the brain. But I don't really know how to make sense of the rest of the figure and analysis (including the UMAP analysis). Is the UMAP analysis the only way to discover that there are two classes of KCs targeted by ORNs and VPNs? This is not a condemnation — there might be other scientists who might find this interesting (I hope?).
6. The ocellar circuit is very exciting to see, but the figure and description is hard to follow. I like the circuit diagram in fig 7f, but that is only a caricature. The real data are shown in fig 7(c,d) but we only see the morphologies of the neurons but not the circuitry. It would be nice to see a schematic of the circuit in the form of a graph with nodes and edges.
7. In Fig 4b, would be nice to show the equation next to the graphic representation.
8. In Fig 4, the SEZ is highlighted since it's not present for the most part in the hemibrain. Would be nice to do the same analysis for all the other regions in supplementary, or provide Jupyter notebooks or links to how such analysis could be performed for each neuropil.
9. Fig 4g,h not obvious the color code is matched for ipsilateral vs bilateral vs contra
10. Given how rare contralateral and bilateral neurons are in general, consider expanding the plot to also show these by fig 4h "super class"
11. Ext Fig 1-2. What is the color code for the dots?

12. what's the lower fraction of attached synapses relative to hemibrain for specific neuropils due to? need more proofreading for correction?

13. was there a technical computational advance that flywire made (CAVE in prep??) that improved/could improve attachment rate?

14. In fig 2b and many other places throughout, the fly brain underneath can be difficult to see, consider making it more visible? Especially hard to see when printed.

Referee #2 (Remarks to the Author):

In this paper, Dorkenwald and colleagues present a new data set consisting of the connectome of an entire brain of an adult *Drosophila melanogaster*. This sophisticated dataset will be of particular interest to those seeking to understand circuits in the fly's brain, while it is simultaneously a leap forward in understanding the connectivity and structure of brains more generally. In particular, this connectome is a significant advance over the prior adult *Drosophila* connectome, the 'hemibrain', published in 2020, because this one encompasses the entire brain, including optic lobes, direct sensory inputs, and descending and ascending connections between the brain and the ventral nerve cord. This enables far more wholistic analyses of the fly's brain, from multisensory integration to tracing how visual signals enter the central brain. The tracing of ocelli signals is presented as an interesting example of what can be done and this yields real, testable predictions about fly brain function.

This paper is written as the document of record supporting the first release of this data, which will evolve over time with more community and specialist inputs. Overall, this is well-done, but I think there could be a few improvements, the most important of which I describe below.

1) I think this manuscript could do with a strong round of editing. Below, I collected some minor and major editing issues that I noted while reading the paper.

a) In the first paragraph of the introduction: The second sentence felt rather unconnected to the first. This whole paragraph could flow better. (Side note: reviewers always appreciate line numbers.)

b) Later, the paragraph beginning "We already showed..." should have a topic sentence that's about the several ways the authors have validated this dataset, which are then enumerated in this paragraph.

c) Later, "but no missing connections" seems rather ambiguous to me as a dependent clause. Do the authors mean avoiding FP but not missing connections, or "while not missing connections". I think authors mean latter.

d) "the additional partners are from the same or different cell types" – unclear what is meant by "additional" here, or what this sentence really means.

e) The discussion felt very disjointed, like a hodge podge of small, disconnected points.

f) Also in the discussion, I'm not sure about the Brain simulation points. Some serious caveats about the approach are likely in order for non-specialists. We're not just going to suddenly successfully simulate the fly brain by using this data set. Citation 115 is less a simulation than a model optimized to perform a task, where the connectome acted as a strong constraint on model architecture. This

part of the discussion could be clarified a bit. As written, it seems a little Polyanna-ish.

2) I found the analyses surrounding Figure 6 rather confusing, and was not sure at the end of it what I had learned, or how the field might make use of these descriptions of the connectivity in the connectome. Does the last panel show that different sensory inputs project to different brain regions, for the most part? In F6c, if one organizes neurons by quantiles, one expects a uniform distribution, but no combination of the subsets of neurons in this panel seem to add up to a uniform distribution – a bunch of neurons must not be accounted for, but I couldn't figure out which. (This includes staring at the supp fig for a while, too.)

3) Some more discussion seems due on the limitations of this study and comparison to other ones.

a) One major difference between this dataset and the hemibrain dataset is that the hemibrain acquired voxels with 8x8x8 nm volumes, while here the volumes are 4x4x40 nm. This must have pros and cons, and lead to differing levels of confidence in some segmentation and identification processes. I think this would be valuable to comment on briefly in the main text, but perhaps in more detail in the methods.

b) Many of the methods, like inferring a neuron's neurotransmitter, rely on methods published elsewhere. Here, it's unreasonable to fully recount these other methods in this paper. But it would be exceedingly helpful to readers to present short summaries of these other methods in the methods section, and perhaps highlight uncertainties, error-rates, or limitations in the different methodologies used.

4) Finally, probably the most important issue: the value of this dataset is really in its accessibility and the tools that allow the community to use it. The hemibrain dataset has been so influential in part because it provided excellent tools to access and analyze the data. As part of this review, I asked a 4th year PhD student in lab to use the Flywire data and tools for a small project in lab, during which he spent ~2 full days accessing the data. I would add that he is a sophisticated user of the hemibrain dataset via their API, but is not a power user of the type who is familiar with catmaid, for instance. Below, we present several issues he brought up that could improve the accessibility to data, the data itself, and documentation of the code. If he spent 1 hour trying to figure something out, if it can be fixed to be easier, it could save 1000s of hours collectively in the community!

Data accessibility (comments from the graduate student)

1) The paper should explicitly state what functions users should use CAVE, fabseg, and NAVis for. For instance, in figs 1b and 1c, the authors could specify which tool can be used to grab each data type. It was time consuming to try to figure out what should be used for connectivity, what should be used for voxels, what should be used for neuron morphology, etc. A table could also help.

2) How can users get voxel level data? Using NAVis? Two days' worth of poking around didn't reveal how to do this. The NAVis tutorial that was most obvious for this ("Voxel / Image Data") seemed to do something else.

3) The links were easy to follow.

4) The flywire Youtube channel was really helpful and could perhaps be linked prominently in the paper or on the website. Similarly with the Slack channel.

5) On the homepage of "flywire.ai", it says "For programmatic analysis, see natverse (R) and CAVE

(Python).” However, fabbseg and NAVis should also be included in this sentence. Also, fabbseg was not included in Figure 1C.

6) The data seemed to be already stored in google cloud buckets – the PhD student would have found it useful if the data was accessible via tensorstore.

7) If neuroglancer was used to generate figures in this paper, it would be awesome to include a supplemental table with the URLs for each figure, so that people can look up exactly what’s in them!

8) For visualizing neurons, it would be helpful to have an easy way to get a set of A-P and D-V and L-R (or medial-lateral) axes on any plot. It is easy to become disoriented.

Data content (from the graduate student)

1) There did not seem to be a clear way for people to edit neurotransmitters. There should be an option for users to input empirically determined neurotransmitters for neurons, as well as the paper(s) supporting the determination. In that case, each neuron type should have a predicted and empirical neurotransmitter (when available) attached to it.

2) The fetched data comes with fields such as “pre_x, pre_y, pre_z, tag, type, hemilineage, confidence, changed, superceded_id, pt_root_id, pt_supervoxel_id, id, user_id,” etc. On the flywire website and relevant github repositories, there should be a table explaining what each datatype means. Many of these fields were pretty opaque and documentation would be immensely helpful.

3) In the output of “fetch_synapses” in fabbseg, one of the columns is “id”. Is this id unique for each presynapse density? It was clear from the paper that the presynapse-postsynapse pair creates multiple presynapse coordinates for the same presynaptic site, but it wasn’t clear how users know which presynaptic coordinates to combine to get all postsynaptic connections coming from one presynapse. In the hemibrain dataset there is a single presynaptic site, but in Flywire it looks as though each synapse lists both pre and post site. This makes it hard to tell how polysynapses are arranged.

4) It would be very helpful to have a consensus name for the neuron type associated with each neuron. Right now, this is in the annotations field and depends on the user who annotated. For instance, TmY17 has the following annotations

- a. Transmedullary Y neuron 17, TmY17 (A. Nern naming system)
- b. Transmedullary Y neuron 17, TmY17, (A.Nern, personal communication)
- c. Transmedullary Y neuron 17, TmY17, TmY17_R (A.Nern, personal communication)
- d. Transmedullary Y neuron 17, TmY17,(A.Nern, personal communication)

If someone searched for TmY17_R, they would miss a bunch that did not have the _R appended. Similarly, a search in this field for L4 neurons yields every single reference to Gal4 drivers in the annotations. It would be helpful to have a single field that contains cell type name information, which is exceedingly useful in querying the hemibrain dataset.

Code functions and accessibility (also from this user)

1) The README file in the CAVEclient github repository is currently only one sentence. It should at least include how to install CAVEclient, and include the link to the readthedocs.io documentation.

2) It would be very useful to have a function that provides the neuropil associated with a given coordinate (function argument). In whichever library is best here.

3) Critical: The documentation in fabbseg assumes the user knows the ID of the neuron they are

investigating, but it is more likely the user knows the neuron type before the ID. Therefore, it would be very useful to have a function that grabs the neuron IDs of all neurons in the same hemilineage, VFB ID, flow, superclass, class, cell type, neurotransmitter, and/or annotation. In analogy with neuprint-python, these could be inputs to the NeuronCriteria function. So the function in neuprint-python lingo would be `fetch_neurons(NC(class="MBON", neurotransmitter="dopamine"))`. The current arrangement might be a historical artifact, since neuron IDs were known before types; but now that all is known, it would be super-useful (the most useful!) functionality to have. If it already exists, it was not obvious to this user.

4) Similarly, the `fetch_synapses` function should allow users to specify the neuron type of the presynapse and postsynapse, not just the ID.

5) `Fetch_synapses` should have a default argument `use_new_id = True`, where the output ids in the "pre" and "post" columns are the most up to date ids, so that users don't have to constantly use the `update_ids` function. This created a lot of confusion for us, and could lead to bugs.

6) Critical: More coding examples would be important to give out. For instance, one example could be a researcher asking "I want to know the inputs and outputs adjacency matrix of all neurons in this type". Almost all users will use this dataset for connectivity purposes, so having this tutorial/example and similar ones would really allow most users to get a running start on their analyses.

7) In the NAVis tutorials, the "Voxel/Image Data" in the "Import/Export" section should include how to grab the grayscale and neuron segmentation, given a list of coordinates or a bounding box. Additionally, the tutorial could include how to show a cross section.

More minor issues:

1) "any normal *Drosophila* individual" should read "female", "wild type" instead of normal, and presumably *melanogaster* in addition the genus. Authors should also note the fly's age somewhere in the paper – I could not find that anywhere.

2) "average F1-score of 99.2% by volume". This phrasing contains jargon, pretty impenetrable to the reader. I am guessing that this is good, but maybe just say so in a way that leaves out the jargon. I could not find "F1-score" defined anywhere in this manuscript.

3) Would be nice to also report synapse density as number of synapses per neurite path length; the current reported value seems to be Euclidian distance.

4) Dendrite-soma-axon paradigm from mammals "may not apply" should read "does not apply". It can't with these cell morphologies, right?

5) Figure 3d – ylabel "cumulative ratio" should read "cumulative fraction", right? If not, please specify what the ratio is.

6) Figure 3f – labeled "exp fit" but caption says it's a truncated power law distribution; "exp fit" implies "exponential fit". A straight untruncated power law seems like it would be a good fit to this distribution and it would be useful to know the exponent... Perhaps the prior analysis was missing these rare connections with many synapses. Also, where the exponential cutoff is mentioned in text, it cites 3g, but should be 3f.

7) Figures 3g and 3h – unity lines would be useful to show here, or fit lines as well as the Pearson correlation coefficients given in caption.

8) Cubic microns seem like more natural unit for this dataset than mm³ (which are tiny numbers), and the manuscript kind of goes back and forth. Neither is an SI unit, so I'm not sure there's a good reason to prefer either.

9) "CT1 arborizes exclusively to medulla". No, I believe CT1 arborizes in lobula, too, making synapses onto every T5 dendrite there. CT1 is likely GABAergic, but it appears to be playing a far more local role, different from the one described in the text of "local feedback gain control". Its local response properties influence local motion detection (Meier & Borst 2019, Gonzalez-Suarez et al. 2022). The authors mention the compartmentalization later, but it's weird to lump it in here with feedback gain control neurons, like APL in the olfactory system, when that is not at all its known function.

10) Figure 5f – Could you choose colors with stronger contrast than green and blue for the two LPI neurons? Also, is this phenomenon true of other LPI neurons?

11) Figure 7a. There appears to be an error with the L/R in the diagram? If we're looking down on the head, these must be flipped, no? This is not related to the left-right flip in the volume, right?

12) "caps out at an input fraction of 30%". This feels like lab slang, and it is also not clear in the text if this cap is the method or a description of the data. The figure shows that this is part of the method; how was 30% chosen for the saturation?

13) I think that in the optic lobe results section, the authors could highlight even more that the hemibrain left out enormous chunk of the peripheral visual system, which is importantly included here! In this section, the authors could mention subsequent reconstructions of other bits of the optic lobe, not connected to the rest of the brain (Shinomiya et al. 2022).

14) It's fantastic that there are all these companion papers in the works, but this paper should also stand alone, which it does. I'm just not quite sure what balance to strike here, in terms of mentioning all these companion papers.

Referee #3 (Remarks to the Author):

This paper by Dorkenwald et al. describes a landmark achievement in neuroscience: the first neuronal wiring diagram of a complete adult brain. The complexity and completeness of this wiring diagram, reconstructed from an adult *Drosophila* female, represent a significant advance over previous connectomes (e.g. *C. elegans*, larval *Drosophila*, or the *Drosophila* hemibrain) and enable new studies that were not previously possible. In particular, unlike the fly hemibrain, this connectome includes both hemispheres and all afferent and efferent neurons. This connectome has already begun to revolutionize the study of neural circuits, as evidenced by recent papers and preprints using this dataset.

Aside from describing this accomplishment, this paper also includes many interesting analyses regarding the major types and features of brain neurons, principles of connectivity at the level of cells and brain regions, the relationship between connectivity and neurotransmitter identity, and how information from different sources flows across the brain. These findings reveal new insight into brain organization and function, and basic metrics described in the paper will likely provide references for the field for years to come. The authors have done a commendable job of visualizing large datasets and choosing appropriate metrics for gaining insight into the data. In my opinion, there is no question that this paper warrants publication in a venue such as Nature where it will reach a wide-ranging audience.

I have no major issues with the authors' analyses or conclusions, only comments aimed at improving the paper.

General comments:

1. For major metrics of connectivity, it would be nice to explicitly compare to the hemibrain data if possible. For example: number of synapses per neuron, the distribution of the number of synapses per connection, median in-degree and out-degree.
2. The projectome (Fig. 4A) is a very interesting dataset, and I think a bit more could be said about the overall connectivity patterns. Which regions are most highly connected? Do highly connected regions tend to be in closer physical proximity? Which are the few regions that show the strongest left-right connectivity?
3. The Discussion feels a bit choppy. I would suggest combining some of the very short sections into longer and more coherent sections, for example focusing on 1) methodology, 2) implications of the connectome, and 3) future directions. While this is the authors' prerogative, I also noticed that the Discussion does not include much interpretation of the scientific results (e.g. connectivity patterns, differences in excitatory vs inhibitory neuron connectivity, information flow).

Specific/minor comments:

Introduction: It sounds like the authors may be drawing a distinction between "wiring diagram" and "connectome", but this was not clearly defined.

p. 5, "third validation method": Is this really a "validation method" in terms of demonstrating the accuracy of predicted synapses? It's certainly important to know how many synapses are attached to orphan segments versus proofread neurons, but I'm not sure it validates the methodology, especially as over half of postsynapses are located in the orphan twigs. This seems like a high number.

p. 7: It would be good to say something brief about how the sensory organs for non-visual sensory neurons were identified – is this based on matching to known neuronal morphologies?

p. 8: I couldn't parse the second half of this sentence: "The optic lobes, which are largely absent from the 1st instar larval, are a major reason that the adult fly brain so dominates its nervous system."

p. 10: In addition to quantifying total volume of each neuron, I wonder if you also quantified cell body volume? Do cell body volumes also vary across orders of magnitude, and do they correlate with path length, cell type, neurotransmitter identity, or other features?

p. 12: "We asked whether large neurons tend to use their many synapses to create stronger connections with individual neurons versus more connections with many different neurons. The total number of synapses established by a neuron was much better correlated with its in and out degrees ($R=0.93$, $R=0.93$ respectively) than its average connection strength." The fact that this supports the second model is not obvious and could be explicitly stated.

p. 13: "GABAergic and glutamatergic neurons had much higher degrees than cholinergic neurons" and "GABAergic neurons were on average longer". I don't think either of these results is obvious from the cited figure panels (extended Fig 3-1), although I appreciate seeing the full distributions. The average or median numbers could be stated explicitly.

p. 19 and Fig. 6B: It may be worth mentioning that the low-rank neurons in the gustatory pathway (rank 1 and 2) look strikingly similar to second-order gustatory neurons identified experimentally (e.g. in Talay et al., Snell et al., Deere et al.), which corroborates the validity of traversal method.

p. 19 and Fig. 6D: The claim that inhibitory neurons are overrepresented in the early layers would be more convincing if excitatory neurons were plotted on the same graphs for comparison (even though they're in the extended data). Also, could you be more specific about what you mean by "we identified a sequence of GABAergic and glutamatergic peaks..."?

Fig. 2C: I thought "optic" neurons are defined as being contained completely within the optic lobes, but it looks like some neurons have innervations in the central brain. Are these all tracts without any synapses?

Fig. 2D: Some of these abbreviations don't seem to be defined anywhere. And I assume the numbers refer to how many neurons are contained within each nerve, but this isn't explicitly stated.

Fig. 2F: I am not sure I can distinguish the two colors of green corresponding to gustatory vs. hygro-sensory neurons.

Fig. 3F: The font seems to be inconsistent with the rest of the figure.

Fig. 4 and elsewhere: The neuropil abbreviations are not defined; please provide a table or cite a reference where the reader can find the full names.

Fig. 4C-D: I think it needs to be stated again in the legend that the left and right sides are flipped in the images. Additionally stating this somewhere in the main text, not just legends and methods, may

also be a good idea.

Fig. 4E: I suggest explicitly stating that the insets are the same data zoomed in on specific x values.

Fig. 7: It would be helpful to include a diagram showing where the ocellar ganglion is located relative to the rest of the brain.

Fig. 7E: "glutamatergic" is misspelled on the y-axis

Ext Fig 1-1: It's not clear to me where the symbols for anterior and posterior are used?

Ext Fig 3-1: Typo in legend - efferent neurons are shown in panel (c) not (b)

Ext Fig 4-3: Some kind of typo here: "the fraction of synapses in the left vs right hemisphere is shown for those neurons receiving most of their neurons laterally"

Author Rebuttals to Initial Comments:

Reviewer comments appear below in black font, and our responses in blue.

REVIEWER 1

Dorkenwald et al report the complete connectome of the whole adult fruit fly brain. This is a landmark achievement and this is a landmark paper. It is of course challenging to summarize both all the methodology leading to this accomplishment and the many discoveries resulting from this connectome mapping study in the format of one (or two) Nature papers. Indeed, there are many (under preparation) companion papers cited in this paper. Therefore, the goal of this paper appears to be to provide a road map to the accomplishments and discoveries of this multi-disciplinary team of scientists. I will review this paper from that perspective — how well does this manuscript summarize the work and provide a useful introduction to both experts in fruit fly systems neuroscience and to the broader community of interested neuroscientists, biologists, computer scientists, and engineers.

In general, the paper is well written and covers a lot of ground in a clear and concise manner. I'm very enthusiastic about this paper. Here are my major comments:

Major comment 1

Flywire used existing FAFB volume and computational tools (both prev published by others and custom developed for flywire) to segment neurons, detect pre/post synapses, predict neurotransmitter type, attach synapse to neuron skeletons, and cell type assignment. However it is unclear to what extent the "connectome" construction is complete, and which aspects are automated due to flywire development and which aspects require more human intervention to achieve completion. It would be helpful to see a flow chart diagram to show what resources flywire developed/invested effort (in terms of person years as well as dollars) in to go from a sample brain to EM volume to a "connectome", along with amount of work left that could be automated vs require further human effort. For example, a majority of neuropils have <50% of post-synapses attached to a proofread segment (Ext Fig 1-3). While the lower completion rates is not unexpected, it would be useful to comment on how to deal with this in interpretation or how to treat synapse counts.

This is a great suggestion. We have now added a flow chart figure to the manuscript (Extended Data Fig. 1-1) showing how the various data products depend on each other, which ones were imported from previous publications, and which ones were generated via the FlyWire collaboration. We also highlight which of these data products are fully automated, semi-automated, or required manual effort. We have now also included text (lines 170+) on the completeness of proofreading of the automated segmentation, as well as on person-years spent proofreading. We also provide Extended Data Figs. 1-2 and 1-3 on the completeness and accuracy of FlyWire's reconstruction (automated segmentation + proofreading) compared with the hemibrain (in terms of synapses attached to a reconstructed neuron and completion rates by neuropil. The FlyWire connectome could be made more complete by attaching more twigs (tiny neuron segments containing mostly post-synapses) - currently 55.3% of post-synapses are unattached in the dataset; this number is comparable to the hemibrain dataset. This is one advantage of FlyWire - community proofreading can continue (lines 188+), as needed for science, and we will continue to release new data versions in Codex.

Major comment 2

This type of project requires concerted effort across many people and academic groups. While it is understandable to not rehash detailed analysis/findings by other manuscripts in prep, this manuscript contains more than 10 "in prep" citations that truncated any meaningful/deeper discussion of the data/result presented in this manuscript.

For example:

CAVE, the proofreading tool to assign synapses to neuron segments, is an "in prep" citation. Given one of the major contributions of the paper is to assign synapse to neurons for creating a wiring diagram, it would be helpful to at least summarize what technical advances were made that allowed the flywire community to generate a "complete" diagram (and distinct from prior art). Is this something that can translate to new EM volumes? to different organisms (e.g. mammalian brains).

There are numerous other instances where the authors simply refers to "in prep" and not discuss more:

gustation, mechanosensation analysis using flywire diagram Shiu et al. 40, Eichler et al. 41

sexually-dimorphic neurons, deutch et al in prep

optic lobe and central brain, kind, garner et al in prep

whole brain network analysis, Lin et al in prep

does not go into any info on optic lobe, but refers to Heckman and Clowney in prep

some VCN are peptidergic for circadian rhythm, Reinhard, Fukuda et al in prep

tracheal system analysis? Colodner et al in prep

hue selectivity, Christenson et al in prep

If the purpose is to highlight how much utility the flywire resource is able to provide, may be better to summarize these succinctly together. is it primarily the completeness of the diagram that enables new science? if so, highlighting these researches together in one section would make it more obvious that it is worth having a "complete" connectome.

We apologize for this - and agree with the Reviewer that the numerous "in prep" citations are less than ideal. Fortunately, almost all of these papers are now published on bioRxiv. We have replaced all "in prep" citations with the bioRxiv reference. We left the last remaining such paper (Deutsch et al.) in place to credit it for the Fruitless/Doublesex gene expression annotations that are already openly available to the community in Codex and plan to replace it with the bioRxiv reference before ours would go into print - or remove it if it is not out by that point. Importantly, the CAVE manuscript is now available: <https://www.biorxiv.org/content/10.1101/2023.07.26.550598v1.abstract>

Major comment 3

The network analysis provided here are "neuropil projections" and "flows" (fig 2b, 4, 6) and some statistics on the connectome graph (in vs out node degrees, etc). and a connectivity analysis is provided for ocellar circuit (fig 7). The authors generally analyzed organization based on hemisphere or neuropil differences, which do not require cell typing. The authors avoided doing central brain analysis since that's covered by the companion manuscript. But given optic lobe neurons account for 73655 out of 114423 intrinsic neurons, there is surprisingly no analysis provided for optic lobe neurons in fig 5. If this is due to another "in prep" paper, consider incorporating the optic lobe anatomy into an earlier figure if the purpose is to demonstrate completeness.

We have now completed typing of ~37,000 optic lobe intrinsic neurons in one optic lobe (right) - these can be reduced to 224 types using morphological and connectivity information. We have now written up results of the cell typing in a new paper on bioRxiv (<https://www.biorxiv.org/content/10.1101/2023.10.12.562119v2>) and reference this work within the current paper. The goal of Fig. 5 is to show that the optic lobes are available for analysis, allowing scientists to investigate complete visual circuits, including those that span an optic lobe, for the first

time. To better highlight this resource we have now extended our displays in Fig. 5 showing unicumnar, multicolumnar, and neurons that span most of the medulla.

In general, fig 5 contains little in the way of concrete “results” and could be improved by adding analysis of the optic lobe. For instance, if the goal of fig 5b is to show the “true tiling”, it would be nice to analyze the nature of the lattice or compute inter neuron distance/angle distributions, etc. It would be nice to add additional high level connectomic analyses of the optic lobe. Disappointingly, very little new information is provided in the “Optic lobes: columns and beyond” section. Specific cell types like Mi1, Dm12, LPi1-2 are singled out for comment here, but the discussion do not necessarily reveal new information concerning these cell types.

We thank the reader for this suggestion - analysis of the optic lobe required first separating the roughly 37K neurons in an optic lobe (just those intrinsic to the optic lobe, not counting the neurons at the boundary between the central brain and optic lobe that either project into the brain (visual projection neurons) or into the optic lobe (visual centrifugal neurons)) into cell types. This was a major effort that we recently completed and have now posted on bioRxiv:

<https://www.biorxiv.org/content/10.1101/2023.10.12.562119v2> . That manuscript provides a complete catalog of types for the optic lobe intrinsic neurons, with the following key findings and contributions:

- * doubling the number of known cell types in the optic lobe
- * most of the newly identified cell types contain between 10 and 100 cells each and integrate information over medium distances in the visual field
- * several existing cell type families expand significantly, suggesting increased complexity in processing color, motion, and form
- * new "serpentine medulla intrinsic" family of types is identified, containing the most types among all families

In addition, the identified cell types are self-consistent and validated through various methods.

Minor comments

1. The definition of segmentation in the nomenclature table at the start of the paper could be improved. “segmentation: product of automated pipeline”, does not convey much useful information. Other definitions could also be written more clearly.

We improved the nomenclature table.

2. A synapse thresholds of 5 and 10 synapses are variously used in Dorkenwald et al and Schlegel et al. Would be nice to be consistent? You are likely establishing conventions for the field?

Thank you for pointing that out. After performing several analyses in different parts of the brain, we think a threshold needs to be adjusted to the individual analysis and clarified that in the text. Schlegel et al. showed that synaptic connections with >10 synapses are stably found in the Hemibrain and both hemispheres in FlyWire. However, this is taking variation between brains and hemispheres into account and not just technical noise. Schlegel et al. do not apply this as a threshold, neither do we think that ≥ 5 represents a threshold that should be applied to all analyses. For instance, the optic lobe paper (Matsliah et al., 2023) showed that the threshold has to be reduced to <5 to reliably identify cell types within the optic lobe based on connectivity.

3. Can you comment on the neurotransmitter type predicted for the visual centrifugal neurons and whether they play a neuromodulatory role? An example is given of OA-AL2b2 but it would be nice to hear if the other VCNs are also octopaminergic,

We checked the neurotransmitter predictions for the VCNs and the majority (305/524) were predicted to be GABAergic, followed by cholinergic (100) and glutamate (75).

4. Would be nice to have a table of neuropil and neuron/cell type acronyms to full names along with the hierarchy. Did I miss it?

We updated the neuropil figure to include a lookup table for all acronyms and colors (Ext. Data Fig. 1-2).

5. I must confess that I don't find the information flow section and fig 6 to be super enlightening. Fig 6b is nice to see as an analysis of how gustatory information potentially propagates through the brain. But I don't really know how to make sense of the rest of the figure and analysis (including the UMAP analysis). Is the UMAP analysis the only way to discover that there are two classes of KCs targeted by ORNs and VPNS? This is not a condemnation — there might be other scientists who might find this interesting (I hope?).

The connectome is complex and we sought ways to reduce its complexity to derive insights - one mechanism was to sort neurons within the central brain relative to the sensory periphery (Fig. 6). This allowed us to estimate proximity to each of twelve afferent populations, to determine how inhibitory neurons are organized relative to the periphery, and to generate a useful visualization of the neurons with inputs in the central brain (Fig. 6c) - not surprisingly, cell classes (e.g., antennal lobe lateral neurons) cluster within this map, but it is interesting to find dispersion within a cell class (e.g., among Kenyon cells or among central complex neurons; Extended Data Fig. 6-2).

We improved the readability of the figure in multiple ways: We better connected 6b to the UMAP plots by showing how the population of neurons at each rank is distributed within the UMAP, and by coloring the render according to the heatmap. We further added more super classes to the percentile plots (6d, Ext. Data Fig. 6-1a) to allow for comparison and added a reference for a uniform distribution highlighting how far away from normal the observed distributions are in the percentile plots (Fig. 6d,e). We expanded Ext. Data Fig. 6-1a to include all super classes.

6. The ocellar circuit is very exciting to see, but the figure and description is hard to follow. I like the circuit diagram in fig 7f, but that is only a caricature. The real data are shown in fig 7(c,d) but we only see the morphologies of the neurons but not the circuitry. It would be nice to see a schematic of the circuit in the form of a graph with nodes and edges.

We experimented with multiple ways of displaying the connectivity of all neurons in the ocellar circuit and were unsuccessful to come up with a graph or matrix that allows the reader to parse the connectivity. Instead, we provide the connectivity of subcircuits in Ext. Data Fig. 7-1 e,f.

7. In Fig 4b, would be nice to show the equation next to the graphic representation.

We have added the equation to the figure.

8. In Fig 4, the SEZ is highlighted since it's not present for the most part in the hemibrain. Would be nice to do the same analysis for all the other regions in supplementary, or provide Jupyter notebooks or links to how such analysis could be performed for each neuropil.

We repeated the same analysis for all neuropils and added them to Supplemental Information 2 and 3.

9. Fig 4g,h not obvious the color code is matched for ipsilateral vs bilateral vs contra

We added a legend below Figure 4 h to clarify that.

10. Given how rare contralateral and bilateral neurons are in general, consider expanding the plot to also show these by fig 4h "super class"

We looked into making a separate plot for showing the distribution of super classes for bilateral and contralateral neurons. We found that this plot is dominated by the "central" class and gives little insight beyond that, so we did not add it to the figure.

11. Ext Fig 1-2. What is the color code for the dots?

Each dot represents a neuropil. The color code is given in Ext. Data Fig. 1-2 and not repeated because of its size. The caption refers to Ext. Data Fig. 1-2.

12. what's the lower fraction of attached synapses relative to hemibrain for specific neuropils due to? need more proofreading for correction?

It is difficult to evaluate the cause of this difference as there can be multiple factors at play, including differences in the EM data or between flies. We did not prioritize any specific neuropil other than starting with the central brain and then completing the optic lobes. Our selection criteria for selecting segments were equally applied across the brain and specific neuropils. The hemibrain project paid special attention to certain neuropils (e.g., the central complex, Hulse et al. <https://elifesciences.org/articles/66039>) which could explain the difference in attachment rate for those regions between hemibrain and FlyWire. On the other hand, FlyWire has significantly more pre-synapses attached across neuropils, and more post-synapses attached for certain neuropils (like PLP, PVLP, and WED, see below). More proofreading can certainly be done to increase the attachment rate and it is one of FlyWire's strengths to allow for future proofreading should it be desired (see line 188+).

13. was there a technical computational advance that flywire made (CAVE in prep??) that improved/could improve attachment rate?

The attachment rate is heavily impacted by orphan twigs and short segments which in turn were never selected for proofreading. Improved alignment of the EM slices is critical for enabling the automated segmentation methods to combine these small fragments to larger ones. The FlyWire team realigned the original FAFB (v14) volume and created the version used by FlyWire (v14.1).

CAVE is a system that allows users to proofread, annotate, and analyze connectome datasets. CAVE was important for the FlyWire effort but it did not introduce advances specific to the attachment of twigs.

14. In fig 2b and many other places throughout, the fly brain underneath can be difficult to see, consider making it more visible? Especially hard to see when printed.

We recognize the difficulty to see the underlying brain in printed versions of the manuscript. We adjusted the opacity of many of the brain renders throughout the manuscript to improve the print quality.

REVIEWER 2

In this paper, Dorkenwald and colleagues present a new data set consisting of the connectome of an entire brain of an adult *Drosophila melanogaster*. This sophisticated dataset will be of particular interest to those seeking to understand circuits in the fly's brain, while it is simultaneously a leap forward in understanding the connectivity and structure of brains more generally. In particular, this connectome is a significant advance over the prior adult *Drosophila* connectome, the 'hemibrain', published in 2020, because this one encompasses the entire brain, including optic lobes, direct sensory inputs, and descending and ascending connections between the brain and the ventral nerve cord. This enables far more holistic analyses of the fly's brain, from multisensory integration to tracing how visual signals enter the central brain. The tracing of ocelli signals is presented as an interesting example of what can be done and this yields real, testable predictions about fly brain function.

This paper is written as the document of record supporting the first release of this data, which will evolve over time with more community and specialist inputs. Overall, this is well-done, but I think there could be a few improvements, the most important of which I describe below.

- 1) I think this manuscript could do with a strong round of editing. Below, I collected some minor and major editing issues that I noted while reading the paper.
 - a) In the first paragraph of the introduction: The second sentence felt rather unconnected to the first. This whole paragraph could flow better. (Side note: reviewers always appreciate line numbers.)
 - b) Later, the paragraph beginning "We already showed..." should have a topic sentence that's about the several ways the authors have validated this dataset, which are then enumerated in this paragraph.
 - c) Later, "but no missing connections" seems rather ambiguous to me as a dependent clause. Do the authors mean avoiding FP but not missing connections, or "while not missing connections". I think authors mean latter.
 - d) "the additional partners are from the same or different cell types" – unclear what is meant by "additional" here, or what this sentence really means.
 - e) The discussion felt very disjointed, like a hodge podge of small, disconnected points.
 - f) Also in the discussion, I'm not sure about the Brain simulation points. Some serious caveats about the approach are likely in order for non-specialists. We're not just going to suddenly successfully simulate the fly brain by using this data set. Citation 115 is less a simulation than a model optimized to perform a task, where the connectome acted as a strong constraint on model architecture. This part of the discussion could be clarified a bit. As written, it seems a little Polyanna-ish.

We thank the Reviewer for these comments on the text. We have now addressed all listed concerns in the manuscript.

- 2) I found the analyses surrounding Figure 6 rather confusing, and was not sure at the end of it what I had learned, or how the field might make use of these descriptions of the connectivity in the connectome. Does the last panel show that different sensory inputs project to different brain regions, for the most part? In F6c, if one organizes neurons by quantiles, one expects a uniform distribution, but no combination of the subsets of neurons in this panel seem to add up to a uniform distribution – a bunch of neurons must not be accounted for, but I couldn't figure out which. (This includes staring at the supp fig for a while, too.)

The connectome is complex and we sought ways to reduce its complexity to derive insights - one mechanism was to sort neurons within the central brain relative to the sensory periphery (Fig. 6). This allowed us to estimate proximity to each of twelve afferent populations, to determine how inhibitory neurons are organized relative to the periphery, and to generate a useful visualization of the neurons with inputs in the central brain (Fig. 6c) - not surprisingly, cell classes (e.g., antennal lobe lateral

neurons) cluster within this map, but it is interesting to find dispersion within a cell class (e.g., among Kenyon cells or among central complex neurons; Extended Data Fig. 6-2).

We improved the readability of the figure in multiple ways: We better connected 6b to the UMAP plots by showing how the population of neurons at each rank is distributed within the UMAP, and by coloring the render according to the heatmap. We further added more super classes to the percentile plots (6d, Ext. Data Fig. 6-1a) to allow for comparison and added a reference for a uniform distribution highlighting how far away from normal the observed distributions are in the percentile plots (Fig. 6d,e). We expanded Ext. Data Fig. 6-1a to include all super classes.

- 3) Some more discussion seems due on the limitations of this study and comparison to other ones.
- One major difference between this dataset and the hemibrain dataset is that the hemibrain acquired voxels with $8 \times 8 \times 8$ nm volumes, while here the volumes are $4 \times 4 \times 40$ nm. This must have pros and cons, and lead to differing levels of confidence in some segmentation and identification processes. I think this would be valuable to comment on briefly in the main text, but perhaps in more detail in the methods.
 - Many of the methods, like inferring a neuron's neurotransmitter, rely on methods published elsewhere. Here, it's unreasonable to fully recount these other methods in this paper. But it would be exceedingly helpful to readers to present short summaries of these other methods in the methods section, and perhaps highlight uncertainties, error-rates, or limitations in the different methodologies used.

We extended the Methods section and now briefly summarize several methods that were used to create data (especially synapses) that were imported into the FlyWire dataset. In particular, we highlight what data (e.g., brain regions) these were trained and evaluated on and what performances were published. We overhauled the discussion section and further highlighted differences to the hemibrain on lines 781+. We also compare the hemibrain and FlyWire datasets with regards to completion rates and number of synapses in Extended Data Figure 1-3:

- 4) Finally, probably the most important issue: the value of this dataset is really in its accessibility and the tools that allow the community to use it. The hemibrain dataset has been so influential in part because it provided excellent tools to access and analyze the data. As part of this review, I asked a 4th year PhD student in lab to use the Flywire data and tools for a small project in lab, during which he spent ~ 2 full days accessing the data. I would add that he is a sophisticated user of the hemibrain dataset via their API, but is not a power user of the type who is familiar with catmaid, for instance. Below, we present several issues he brought up that could improve the accessibility to data, the data

itself, and documentation of the code. If he spent 1 hour trying to figure something out, if it can be fixed to be easier, it could save 1000s of hours collectively in the community!

We agree that it is important to reduce the barrier for entry for new users as much as possible to facilitate wide dissemination of the dataset. This is why we designed Codex (codex.flywire.ai) for easy access to the data, even without any programming knowledge. While revising the manuscript, we have added the following features to Codex: Motif-based search with dedicated UI, catalog of optic lobe cell types, integrated 3D viewer with annotation-based coloring of cells, tools for finding similar cells based on connectivity metrics. We have now also added Python notebooks that provide example code for programmatic queries to the connectome data, and have recorded a workshop that goes along with those notebooks where we walked a group (>50) users through them (see youtube link). Additionally, the data is available through multiple UI tools for non-programmatic access that go beyond the functionality provided in neuprint (e.g., in addition to Codex, also brainconnects.io, [catmaid spaces](https://catmaid.spaces.flybrainlab.org), [flybrainlab/NeuroNLP](https://flybrainlab.org/NeuroNLP) - all of these tools are linked through Codex and our website). We have setup channels for community members to request new features in Codex as well. This work is ongoing and we will continue to improve our tools, add code examples and hold workshops for the community to make the dataset more accessible.

Data accessibility (comments from the graduate student)

1) The paper should explicitly state what functions users should use CAVE, fabseg, and NAVis for. For instance, in figs 1b and 1c, the authors could specify which tool can be used to grab each data type. It was time consuming to try to figure out what should be used for connectivity, what should be used for voxels, what should be used for neuron morphology, etc. A table could also help.

We created tutorial notebooks and videos to enable users to find the tool they need to access the data they want. Where possible (size constraints), we made data sources available for download in Codex.

Tutorial video:

<https://www.youtube.com/watch?v=B5EeqVIOqjk>

Notebooks:

CAVE:

<https://github.com/seung-lab/FlyConnectome/blob/main/CAVE%20tutorial.ipynb>

Image, segmentation and meshes:

<https://github.com/seung-lab/FlyConnectome/blob/main/Segmentation%20and%20Mesh%20Access.ipynb>

Neuropils and point lookups:

<https://github.com/seung-lab/FlyConnectome/blob/main/Neuropils%20and%20Point%20Lookups.ipynb>

2) How can users get voxel level data? Using NAVis? Two days' worth of poking around didn't reveal how to do this. The NAVis tutorial that was most obvious for this ("Voxel / Image Data") seemed to do something else.

We created a jupyter notebook to walk a user through accessing the EM volume image, segmentation and mesh data:

<https://github.com/seung-lab/FlyConnectome/blob/main/Segmentation%20and%20Mesh%20Access.ipynb>

3) The links were easy to follow.

4) The flywire Youtube channel was really helpful and could perhaps be linked prominently in the paper or on the website. Similarly with the Slack channel.

5) On the homepage of "flywire.ai", it says "For programmatic analysis, see natverse (R) and CAVE (Python)." However, fabseg and NAVis should also be included in this sentence. Also, fabseg was not included in Figure 1C.

We have added fabseg to Figure 1C.

6) The data seemed to be already stored in google cloud buckets – the PhD student would have found it useful if the data was accessible via tensorstore.

We use cloudvolume to interface with volumetric data. Cloudvolume is a widely used tool in connectomics and has comparable functionality to tensorstore while (at the moment) being more versatile in terms of data formats and multi-scale support. We created an example notebook for how to access the data with cloudvolume:

<https://github.com/seung-lab/FlyConnectome/blob/main/Segmentation%20and%20Mesh%20Access.ipynb>

7) If neuroglancer was used to generate figures in this paper, it would be awesome to include a supplemental table with the URLs for each figure, so that people can look up exactly what's in them!

We did not use neuroglancer to make figures and instead used meshparty (<https://meshparty.readthedocs.io/en/latest/>) which enables programmatic generation of 3d renderings and is compatible with cloudvolume. Meshparty has extensive documentation and interfaces well with the FlyWire dataset.

8) For visualizing neurons, it would be helpful to have an easy way to get a set of A-P and D-V and L-R (or medial-lateral) axes on any plot. It is easy to become disoriented.

We assume that this comment refers to visualization in neuroglancer (since figures in the paper have axes labels). This is a great idea! While we were not able to add this feature to neuroglancer for this revision, we have added it to our list of future improvements.

Data content (from the graduate student)

1) There did not seem to be a clear way for people to edit neurotransmitters. There should be an option for users to input empirically determined neurotransmitters for neurons, as well as the paper(s) supporting the determination. In that case, each neuron type should have a predicted and empirical neurotransmitter (when available) attached to it.

While our data releases are static to facilitate reproducible analyses, community annotations can be added over time. These allow users to add information for neurotransmitters and papers among other annotations. As an example, the cell shown below (screen shot from Codex) has community labels at right - this list can include any submitted annotation from the community, including neurotransmitter information. This particular cell has been curated, and the aSP10 annotation promoted to its cell type name.

Additionally, we added a feature to codex that allows users to flag issues with a cell such as bad neurotransmitter information (“Report error / issue with this cell”).

2) The fetched data comes with fields such as “pre_x, pre_y, pre_z, tag, type, hemilineage, confidence, changed, superceded_id, pt_root_id, pt_supervoxel_id, id, user_id,” etc. On the flywire website and relevant github repositories, there should be a table explaining what each datatype means. Many of these fields were pretty opaque and documentation would be immensely helpful.

We agree that better documentation of the individual columns returned was needed. It was not clear how the data described by the Reviewer was obtained, what data source was queried, and what tool was used. For CAVE, we added descriptions for the columns returned to the queries described in the tutorial notebook above.

3) In the output of “fetch_synapses” in fabbseg, one of the columns is “id”. Is this id unique for each presynapse density? It was clear from the paper that the presynapse-postsynapse pair creates multiple presynapse coordinates for the same presynaptic site, but it wasn’t clear how users know which presynaptic coordinates to combine to get all postsynaptic connections coming from one presynapse. In the hemibrain dataset there is a single presynaptic site, but in Flywire it looks as though each synapse lists both pre and post site. This makes it hard to tell how polysynapses are arranged.

The “id” column returned by fabbseg (and any other tool) for synapses describes a unique synaptic connection without identifying presynapsynaptic sites. It is currently not straightforward to tell how polysynapses are arranged. This is a challenging problem to do post-hoc for the Buhmann synapses and was attempted and discussed multiple times by and with community members. This is on our radar and we will make this information available when it is computed.

4) It would be very helpful to have a consensus name for the neuron type associated with each neuron. Right now, this is in the annotations field and depends on the user who annotated. For instance, TmY17 has the following annotations

- a. Transmedullary Y neuron 17, TmY17 (A. Nern naming system)
- b. Transmedullary Y neuron 17, TmY17, (A.Nern, personal communication)
- c. Transmedullary Y neuron 17, TmY17, TmY17_R (A.Nern, personal communication)
- d. Transmedullary Y neuron 17, TmY17,(A.Nern, personal communication)

If someone searched for TmY17_R, they would miss a bunch that did not have the _R appended. Similarly, a search in this field for L4 neurons yields every single reference to Gal4 drivers in the annotations. It would be helpful to have a single field that contains cell type name information, which is exceedingly useful in querying the hemibrain dataset.

We agree that consolidated annotations for neurons are important for programmatic analysis. We now currently have annotations for neurons in Codex list under the “cell type” or “hemibrain type” headings. These annotations come from three primary sources: i) Schlegel et al. provide consolidated and normalized type annotations fo neurons in the central brain, including cells at the boundary

between the central brain and optic lobes; ii) The recent paper on the optic lobe cell typing (Matsliah et al., <https://www.biorxiv.org/content/10.1101/2023.10.12.562119v2>) adds standardized cell types for ~37k neurons in one optic lobe (and we will soon mirror these to the other optic lobe) listed under the "cell type" heading; there is an additional Optic Lobe Cell Catalog page in Codex (https://codex.flywire.ai/app/optic_lobe_catalog); iii) Deutsch et al. (in prep) provide annotations for xx sexually-dimorphic neurons that express Fruitless or Doublesex, listed under the "cell type" heading, with the gene they express (Fruitless or Doublesex) listed under "gene". These are well suited for programmatic analyses. The community annotations (last column in Codex, see above), by design, contain a variety of labels; anyone can contribute to this category of label, to lower the barrier for users to add annotations and encourage users to contribute. These annotations are well suited for discovery and collection of the community's knowledge but additional work is required to distill them into a vetted or curated database of annotations.

Code functions and accessibility (also from this user)

1) The README file in the CAVEclient github repository is currently only one sentence. It should at least include how to install CAVEclient, and include the link to the readthedocs.io documentation.

The CAVEclient can be installed with "pip install caveclient". We added this to the readme. We added a CAVE tutorial that shows how to query data specific to FlyWire: <https://github.com/seung-lab/FlyConnectome/blob/main/CAVE%20tutorial.ipynb>

2) It would be very useful to have a function that provides the neuropil associated with a given coordinate (function argument). In whichever library is best here.

We added a notebook showing how to load the neuropil meshes and doing point lookups within them: <https://github.com/seung-lab/FlyConnectome/blob/main/Neuropils%20and%20Point%20Lookups.ipynb>

3) Critical: The documentation in fabseg assumes the user knows the ID of the neuron they are investigating, but it is more likely the user knows the neuron type before the ID. Therefore, it would be very useful to have a function that grabs the neuron IDs of all neurons in the same hemilineage, VFB ID, flow, superclass, class, cell type, neurotransmitter, and/or annotation. In analogy with neuprint-python, these could be inputs to the NeuronCriteria function. So the function in neuprint-python lingo would be `fetch_neurons(NC(class="MBON", neurotransmitter="dopamine"))`. The current arrangement might be a historical artifact, since neuron IDs were known before types; but now that all is known, it would be super-useful (the most useful!) functionality to have. If it already exists, it was not obvious to this user.

4) Similarly, the `fetch_synapses` function should allow users to specify the neuron type of the presynapse and postsynapse, not just the ID.

5) `Fetch_synapses` should have a default argument `use_new_id = True`, where the output ids in the "pre" and "post" columns are the most up to date ids, so that users don't have to constantly use the `update_ids` function. This created a lot of confusion for us, and could lead to bugs.

We worked with Philipp Schlegel who updated fabseg and released version 2.0 fixing many of the issues mentioned here.

6) Critical: More coding examples would be important to give out. For instance, one example could be a researcher asking "I want to know the inputs and outputs adjacency matrix of all neurons in this

type". Almost all users will use this dataset for connectivity purposes, so having this tutorial/example and similar ones would really allow most users to get a running start on their analyses.

We added a CAVE tutorial here:

<https://github.com/seung-lab/FlyConnectome/blob/main/CAVE%20tutorial.ipynb>

7) In the NAVis tutorials, the "Voxel/Image Data" in the "Import/Export" section should include how to grab the grayscale and neuron segmentation, given a list of coordinates or a bounding box. Additionally, the tutorial could include how to show a cross section.

Navis is not the correct tool for accessing the volumetric data. Cloudvolume facilitates access to imagery, segmentation and mesh access. Here is a tutorial for the FlyWire dataset:

<https://github.com/seung-lab/FlyConnectome/blob/main/Segmentation%20and%20Mesh%20Access.ipynb>

More minor issues:

1) "any normal Drosophila individual" should read "female", "wild type" instead of normal, and presumably melanogaster in addition the genus. Authors should also note the fly's age somewhere in the paper – I could not find that anywhere.

We adjusted the text accordingly and added the age and genome type information to the methods.

2) "average F1-score of 99.2% by volume". This phrasing contains jargon, pretty impenetrable to the reader. I am guessing that this is good, but maybe just say so in a way that leaves out the jargon. I could not find "F1-score" defined anywhere in this manuscript.

We added a definition of F1-Score to the methods

3) Would be nice to also report synapse density as number of synapses per neurite path length; the current reported value seems to be Euclidian distance.

We added the synapse density per neurite path length to the manuscript were we introduce the total path length of all neurons (~149 m). With ~122 million attached presynapses, this averages to 0.82 presynapses / μm path length.

4) Dendrite-soma-axon paradigm from mammals "may not apply" should read "does not apply". It can't with these cell morphologies, right?

That's right. Fixed.

5) Figure 3d – ylabel "cumulative ratio" should read "cumulative fraction", right? If not, please specify what the ratio is.

That's right. Fixed.

6) Figure 3f – labeled "exp fit" but caption says it's a truncated power law distribution; "exp fit" implies "exponential fit". A straight untruncated power law seems like it would be a good fit to this distribution and it would be useful to know the exponent... Perhaps the prior analysis was missing these rare connections with many synapses. Also, where the exponential cutoff is mentioned in text, it cites 3g, but should be 3f.

We ended up removing this plot from the paper. We added it initially to compare with the “same” plot in the hemibrain paper. However, we realized that our fit logic differed from the one used there and we were able to reproduce the fitting logic that was used in the hemibrain paper.

7) Figures 3g and 3h – unity lines would be useful to show here, or fit lines as well as the pearson correlation coefficients given in caption.

We added diagonals to the scatter plots where possible.

8) Cubic microns seem like more natural unit for this dataset than mm³ (which are tiny numbers), and the manuscript kind of goes back and forth. Neither is an SI unit, so I’m not sure there’s a good reason to prefer either.

We used units that provided the shortest number of digits possible. With cubic, the steps between units are 9 letters, and volumes such as 0.0018 mm³ would read 1,800,000 μm³. We argue that this is not better than using m³.

9) “CT1 arborizes exclusively to medulla”. No, I believe CT1 arborizes in lobula, too, making synapses onto every T5 dendrite there. CT1 is likely GABAergic, but it appears to be playing a far more local role, different from the one described in the text of “local feedback gain control”. Its local response properties influence local motion detection (Meier & Borst 2019, Gonzalez-Suarez et al. 2022). The authors mention the compartmentalization later, but it’s weird to lump it in here with feedback gain control neurons, like APL in the olfactory system, when that is not at all its known function.

Thank you for catching that. We removed the mention of the CT1 in this context.

10) Figure 5f – Could you choose colors with stronger contrast than green and blue for the two LPi neurons? Also, is this phenomenon true of other LPi neurons?

We updated the plot with better colors.

11) Figure 7a. There appears to be an error with the L/R in the diagram? If we’re looking down on the head, these must be flipped, no? This is not related to the left-right flip in the volume, right?

We revisited the figure and the annotations on the top-down view appear correct. However, we found a mistake in the side annotations for the ocellar retinula cells. We apologize for the confusion the side flip created and added better and more annotations to the 3d renderings throughout the manuscript.

12) “caps out at an input fraction of 30%”. This feels like lab slang, and it is also not clear in the text if this cap is the method or a description of the data. The figure shows that this is part of the method; how was 30% chosen for the saturation?

We adjusted the text.

13) I think that in the optic lobe results section, the authors could highlight even more that the hemibrain left out enormous chunk of the peripheral visual system, which is importantly included here! In this section, the authors could mention subsequent reconstructions of other bits of the optic lobe, not connected to the rest of the brain (Shinomiya et al. 2022).

We have now completed typing of ~37,000 optic lobe intrinsic neurons in one optic lobe (right) - these can be reduced to 224 types using morphological and connectivity information. We have now written up results of the cell typing in a new paper on bioRxiv (<https://www.biorxiv.org/content/10.1101/2023.10.12.562119v2>) and reference this work within the current paper. We extended our displays in Fig. 5 showing unicumnar, multicolumnar, and neurons that span most of the medulla.

14) It's fantastic that there are all these companion papers in the works, but this paper should also stand alone, which it does. I'm just not quite sure what balance to strike here, in terms of mentioning all these companion papers.

We apologize for this - and agree with the Reviewer that the numerous "in prep" citations are less than ideal. Fortunately, almost all of these papers are now published on bioRxiv. We have replaced all "in prep" citations with the bioRxiv reference. We left the last remaining such paper (Deutsch et al.) in place to credit it for the Fru/Dsx annotations that are already available to the community and plan to replace it with the biorxiv reference before ours would go into print - or remove it if it is not out by that point.

REVIEWER 3

This paper by Dorkenwald et al. describes a landmark achievement in neuroscience: the first neuronal wiring diagram of a complete adult brain. The complexity and completeness of this wiring diagram, reconstructed from an adult *Drosophila* female, represent a significant advance over previous connectomes (e.g. *C. elegans*, larval *Drosophila*, or the *Drosophila* hemibrain) and enable new studies that were not previously possible. In particular, unlike the fly hemibrain, this connectome includes both hemispheres and all afferent and efferent neurons. This connectome has already begun to revolutionize the study of neural circuits, as evidenced by recent papers and preprints using this dataset.

Aside from describing this accomplishment, this paper also includes many interesting analyses regarding the major types and features of brain neurons, principles of connectivity at the level of cells and brain regions, the relationship between connectivity and neurotransmitter identity, and how information from different sources flows across the brain. These findings reveal new insight into brain organization and function, and basic metrics described in the paper will likely provide references for the field for years to come. The authors have done a commendable job of visualizing large datasets and choosing appropriate metrics for gaining insight into the data. In my opinion, there is no question that this paper warrants publication in a venue such as *Nature* where it will reach a wide-ranging audience.

I have no major issues with the authors' analyses or conclusions, only comments aimed at improving the paper.

General comments:

1. For major metrics of connectivity, it would be nice to explicitly compare to the hemibrain data if possible. For example: number of synapses per neuron, the distribution of the number of synapses per connection, median in-degree and out-degree.

We agree that comparisons to the hemibrain are important. For this analysis, we would like to refer to our companion paper by Schlegel et al. (also available via biorxiv: <https://www.biorxiv.org/content/10.1101/2023.06.27.546055v2.abstract>) which performs comparisons with the hemibrain in depth.

2. The projectome (Fig. 4A) is a very interesting dataset, and I think a bit more could be said about the overall connectivity patterns. Which regions are most highly connected? Do highly connected regions tend to be in closer physical proximity? Which are the few regions that show the strongest left-right connectivity?

This is a great suggestion. We extended our description and interpretation of the projectome in the main text (lines 471+).

We included text on the regions that are most strongly connected: "The largest weights in the projectome tend to be internal to individual neuropils, such as ME to ME or FB to FB. The largest inter-neuropil projections overall are LO to ME, while within the central brain the largest inter-neuropil projections are MB-ML to MB-CA."

We added an analysis to compare the neuropil distance (distance between neuropil centers) with their connection strength to Ext. Data Fig. 4-2. We did not find a correlation between distance and connection strength which we quantified by the number of neurons projecting between the two:

3. The Discussion feels a bit choppy. I would suggest combining some of the very short sections into longer and more coherent sections, for example focusing on 1) methodology, 2) implications of the connectome, and 3) future directions. While this is the authors' prerogative, I also noticed that the Discussion does not include much interpretation of the scientific results (e.g. connectivity patterns, differences in excitatory vs inhibitory neuron connectivity, information flow).

We recognized that the discussion needed improvements and rewrote the discussion. We roughly followed the reviewer's suggestion on regrouping the points made in the discussion.

Specific/minor comments:

Introduction: It sounds like the authors may be drawing a distinction between "wiring diagram" and "connectome", but this was not clearly defined.

p. 5, "third validation method": Is this really a "validation method" in terms of demonstrating the accuracy of predicted synapses? It's certainly important to know how many synapses are attached to orphan segments versus proofread neurons, but I'm not sure it validates the methodology, especially as over half of postsynapses are located in the orphan twigs. This seems like a high number.

Thank you for calling this out. We rephrased the paragraph and no longer suggest that this is a validation method but still present the data as is.

p. 7: It would be good to say something brief about how the sensory organs for non-visual sensory neurons were identified – is this based on matching to known neuronal morphologies?

Yes, this was largely based on neuronal morphologies (matching) and using nerve assignments. We added that to the text.

p. 8: I couldn't parse the second half of this sentence: "The optic lobes, which are largely absent from the 1st instar larval, are a major reason that the adult fly brain so dominates its nervous system."

That sentence was not clear and we rewrote it.

p. 10: In addition to quantifying total volume of each neuron, I wonder if you also quantified cell body

volume? Do cell body volumes also vary across orders of magnitude, and do they correlate with path length, cell type, neurotransmitter identity, or other features?

We thank the Reviewer for this suggestion. While we do not have access to the cell body volume, we quantified cell body volumes for most intrinsic neurons. The cell body volume is well correlated with the other measures such as path length. We added the accompanying plots to Ext. Data Fig. 3-2:

p. 12: "We asked whether large neurons tend to use their many synapses to create stronger connections with individual neurons versus more connections with many different neurons. The total number of synapses established by a neuron was much better correlated with its in and out degrees ($R=0.93$, $R=0.93$ respectively) than its average connection strength." The fact that this supports the second model is not obvious and could be explicitly stated.

We fixed the text to clarify which of the two hypotheses the observation supports and why.

p. 13: "GABAergic and glutamatergic neurons had much higher degrees than cholinergic neurons" and "GABAergic neurons were on average longer". I don't think either of these results is obvious from the cited figure panels (extended Fig 3-1), although I appreciate seeing the full distributions. The average or median numbers could be stated explicitly.

We added median and average numbers to the plots and inserted vertical lines indicating medians in Ext. Data Fig. 3-1j-l.

p. 19 and Fig. 6B: It may be worth mentioning that the low-rank neurons in the gustatory pathway (rank 1 and 2) look strikingly similar to second-order gustatory neurons identified experimentally (e.g. in Talay et al., Snell et al., Deere et al.), which corroborates the validity of traversal method.

Thank you for the suggestion and the references. We added this to the main text.

p. 19 and Fig. 6D: The claim that inhibitory neurons are overrepresented in the early layers would be more convincing if excitatory neurons were plotted on the same graphs for comparison (even though they're in the extended data). Also, could you be more specific about what you mean by "we identified a sequence of GABAergic and glutamatergic peaks...?"

We added the graphs for cholinergic neurons to the plots in Fig. 6d. The graphs for gabaergic and glutamatergic neurons are not only above the baseline but peak multiple times. We marked these in the plot and improved the description in the text.

Fig. 2C: I thought "optic" neurons are defined as being contained completely within the optic lobes,

but it looks like some neurons have innervations in the central brain. Are these all tracts without any synapses?

Strikingly, yes. They are tracts without synapses.

Fig. 2D: Some of these abbreviations don't seem to be defined anywhere. And I assume the numbers refer to how many neurons are contained within each nerve, but this isn't explicitly stated.

Thank you for catching that. We added all abbreviations to the figure caption.

Fig. 2F: I am not sure I can distinguish the two colors of green corresponding to gustatory vs. hygro-sensory neurons.

We changed the color of the hygro-sensory neurons to make them more differentiable.

Fig. 3F: The font seems to be inconsistent with the rest of the figure.

Indeed. We fixed that.

Fig. 4 and elsewhere: The neuropil abbreviations are not defined; please provide a table or cite a reference where the reader can find the full names.

We replaced Ext. Data Fig. 1-2 and it now shows all abbreviations next to the full names.

Fig. 4C-D: I think it needs to be stated again in the legend that the left and right sides are flipped in the images. Additionally stating this somewhere in the main text, not just legends and methods, may also be a good idea.

We added a reference to the flip to the figure caption.

Fig. 4E: I suggest explicitly stating that the insets are the same data zoomed in on specific x values.

Done.

Fig. 7: It would be helpful to include a diagram showing where the ocellar ganglion is located relative to the rest of the brain.

Fig. 7E: "glutamatergic" is misspelled on the y-axis

Fixed.

Ext Fig 1-1: It's not clear to me where the symbols for anterior and posterior are used?

We used the standard notations for vectors into and out of a page.

Ext Fig 3-1: Typo in legend - efferent neurons are shown in panel (c) not (b)

Done.

Ext Fig 4-3: Some kind of typo here: "the fraction of synapses in the left vs right hemisphere is shown for those neurons receiving most of their neurons laterally"

We fixed the text.

Reviewer Reports on the First Revision:

Referees' comments:

Referee #1 (Remarks to the Author):

I commend the authors for their revision. Most of my comments have been addressed and the paper reads much clearer now. I appreciate the new Extended Data figures laying out the nomenclature in more detail as well. A few minor suggestions.

I congratulate the authors for their remarkable achievement and they are certainly allowed a victory lap. But it would take little away from their accomplishment to acknowledge and discuss limitations a bit further. Two comments here:

1. This connectome will not be the final word on the fly brain, especially given the current completion rates. Some discussion and acknowledgement of notable missing elements in the present connectome would be valuable to the community. What are some connections which will we the current dataset is unlikely to be good enough to proofread? For instance, missing connections from the photoreceptors to L1, etc.

Further, is there a resource (perhaps a live webpage on codex), where completion rates on a cell type by cell type basis can be looked up?

2. Continuing on the theme of limitations of the present study, are there lessons learned about what could yet be improved on the imaging or staining side of things?

And finally, a minor point. Previous version said 127,978 neurons. This version says 139,255 neurons. This is good progress, but leaves open the question of completion rate. How many more neurons remain to be proofread? Completeness is now specified by pre- and post- synapse connection rates, which is great. Would be nice to also specify the neuron completion rate. How many soma remain to be proofread?

Referee #2 (Remarks to the Author):

The authors have addressed my comments appropriately from the last round. In particular, my graduate student reports that the additions to the software are clear and helpful and make it more quickly accessible. The paper also reads more clearly and smoothly after the latest round of edits. The new data on the optic lobes is a welcome addition.

Minor

Line 110 – Admonishing readers whom to cite doesn't quite feel right. This sort of text seems like it belongs on the software front page, rather than in the text of a paper. Stressing the importance of the companion paper seems totally reasonable, of course.

Line 429 – in degree or out degree here?

I notice that most plots with axes have ticklabels but not ticks. See for instance 3dfgh, 4efg, etc. This is extremely unconventional. Even though I don't see readers trying to align datapoints with specific tick marks, it seems like it would be useful to identify what is being labeled with each number. In principle, readers should be able to try to make precise alignment between axes and data positions.

If meshparty offers functionality to copy and paste links for 3D renderings so anyone can view it, it would be wonderful to include those links for readers.

This last item is under minor because it's something that I think would be in the spirit of data openness that the authors advocate in their overall approach and in the discussion of the paper: Although Nature journals do not require it, it would be nice to distribute code to generate the paper's figure panels from the database.

Referee #3 (Remarks to the Author):

As stated in my initial review, this is clearly a landmark paper that does a commendable job of describing the methods and gaining insights from this immense dataset. My previous comments have been addressed. I just have a few minor comments on the revised paper, noted below, which I'm sure the authors will easily address.

1) I noticed that even though >10k additional neurons have been annotated since the initial submission, some of the neuron numbers are smaller than before: for example, fewer neurons fully contained in the central brain (line 251) and fewer neurons in some nerves (Fig. 2D). It's not clear to me why the numbers decreased (were the neurons initially miscategorized?).

2) Fig 3F: Now that the power law fit was removed from the graph, I'm not sure what the text is referring to in lines 359-362.

3) Fig 3G-H: May want to clarify what the colored dots are.

4) Extended data Fig. 3-2A: Should include the color legend as in the previous extended data figure.

5) As Reviewer 2 mentioned in the first round, the L/R annotations in Fig 7A seem like they're flipped – why does "L" appear on the fly's right side if we're looking down on the fly? The authors said they checked this so I may be missing something, but if two reviewers had the same confusion then it probably warrants further clarification in the legend.

6) At least in my version, some of the figure panels look blurry (e.g. Fig. 2A-B).

Author Rebuttals to First Revision:

Reviewer 1

I commend the authors for their revision. Most of my comments have been addressed and the paper reads much clearer now. I appreciate the new Extended Data figures laying out the nomenclature in more detail as well. A few minor suggestions.

We thank the Reviewer for their thorough and constructive review that we believe made the manuscript better and clearer!

I congratulate the authors for their remarkable achievement and they are certainly allowed a victory lap. But it would take little away from their accomplishment to acknowledge and discuss limitations a bit further. Two comments here:

1. This connectome will not be the final word on the fly brain, especially given the current completion rates. Some discussion and acknowledgement of notable missing elements in the present connectome would be valuable to the community. What are some connections which will we the current dataset is unlikely to be good enough to proofread? For instance, missing connections from the photoreceptors to L1, etc.

We agree that it is important to better highlight this in the paper. We addressed this by adding a section to the Discussion on the limitations of our dataset.

Further, is there a resource (perhaps a live webpage on codex), where completion rates on a cell type by cell type basis can be looked up?

With our current method, completion rates can only be calculated on the basis of synapses not neurons. The existence of synapses that were not assigned to a neuron provides us with a base to compute completion for each neuropil. Since we do not know what neuron these synapses should have been assigned to, we cannot calculate this statistic for neurons or their types. We have now added an overview of the completion rates (neuropil based) to Codex (example: <https://codex.flywire.ai/app/neuropils?selected=EB>).

2. Continuing on the theme of limitations of the present study, are there lessons learned about what could yet be improved on the imaging or staining side of things?

Better imaging and staining (higher SNR, better resolution) generally improves the result of the segmentation but usually comes at a cost, e.g. imaging times. In general, we find it difficult to make these kinds of inferences post-hoc from our dataset without a more detailed analysis of the edit locations within the dataset.

And finally, a minor point. Previous version said 127,978 neurons. This version says 139,255 neurons. This is good progress, but leaves open the question of completion rate. How many more neurons remain to be proofread? Completeness is now specified by pre- and post- synapse connection rates, which is great. Would be nice to also specify the neuron completion rate. How many soma remain to be proofread?

Nominally, this indeed looks like a large change but most of the additional neurons were photoreceptors (R1-8) and almost all were from the optic lobes. This addition was expected and foretold with hatched lines in Figure 2 of the initial submission. While further proofreading might uncover additional neurons, there is currently no list of known unproofread neurons because of the through nature of our proofreading approach (see Proofreading strategy to complete the connectome in the methods section).

Reviewer 2

The authors have addressed my comments appropriately from the last round. In particular, my graduate student reports that the additions to the software are clear and helpful and make it more quickly accessible. The paper also reads more clearly and smoothly after the latest round of edits. The new data on the optic lobes is a welcome addition.

We thank the Reviewer for their thorough and constructive review that we believe made the manuscript better and clearer! We are also grateful to the graduate student who evaluated our access tools.

Minor

Line 110 – Admonishing readers whom to cite doesn't quite feel right. This sort of text seems like it belongs on the software front page, rather than in the text of a paper. Stressing the importance of the companion paper seems totally reasonable, of course.

We have now removed this sentence since the rest of that paragraph highlights the findings in the Schlegel et al. study and also refers to it as the 'companion' paper. We have now added a sentence in the methods about citations.

Line 429 – in degree or out degree here?

Thank you for catching this! Fixed.

I notice that most plots with axes have ticklabels but not ticks. See for instance 3dfgh, 4efg, etc. This is extremely unconventional. Even though I don't see readers trying to align datapoints with specific tick marks, it seems like it would be useful to identify what is being labeled with each number. In principle, readers should be able to try to make precise alignment between axes and data positions.

For the final version of the manuscript, we remade almost all plots with ticks.

If meshparty offers functionality to copy and paste links for 3D renderings so anyone can view it, it would be wonderful to include those links for readers.

MeshParty is a programmatic tool. We included an example for how to create renderings of neurons in FlyWire with MeshParty here: <https://github.com/seung-lab/FlyConnectome/blob/main/Renderings%20with%20MeshParty.ipynb> and provide extensive documentation here: <https://meshparty.readthedocs.io/en/latest/guide/visualization.html>

This last item is under minor because it's something that I think would be in the spirit of data openness that the authors advocate in their overall approach and in the discussion of the paper: Although Nature journals do not require it, it would be nice to distribute code to generate the paper's figure panels from the database.

We agree with the reviewer and will continue to work on more code examples for how to use the tools available to access the data. We will make code available to produce some of the figures.

Reviewer 3

As stated in my initial review, this is clearly a landmark paper that does a commendable job of describing the methods and gaining insights from this immense dataset. My previous comments have been addressed. I just have a few minor comments on the revised paper, noted below, which I'm sure the authors will easily address.

We thank the Reviewer for their thorough and constructive review that we believe made the manuscript better and clearer!

1) I noticed that even though >10k additional neurons have been annotated since the initial submission, some of the neuron numbers are smaller than before: for example, fewer neurons fully contained in the central brain (line 251) and fewer neurons in some nerves (Fig. 2D). It's not clear to me why the numbers decreased (were the neurons initially miscategorized?).

Most of the additional neurons were photoreceptors (R1-8) and almost all were from the optic lobes. This led to an increase in the sensory neuron and optic categories but had little to no impact on the other neuron categories. The mentioned minor decrease of the number of neurons in the central brain were indeed due to changes in the annotations. So, yes they were initially miscategorized.

2) Fig 3F: Now that the power law fit was removed from the graph, I'm not sure what the text is referring to in lines 359-362.

We thank the Reviewer for catching this. This paragraph should have been removed and will not be part of the final manuscript.

3) Fig 3G-H: May want to clarify what the colored dots are.

These are the statistics for the neurons in panel c. We only mentioned that in the description of panel c and now added this reference to the other panels as well.

4) Extended data Fig. 3-2A: Should include the color legend as in the previous extended data figure.

Fixed.

5) As Reviewer 2 mentioned in the first round, the L/R annotations in Fig 7A seem like they're flipped – why does "L" appear on the fly's right side if we're looking down on the fly? The authors said they checked this so I may be missing something, but if two reviewers had the same confusion then it probably warrants further clarification in the legend.

This is indeed due to the flip and we extended our description in the figure caption.

6) At least in my version, some of the figure panels look blurry (e.g. Fig. 2A-B).

We made sure that the final figures are in high resolution for the final version.